# An mRNA processing pathway suppresses metastasis by governing translational control from the nucleus

Albertas Navickas [1,2,3,4,9,10], Hosseinali Asgharian [1,2,3,4,10], Juliane Winkler [5], Lisa Fish[1,2,3,4], Kristle Garcia [1,2,3,4], Daniel Markett [1,2,3,4], Martin Dodel[6], Bruce Culbertson [1,2,3,4], Sohit Miglani [1,2,3,4], Tanvi Joshi [1,2,3,4], Keyi Yin[1,2,3,4], Phi Nguyen[1,2,3,4], Steven Zhang[1,2,3,4], Nicholas Stevers [1,2,3,4], Hun-Way Hwang [7], Faraz Mardakheh [6], Andrei Goga [3,5,8] & Hani Goodarzi [1,2,3,4] ✉

Cancer cells often co-opt post-transcriptional regulatory mechanisms to achieve pathologic expression of gene networks that drive metastasis. Translational control is a major regulatory hub in oncogenesis; however, its effects on cancer progression remain poorly understood. Here, to address this, we used ribosome profiling to compare genome-wide translation efficiencies of poorly and highly metastatic breast cancer cells and patient-derived xenografts. We developed dedicated regression-based methods to analyse ribosome profiling and alternative polyadenylation data, and identified heterogeneous nuclear ribonucleoprotein C (HNRNPC) as a translational controller of a specific mRNA regulon. We found that HNRNPC is downregulated in highly metastatic cells, which causes HNRNPC-bound mRNAs to undergo 3′ untranslated region lengthening and, subsequently, translational repression. We showed that modulating HNRNPC expression impacts the metastatic capacity of breast cancer cells in xenograft mouse models. In addition, the reduced expression of HNRNPC and its regulon is associated with the worse prognosis in breast cancer patient cohorts.

Cancer cells often co-opt post-transcriptional regulatory networks to activate pro-metastatic gene expression programmes[1–3]. Therefore, all stages of the messenger RNA life cycle—including alternative splicing, post-transcriptional modification, translation and decay— have been implicated in cancer progression[2–4]. Translational control has been increasingly recognized as an important regulatory node in tumourigenesis[5]; however, our understanding on how translational deregulation acts in the later stages of cancer remains incomplete.

To activate one of the main routes to metastasis, cancer cells have been shown to exploit the translational upregulation of several factors involved in the epithelial-to-mesenchymal transition[6,7]. Furthermore, numerous studies have observed a global tendency towards

[1]Department of Biochemistry and Biophysics, University of California, San Francisco, CA, USA. [2]Department of Urology, University of California, San Francisco, CA, USA. [3]Helen Diller Family Comprehensive Cancer Center, University of California, San Francisco, CA, USA. [4]Bakar Computational Health Sciences Institute, University of California, San Francisco, CA, USA. [5]Department of Cell and Tissue Biology, University of California, San Francisco, CA, USA. [6]Centre for Cancer Cell and Molecular Biology, Barts Cancer Institute, Queen Mary University of London, London, UK. [7]Department of Pathology, University of Pittsburgh, Pittsburgh, PA, USA. [8]Department of Medicine, University of California, San Francisco, CA, USA. [9]Present address: Institut Curie, CNRS UMR3348, INSERM U1278, Orsay, France. [10]These authors contributed equally: Albertas Navickas, Hosseinali Asgharian. ✉e-mail: hani.goodarzi@ucsf.edu

3′ untranslated region (UTR) shortening in cancer[8–12], suggesting consequences from reduced interactions with RNA-binding proteins (RBPs) and microRNAs (miRNAs)[13], including altered translation[5]. In some cases, these observations could be attributed to changes in the expression of specific mRNA cleavage and polyadenylation factors[10,11], although in many instances the underlying molecular mechanisms remain unknown. Similarly, we have previously demonstrated that the translational reprogramming that accompanies changes in transfer RNA expression landscape drives metastasis in breast cancer[14]. Importantly, a systematic characterization of translational control and its links to other aspects of RNA metabolism in metastasis is still lacking.

In this Article, we applied genome-wide experimental and computational approaches to address the changes in mRNA translation that accompany metastatic progression in breast cancer. We performed ribosome profiling in both cell line- and patient-derived models of breast cancer metastasis, and used Ribolog, a novel analytical framework, to identify the underlying regulatory programmes that govern changes in the translational control landscape. By applying these tools, we identified a functional interplay between nuclear RNA processing and translational control that disrupts the expression of a metastasis-suppressive regulon.

## Results

### Translational reprogramming accompanies breast cancer metastasis

To capture changes in the translational landscape that are associated with breast cancer metastasis, we performed ribosome profiling (Ribo-seq) on a commonly used triple receptor negative model of breast cancer metastasis, MDA-MB-231 breast cancer cells and their lung metastatic derivative cell line, MDA-LM2 (ref. 15). We predominantly recovered 33–34-nucleotide-long ribosome-protected mRNA footprints, aligning in frame with annotated coding sequences (CDSs)[16], confirming the high quality of our dataset (Extended Data Fig. 1a,b). We then sought to measure relative changes in translational activity genome-wide by calculating translational efficiency ratios (TERs) between MDA-LM2 and parental MDA-MB-231 cells.

To perform reliable differential analysis of Ribo-seq data and systematically account for possible confounders, we developed an analytical framework for comparison of translation efficiencies (TEs, representing the ratio between ribosome-protected mRNA footprint (RPF) and mRNA abundance sequencing read counts), aiming for as few a priori assumptions as possible. The resulting method, which we have named Ribolog, relies on logistic regression to model individual Ribo-seq and RNA sequencing (RNA-seq) reads. Ribolog calculates the log odds of RPF:RNA reads and its dependence on experimental covariates in order to estimate logTER (that is, log fold change in TE) and its associated raw and corrected P value in the coding transcriptome (for the details and advantages of this approach, see Methods).

First, we used Ribolog to calculate the TE changes between poorly and highly metastatic breast cancer cells, and detected numerous differentially translated mRNAs (Fig. 1a). We then assessed the impact of changes in TE on the proteome by comparing the abundance of proteins in MDA-LM2 and parental MDA-MB-231 cells using tandem mass tag labelling and mass spectrometry (TMT–MS). As expected, we observed broad changes in the proteome as cells become more metastatic (Extended Data Fig. 1c). Moreover, we observed that the changes in protein levels can be partially but not completely explained by changes in the mRNA levels ($R = 0.46$, $P < 1 \times 10^{-193}$ between mRNA and protein log fold changes), which points to regulators of protein synthesis and decay as another source of variation. Consistently, we observed that changes in protein levels showed a stronger correlation with changes in ribosome density than with differences in mRNA levels ($R = 0.53$, $P < 1 \times 10^{-270}$). To formalize differential TE as a key factor in the observed modulations, we corrected changes in the protein levels by their respective changes in mRNA abundance, and observed that the

resulting measures were significantly correlated with logTER values from Ribo-seq ($R = 0.3$, $P < 2 \times 10^{-16}$; Extended Data Fig. 1d). These findings suggest that post-transcriptional regulation of TE has a significant impact on protein levels in highly metastatic cells.

Given the extent of translational reprogramming observed in MDA-LM2 cells relative to their poorly metastatic parental line, we sought to systematically identify cis-regulatory elements in RNA that are significantly associated with the observed changes in TE. For this analysis, we used FIRE[17] algorithm to search for RNA motifs that are enriched in the 3′ UTRs of mRNAs with differential TE. FIRE identified poly(U) sequence motifs that were enriched in translationally repressed mRNAs in MDA-LM2 compared with parental cells (Fig. 1a). To extend our findings to other clinically relevant models of breast cancer metastasis, we also performed Ribo-seq on two sets of poorly and highly metastatic human breast cancer patient-derived xenografts (PDXs)[18–20] (Extended Data Fig. 1e,f). We then compared TEs in the two highly metastatic PDXs (HCI-001 and HCI-010) with those in the two poorly metastatic PDXs (HCI-002 and STG139). We observed broad differences in the translational landscape of these PDXs, and similar to the results from the breast cancer cell lines, we observed significantly reduced translation of mRNAs with poly(U) motifs in their 3′ UTRs (Fig. 1b).

### HNRNPC controls the translation of its 3′ UTR-bound regulon

Poly(U) motifs are recognized by many RBPs, and therefore function in a context-dependent manner[21]. To identify the most likely trans-factors interacting with the poly(U) sequences in translationally repressed mRNAs, we used information from the sequence context in which the poly(U) motifs are embedded. For this analysis we used DeepBind[22] algorithm and identified heterogeneous nuclear ribonucleoprotein C (HNRNPC) as the candidate most likely to bind the poly(U) motifs of interest (Extended Data Fig. 1g). In agreement with a potential role for HNRNPC involvement in a translational deregulation programme in metastatic breast cancer, HNRNPC was modestly but significantly downregulated in highly metastatic cells, both at the mRNA (log fold change −0.5, $P = 0.05$, determined by RNA-seq) and protein level (log fold change of −0.24, $P = 0.04$, determined by mass spectrometry (MS)[14]). HNRNPC ranks in the top 10% of proteins in MDA-MB-231 cells that can be detected by MS[23], and therefore a slight relative decrease in protein levels corresponds to a large decrease in absolute HNRNPC abundance.

To explore the possibility that HNRNPC is a trans-factor that binds the identified translational regulatory poly(U) elements, we performed HNRNPC cross-linking and immunoprecipitation coupled to sequencing (CLIP-seq)[24] in MDA-MB-231 cells. As controls, we also performed CLIP-seq for two other poly(U) binding proteins, TIA1 and ELAVL1, that ranked high in the DeepBind analysis (Extended Data Fig. 1g). In agreement with the existing data[25] and DeepBind predictions, poly(U) motifs were significantly enriched within HNRNPC-bound sequences (Fig. 1c). Furthermore, we detected a substantial amount of HNRNPC binding to poly(U) elements in 3′ UTRs across our own as well as previously published HNRNPC CLIP-seq datasets[25] (Extended Data Fig. 1h). Finally, if the poly(U) motifs in the 3′ UTRs of translationally repressed mRNAs are bound by HNRNPC, mRNAs that are bound by HNRNPC in their 3′ UTRs should also be translationally repressed in highly metastatic cells. To test this, we performed a gene-set enrichment analysis, using the set of HNRNPC-bound 3′ UTRs to assess their patterns of enrichment and depletion across the TE values from both breast cancer cell lines and PDXs. As shown in Fig. 1d,e (and Extended Data Fig. 1i,j), we observed a consistent enrichment of this HNRNPC regulon among the genes with lower TE in the highly metastatic cells. Importantly, HNRNPC regulon was substantially more translationally repressed than TIA1 or ELAVL1 targets (Extended Data Fig. 1k).

To confirm the causal role of HNRNPC in controlling the translation of this regulon, we used CRISPR interference[26] (CRISPRi) to knock down HNRNPC in MDA-MB-231 cells (fold change of 0.34,

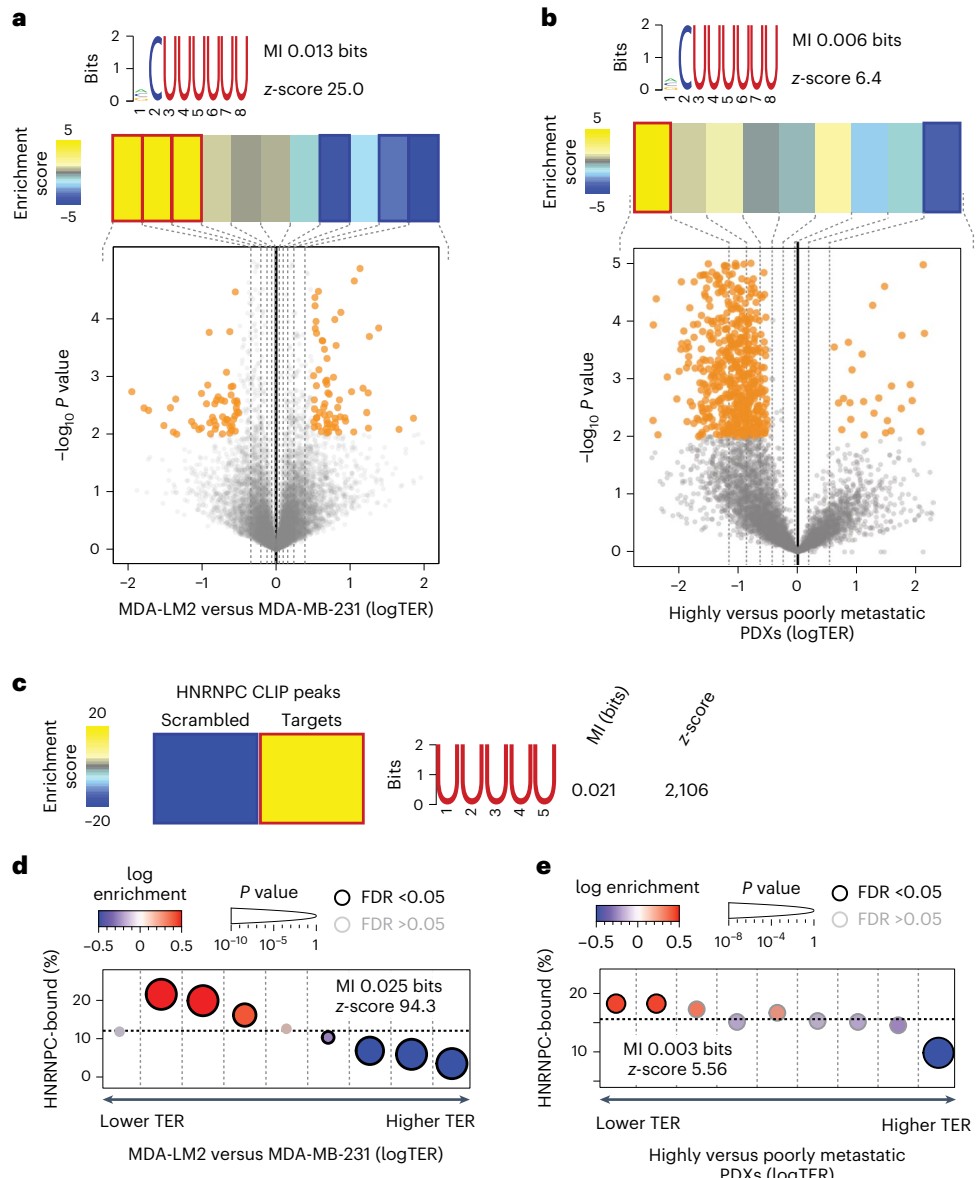

**Fig. 1 | HNRNPC target mRNAs are translationally repressed in highly metastatic breast cancer cells and PDXs. a**, Bottom: volcano plot showing the distribution of changes in TER (logTER) in MDA-LM2 compared with parental MDA-MB-231 cells. Statistically significant (logistic regression, $P < 0.01$) observations are highlighted in orange. Top: enrichment of the poly(U) motif in the mRNA 3′ UTRs as a function of logTER between MDA-LM2 and MDA-MB-231 cells. mRNAs are divided into equally populated bins based on their logTER (dashed vertical lines delineate the bins). Bins with significant enrichment (hypergeometric test, corrected $P < 0.05$; red) or depletion (blue) of poly(U) motifs are denoted with a bolded border. Also included are mutual information (MI) values and their associated $z$-scores. **b**, Volcano plot showing the distribution of changes in TE in highly versus poorly metastatic breast cancer PDXs, as described for **a**. **c**, Heat map showing the enrichment of poly(U) motifs

among the HNRNPC binding sites (as determined by CLIP-seq) as compared with scrambled sequences (with di-nucleotide frequency held constant). The bolded border denotes a statistically significant enrichment (hypergeometric test, corrected $P < 0.05$; red). MI value and associated $z$-score are shown. **d**, Enrichment of the HNRNPC target mRNAs as a function of logTER between MDA-LM2 and MDA-MB-231 cells. mRNAs are binned as in **a**; the $y$ axis shows the frequency of the HNRNPC targets (3′ UTR-bound) that we identified in each bin (dashed horizontal line denotes the average HNRNPC target frequency across all transcripts). Bins with significant enrichment (logistic regression, FDR <0.05; red) or depletion (blue) of HNRNPC targets are denoted with a black border. **e**, Enrichment patterns of HNRNPC target mRNAs as a function of logTER between highly and poorly metastatic breast cancer PDXs, as in **d**.

$P < 0.01$, determined by quantitative reverse transcription polymerase chain reaction (RT–qPCR)), and used Ribo-seq to compare TEs in control and HNRNPC-deficient cells. HNRNPC knockdown (KD) affected the translational landscape in MDA-MB-231 cells and, specifically, caused translational repression of HNRNPC target mRNAs (Fig. 2a and Extended Data Fig. 2a). We found that for the most part the same HNRNPC regulon mRNAs were translationally repressed in MDA-LM2 and HNRNPC KD cells, further highlighting the role of HNRNPC as a regulator of TE (Extended Data Fig. 2b). We individually

validated several targets translationally repressed in highly metastatic and HNRNPC KD cells (Extended Data Fig. 2c,d). However, HNRNPC is a predominantly nuclear protein and a known regulator of alternative splicing[27], a fact that is reflected in our CLIP-seq data as well on the basis of its pervasive binding to intronic sequences (Extended Data Fig. 1h). Therefore, it was unclear how HNRNPC, as a nuclear protein, could impact the translation of its targets in the cytoplasm. We thus hypothesized that HNRNPC might act indirectly to control the translation of its targets.

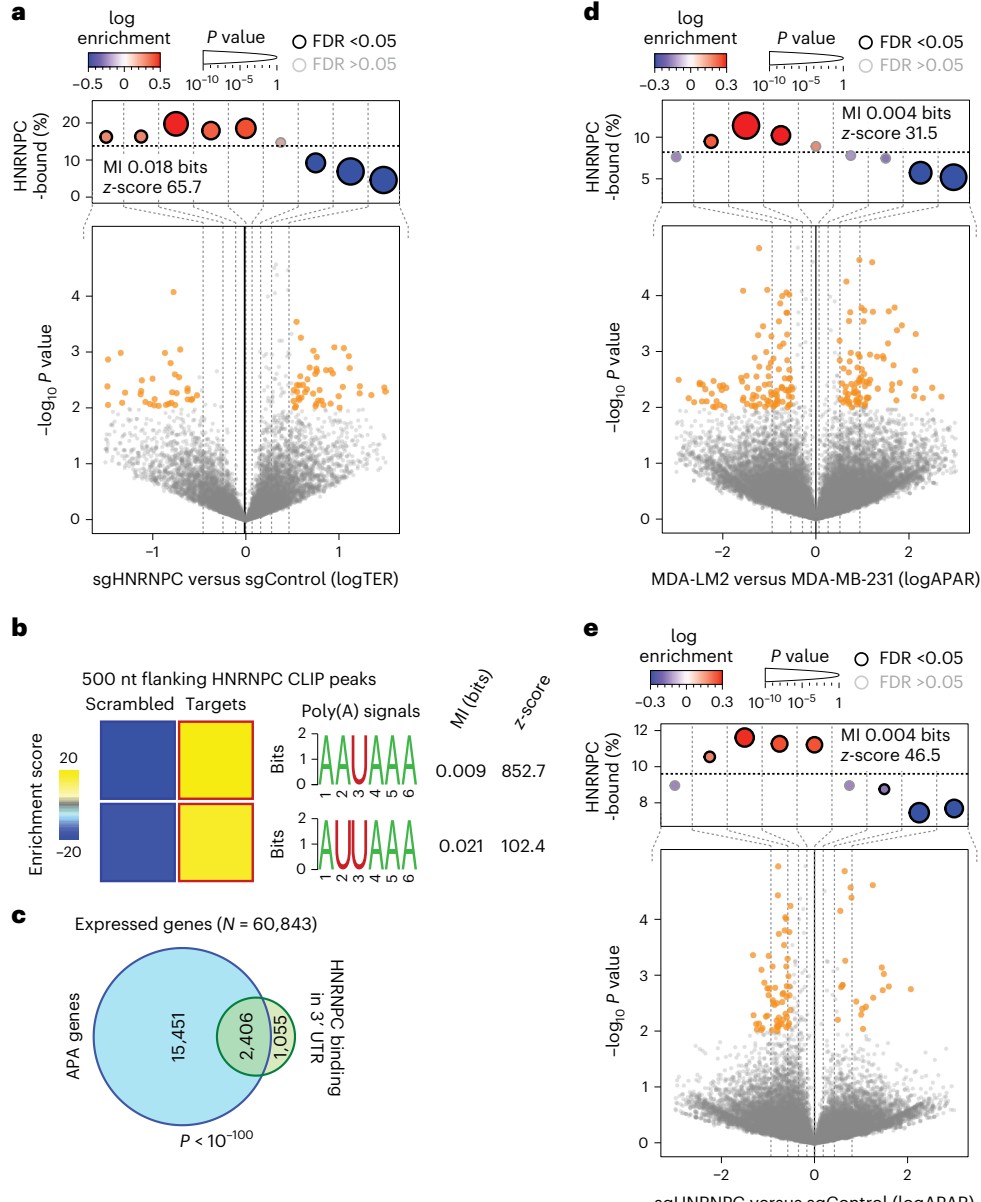

**Fig. 2 | HNRNPC binding impacts the translation and APA of its targets.**
**a**, Bottom: volcano plot showing the distribution of changes in TER (logTER) in sgHNRNPC compared with sgControl MDA-MB-231 cells. Statistically significant (logistic regression, $P < 0.01$) observations are highlighted in orange. Top: enrichment of the HNRNPC targets as a function of logTER between sgHNRNPC and sgControl cells. mRNAs are divided into equally populated bins according to logTER (dashed vertical lines delineate the bins); the $y$ axis shows the frequency of the HNRNPC targets that we identified in each bin (dashed horizontal line denotes the average HNRNPC target frequency across all transcripts). Bins with significant enrichment (logistic regression, FDR <0.05; red) or depletion (blue) of HNRNPC targets are denoted with a black border. Also included are mutual information (MI) values and their associated $z$-scores. **b**, Heat maps showing the enrichment of canonical poly(A) signals in the vicinity (500 nt flanking) of

HNRNPC binding peaks in 3′ UTRs (as determined by CLIP-seq). The bolded border denotes a statistically significant enrichment (hypergeometric test, corrected $P < 0.05$; red). MI values and associated $z$-scores are shown. **c**, Venn diagram showing the overlap between HNRNPC 3′ UTR target mRNAs and mRNAs showing APA. $P$ value calculated using hypergeometric test. **d**, Bottom: volcano plot showing distribution of changes in APA ratio (logAPAR; for detailed description, see Methods) in MDA-LM2 compared with MDA-MB-231 cells. Top: enrichment of the HNRNPC-bound 3′ UTRs as a function of APAR between MDA-LM2 and parental MDA-MB-231 cells; statistics as in **a**. **e**, Bottom: volcano plot showing distribution of changes in APAR in sgHNRNPC compared with sgControl cells. Top: enrichment of the HNRNPC-bound 3′ UTRs as a function of APAR between sgHNRNPC and sgControl cells; statistics as in **a**.

## HNRNPC controls the APA of its targets

To better capture the regulatory context within which HNRNPC functions, we performed a systematic search for additional *cis*-regulatory elements in the vicinity of HNRNPC binding sites on 3′ UTRs. Interestingly, as shown in Fig. 2b, we observed a highly significant enrichment of canonical poly(A) signals (AAUAAA and AUUAAA)[13] within the 500 nucleotide flanking regions of HNRNPC CLIP-seq peaks in 3′ UTRs.

This observation led us to the hypothesis that HNRNPC controls translation by regulating 3′ UTR length via alternative polyadenylation (APA) site selection. Consistently, the majority of HNRNPC 3′ UTR targets carry annotated APA sites, significantly more than expected by chance (Fig. 2c). In line with this, HNRNPC has been previously implicated in the control of APA in other studies; however, the mechanism through which HNRNPC impacts polyadenylation remained uncertain[28,29].

To confirm that the changes in TE we observed in highly metastatic cells coincide with alteration in poly(A) site selection, we performed mRNA 3′-end sequencing in the parental MDA-MB-231 and MDA-LM2 cells, as well as in control and HNRNPC KD cells. To measure changes in poly(A) site selection, we tabulated the number of reads that mapped to each annotated poly(A) site across the transcriptome. We then used a quantity we call logAPAR (log fold change in proximal-to-distal APA ratio) to identify poly(A) site switches between conditions.

To assess the statistical significance of the observed changes in APA (that is, non-zero logAPAR), we developed a novel method named APAlog. APAlog runs multinomial logistic regression to test differential usage of two or more poly(A) sites per transcript, and simultaneously calculates logAPAR and its associated raw and corrected *P* value. APAlog functions in three modes: (1) identifying transcripts with the highest overall variability in poly(A) site usage across conditions, (2) comparing all non-canonical poly(A) sites with one canonical (reference) poly(A) site per transcript and (3) comparing all pairs of poly(A) sites per transcript (for details, see Methods). Using APAlog, we found that HNRNPC target mRNAs undergo 3′ UTR lengthening (that is, proximal-to-distal poly(A) site switch) in MDA-LM2 (Fig. 2d and Extended Data Fig. 2e) and HNRNPC KD (Fig. 2e and Extended Data Fig. 2f,g) cells.

To confirm our observations in an exogenous setting, we designed a massively parallel reporter assay that simultaneously monitors reporter mRNA poly(A) site selection and translation. We used a bidirectional promoter vector, driving mCherry and blue fluorescent protein (BFP) expression, and containing two APA sites downstream of BFP. We then cloned a library of CLIP-seq-derived HNRNPC binding sites or matched scrambled controls upstream of the proximal poly(A) site in the reporter (Extended Data Fig. 2h). We assessed the reporter protein expression in MDA-MB-231 cells by flow cytometry, and the poly(A) site choice by 3′-end RNA-seq, in both total and sorted cell populations, where we separated the cells on the basis of the BFP/mCherry ratio. HNRNPC KD led to a lower BFP/mCherry signal in bulk (Extended Data Fig. 2i) and a preferential choice of the distal poly(A) site in reporter mRNAs bearing HNRNPC binding sites when compared with matched shuffled controls (Extended Data Fig. 2j). Furthermore, lower reporter protein expression was significantly associated with longer reporter 3′ UTRs with HNRNPC binding sites as compared with shuffled controls. Importantly, this observation was dependent on HNRNPC (Extended Data Fig. 2k). In sum, our results demonstrate that HNRNPC acts as a direct mediator of an alternative poly(A) site selection programme in metastatic breast cancer with broad consequences on the translational landscape.

## HNRNPC acts with PABPC4 to control APA

To obtain insights into how HNRNPC differentially controls polyadenylation of its target RNAs in parental MDA-MB-231 and MDA-LM2 cells, we immunoprecipitated HNRNPC in both cell lines and identified interacting proteins by MS. We specifically searched for ways in which the HNRNPC interactome switches between poorly and highly metastatic cells. First, in agreement with the canonical role of HNRNPC as a splicing regulator, we detected numerous splicing factors among HNRNPC interactors (Supplementary Table 1). However, when comparing the HNRNPC interactomes between poorly and highly metastatic cells, we did not observe broad changes in the interaction between HNRNPC and other splicing factors. In contrast, we found that a group of proteins implicated in mRNA transport from the nucleus was significantly depleted from the HNRNPC interactome in MDA-LM2 cells (Extended Data Fig. 3a). Upon closer inspection of the proteins in this set, we noted multiple poly(A)-binding proteins that play canonical roles in the mRNA nuclear export cascade[30]. We found that PABPC4 and PABPN1 were among the depleted HNRNPC interactors in MDA-LM2 cells (Fig. 3a and Extended Data Fig. 3b). We individually confirmed an RNA-dependent interaction between HNRNPC and PABPC4 or PABPN1 in MDA-MB-231 cells by co-immunoprecipitation (co-IP) and western blotting (Fig. 3b).

Next, to assess which, if any, of these factors acts in concert with HNRNPC to regulate poly(A) site selection, we depleted PABPC4 and PABPN1 in MDA-MB-231 cells using CRISPRi (fold change of 0.21 and 0.69, respectively, *P* < 0.05, as determined by RT–qPCR) and employed 3′-end RNA-seq to compare the APA landscapes in control and KD cells. We found that PABPC4 KD, but not PABPN1 KD, resulted in APA changes similar to those observed in HNRNPC-deficient cells (Extended Data Fig. 3c,d). Importantly, PABPC4 was also downregulated in MDA-LM2 cells compared with parental MDA-MB-231 cells at both the mRNA (log fold change −0.9, *P* = 0.02, determined by RNA-seq) and protein level (log fold change −0.5, *P* < 0.002, determined by MS[14]).

To further investigate the HNRNPC-PABPC4 regulon in cells, we performed PABPC4 PAPERCLIP[31] in MDA-MB-231 cells. First, as expected, and similar to our observations for HNRNPC, we noted that PABPC4 peaks were significantly enriched in the vicinity of canonical poly(A) signals (Extended Data Fig. 3e). Moreover, we observed that the majority (96%) of HNRNPC 3′ UTR targets were also bound by PABPC4 in vivo (Fig. 3c). To confirm that HNRNPC and PABPC4 act in concert to control the APA of HNRNPC targets, we performed 3′-end RNA-seq comparing HNRNPC/PABPC4 double KD cells with PABPC4 KD alone. Unlike Fig. 2e, in this comparison, we did not observe the significant proximal-to-distal switching within the HNRNPC regulon that we had observed in HNRNPC and PABPC4 single KD cells (Fig. 3d and Extended Data Fig. 3f). This finding indicates that the regulatory function of HNRNPC in poly(A) site selection is contingent on PABPC4 expression, and demonstrates an epistatic interaction between these two genes.

As genes associated with mRNA nuclear export were depleted from the HNRNPC interactome in highly metastatic cells, we considered that HNRNPC could impact the nuclear export of its targets. To address this, we performed subcellular fractionation of MDA-MB-231 and MDA-LM2, as well as control and HNRNPC KD cells, and extracted their nuclear and cytoplasmic RNA. As expected, we recovered mature mRNA predominantly in the cytoplasmic fraction, while pre-mRNA or known nucleus-retained RNA was recovered exclusively in the nuclear fraction (Extended Data Fig. 3g). However, we did not detect any differences between the two compartments when we assessed changes in the APA using 3′-end RNA-seq and APAlog (Extended Data Fig. 3h,i). Importantly, using the data obtained with cytoplasmic RNA, we recapitulated our findings from Fig. 2d,e, showing that longer 3′ UTR-bearing HNRNPC target mRNAs are exported to the cytoplasm, in both highly metastatic and HNRNPC-deficient cells (Extended Data Fig. 3j,k). We also repeated Ribo-seq analysis from the cytoplasmic fraction described above. Here, again, we largely recapitulated our data from Figs. 1a and 2a, showing that HNRNPC targets undergo translational repression in the cytoplasm in HNRNPC low conditions (Extended Data Fig. 3l,m).

## RNA interference pathway targets the HNRNPC regulon

We reasoned that the extended 3′ UTRs carry more translationally repressive *cis*-regulatory elements, such as miRNA binding sites. Consistent with this hypothesis, we observed a significant overlap between HNRNPC-bound extended 3′ UTRs and miRNA/argonaute (AGO2) targets[32] (Fig. 3e). miRNAs are known repressors of mRNA stability and translation[33]. We observed, accordingly, that mRNAs both bound by HNRNPC and containing functional miRNA binding sites[32] had significantly lower TE than non-target mRNAs in MDA-LM2 compared with parental MDA-MB-231 and in HNRNPC-deficient compared with control cells (Extended Data Fig. 4a,b). Furthermore, translationally repressed mRNAs in MDA-LM2 and HNRNPC-deficient cells were enriched among AGO2 targets (Fig. 3f and Extended Data Fig. 4c). To validate that miRNAs contribute to the translational repression of HNRNPC targets via an AGO2-mediated mechanism, we performed Ribo-seq in control and HNRNPC-depleted MDA-MB-231 cells, in both control and AGO2 KD backgrounds (AGO2 fold change of 0.55, *P* < 0.05, as determined by RT–qPCR). We found that HNRNPC-dependent translational repression

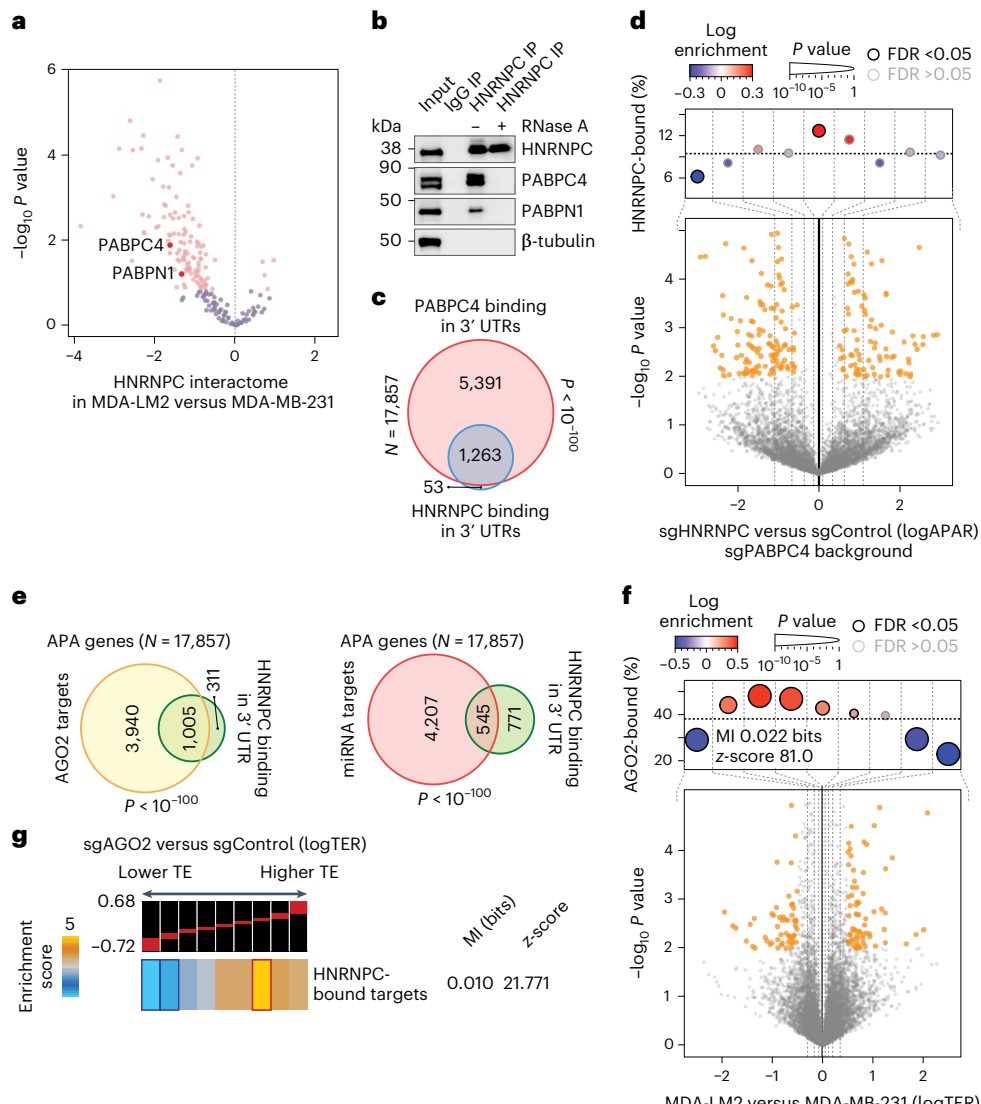

**Fig. 3 | PABPC4 acts in concert with HNRNPC to control APA and directs its target mRNAs to AGO2-dependent translational repression. a**, Volcano plot showing the distribution of changes in relative protein interaction with HNRNPC (as determined by HNRNPC or control (isotype IgG) co-IP–MS) in MDA-LM2 compared with MDA-MB-231 cells. Statistically significant (FDR-adjusted *P* value <0.25, generalized linear model controlling for input abundances) observations are highlighted in pink. **b**, Co-IPs of HNRNPC or control IgG were analysed by western blotting. RNase A was included in the lysates where indicated. Representative image from two independent experiments. **c**, Venn diagram showing the overlap between HNRNPC and PABPC4 3′ UTR targets (as determined by CLIP-seq and PAPER-CLIP, respectively). *P* value calculated using hypergeometric test. **d**, Bottom: volcano plot showing distribution of changes in APA ratio (logAPAR) in HNRNPC/PABPC4 double KD compared with PABPC4 KD MDA-MB-231 cells. Top: enrichment of the HNRNPC-bound 3′ UTRs as a

function of APAR between sgHNRNPC/sgPABPC4 and sgControl/sgPABPC4 cells; statistics as in Fig. 2a. **e**, Venn diagram showing the overlap between HNRNPC 3′ UTR targets and AGO2 bound mRNAs (top) or miRNA target mRNAs (bottom), as determined by CLIP-seq analyses. *P* values calculated using hypergeometric test. **f**, Bottom: volcano plot showing distribution of changes in TER (logTER) in sgAGO2 compared with sgControl MDA-MB-231 cells. Top: enrichment of the AGO2 targets as a function of logTER between sgAGO2 and sgControl cells; statistics as in Fig. 2a. **g**, Enrichment of the HNRNPC targets as a function of logTER between sgAGO2 and sgControl cells. mRNAs are distributed into equally populated bins according to their logTER (the red bars on the black background show the range of values in each bin). Bins with significant enrichment (hypergeometric test, corrected *P* < 0.05; red) or depletion (blue) of HNRNPC targets (3′ UTR-bound) are denoted with a bolded border. Also included are mutual information (MI) value and its associated *z*-score.

was contingent on AGO2 expression (Extended Data Fig. 4d). Similarly, when we compared TEs in control and AGO2 KD cells, we found that mRNAs with higher TEs in AGO2-depleted cells were enriched in HNRNPC-bound transcripts (Fig. 3g).

### HNRNPC and PABPC4 act as suppressors of metastasis in xenograft models

We next used a xenograft mouse model of metastasis to measure the impact of perturbing this HNRNPC-mediated pathway on the metastatic capacity of the cell. We performed lung colonization assays in NOD

*scid* gamma (NSG) mice by intravenously injecting control and HNRNPC KD cells, constitutively expressing luciferase. We monitored the metastatic burden in the lungs of these mice by in vivo bioluminescence imaging, and observed an over ten-fold increase in lung colonization capacity induced by HNRNPC KD (Fig. 4a). To ensure that these findings are generalizable to other genetic backgrounds, we repeated the experiment with HCC1806 breast cancer cells and observed a consistent increase in the metastatic capacity of these cells upon HNRNPC downregulation (Extended Data Fig. 4e). Conversely, overexpressing HNRNPC in MDA-LM2 cells reduced the metastatic colonization by breast cancer cells (Fig. 4b).

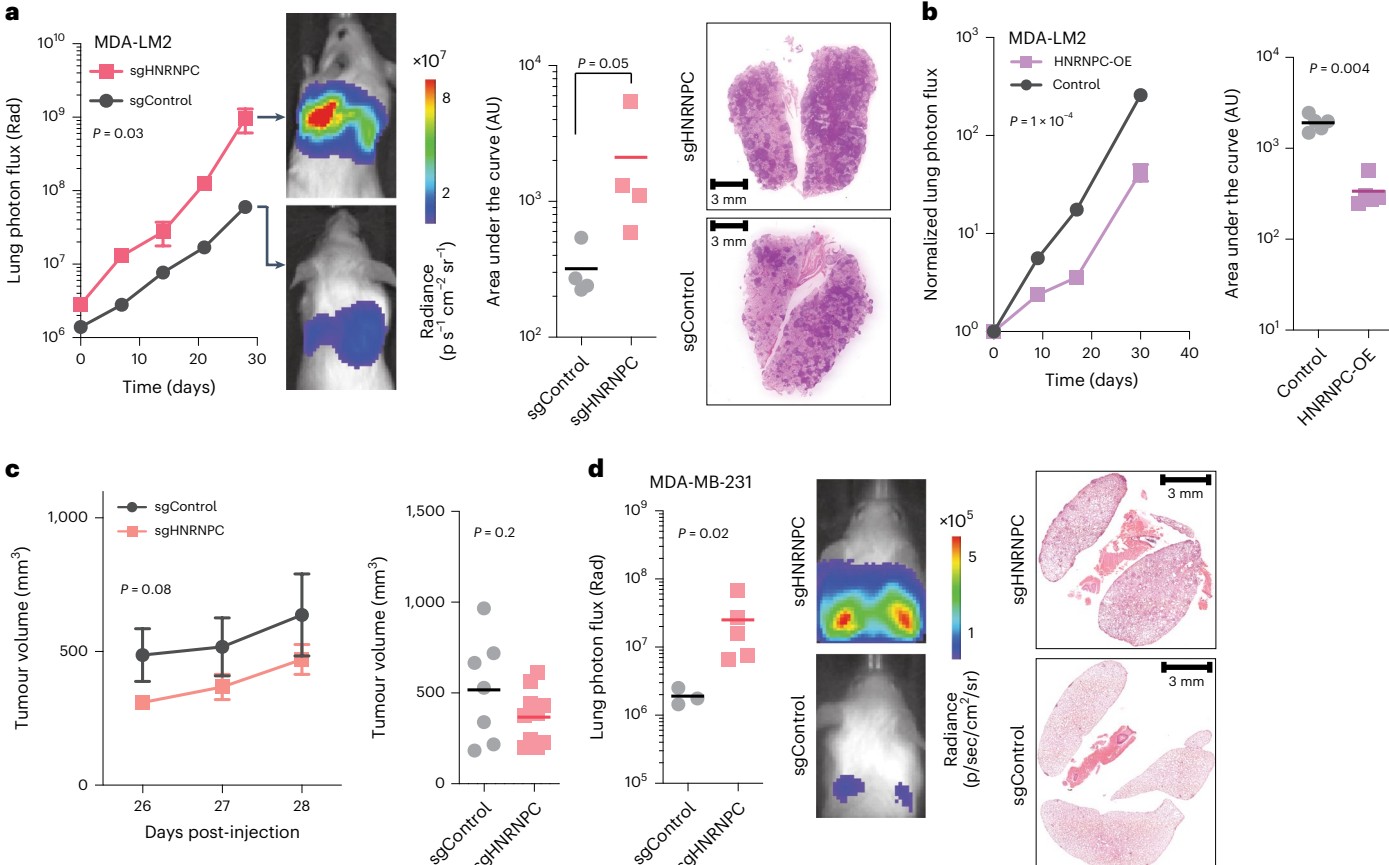

**Fig. 4 | HNRNPC levels impact in vivo metastatic colonization of breast cancer cells. a**, MDA-LM2 cells stably expressing sgHNRNPC or sgControl were injected via tail vein into NSG mice. Bioluminescence was measured at the indicated times (mean values are shown, error bars indicating standard error of the mean (s.e.m.); *P* value calculated using two-way analysis of variance (ANOVA)); area under the curve was measured at the final timepoint (*P* value calculated using one-tailed Mann–Whitney *U* test). Lung sections were stained with H&E (representative images shown). *n* = 4–5 mice per cohort. **b**, MDA-LM2 cells stably overexpressing mCherry or HNRNPC were injected via tail vein into NSG mice. Bioluminescence was measured at the indicated times (mean values are shown, error bars indicating s.e.m.; *P* value calculated using two-way ANOVA); area under the curve was measured at the final timepoint (*P* value calculated using one-tailed

Mann–Whitney *U* test). *n* = 4–5 mice per cohort. **c**, MDA-MB-231 cells stably expressing sgHNRNPC or sgControl were injected orthotopically into mammary fat pads of NSG mice. Tumour volume was measured at indicated times (mean values are shown, error bars indicating s.e.m.; *P* value calculated using two-way ANOVA); tumour volume was compared at the time of tumour resection (*P* value calculated using one-tailed Mann–Whitney *U* test). **d**, MDA-MB-231 cells stably expressing sgHNRNPC or sgControl were injected orthotopically into mammary fat pads of NSG mice. Four weeks later, the primary tumours were resected and lung colonization was measured at indicated times. Lung bioluminescence was measured at the final timepoint (*P* value calculated using one-tailed Mann–Whitney *U* test). Lung sections were stained with H&E (representative images shown). *n* = 4–5 mice per cohort.

Intravenous injection of cancer cells in mice recapitulates the last steps of metastasis, including extravasation and proliferation in the distant tissue. To assess the role of HNRNPC in the earlier steps of metastasis, we performed spontaneous metastasis assays by orthotopically injecting control and HNRNPC-depleted cells into the mammary fat pads of NSG mice. After primary tumour resection, we followed the lung colonization by cancer cells using in vivo imaging. While HNRNPC KD cells showed slightly (albeit non-significantly) reduced primary tumour volumes (Fig. 4c), it had a markedly and significantly increased metastatic lung colonization (Fig. 4d), in agreement with our data from intravenous injection assays (Fig. 4a).

In its canonical role, HNRNPC acts as a regulator of RNA splicing, and its function in metastasis may in fact be a consequence of these parallel regulatory programmes. To assess this possibility, we sought to independently test PABPC4, which acts in concert with HNRNPC to control the APA of its targets. For this, we compared the metastatic lung colonization by PABPC4 KD, as well as PABPC4/HNRNPC double KD and control MDA-MB-231 cells (Extended Data Fig. 4f). In line with PABPC4 controlling the APA of a metastasis-associated mRNA regulon, PABPC4-depleted cells showed significantly increased metastatic

potential when compared with control cells. More importantly, knocking down HNRNPC in the PABPC4 KD background did not result in an increase metastatic potential of cells. This is consistent with HNRNPC and PABPC4 acting as components of the same regulatory pathway, and showing an epistatic genetic interaction.

**PDLIM5 acts downstream of HNRNPC to suppress metastasis**
To better understand how the deregulation of APA and TE leads to increased metastatic potential, we sought to identify relevant targets downstream of the HNRNPC–PABPC4 regulatory axis. First, to complement our ribosome profiling results, we also compared the protein abundances in control and HNRNPC KD cells using TMT-MS. We observed that (1) consistent with its role in translational control, a large number of proteins were dysregulated upon HNRNPC depletion (Extended Data Fig. 5a), and (2) changes in the protein landscape of HNRNPC KD cells were significantly correlated with those between highly and poorly metastatic cells (Extended Data Fig. 5b). In other words, a significant portion of changes in the TE of MDA-LM2 cells relative to parental MDA-MB-231 can be explained by lower HNRNPC activity. Furthermore, gene-set enrichment analysis of this data revealed

that HNRNPC KD caused the downregulation of proteins interacting with SH3 (Src homology 3) domain proteins and actin filaments, among other Gene Ontology terms (Extended Data Fig. 5c).

To identify genes that are part of this HNRNPC regulon and act downstream of this pathway to influence metastatic progression, we systematically integrated the datasets comparing poorly and highly metastatic cells, as well as HNRNPC KD and control cells. We specifically searched for mRNAs that (1) were translationally repressed and (2) demonstrated proximal-to-distal poly(A) site switching in MDA-LM2 and HNRNPC-deficient cells (16 and 30 transcripts, respectively). We focused on transcripts that (3) were bound by HNRNPC and PABPC4 (1,263 mRNAs) to identify the direct downstream targets. This approach nominated PDLIM5 (Fig. 5a), a member of cytoskeleton-associated protein family[34], as a robust target of this HNRNPC-mediated pathway.

In agreement with our ribosome profiling and TMT–MS data (Extended Data Fig. 5d), we detected lower levels of PDLIM5 protein in MDA-LM2 and HNRNPC-deficient cells, as compared with control MDA-MB-231 cells (Fig. 5b). In contrast, PDLIM5 mRNA abundance was similar in these conditions, which is consistent with PDLIM5 protein levels being regulated at the translational level (Extended Data Fig. 5e). We also confirmed by isoform-specific RT–qPCR the proximal-to-distal poly(A) site switch for PDLIM5 mRNA (Fig. 5c). To further confirm the regulation conferred by the 3′ regulatory sequences in PDLIM5 mRNA, we constructed a series of reporters, where we fused N-terminal FLAG tag with PDLIM5 CDS and several genetically encoded variants of PDLIM5 3′ UTR (Extended Data Fig. 5f). We found that the sequence downstream of the proximal poly(A) site in the PDLIM5 3′ UTR is responsible for the decrease in reporter protein quantity, as compared with full-length 3′ UTR (Fig. 5d).

To assess the role of PDLIM5 in breast cancer progression, we performed in vivo lung colonization assays with control and PDLIM5-deficient MDA-MB-231 and HCC1806 cells. As shown in Fig. 5e and Extended Data Fig. 5g, PDLIM5 KD (fold change of 0.18, $P < 0.01$, as determined by RT–qPCR) led to a significant increase in metastatic lung colonization of xenografted mice. In our model, PDLIM5 depletion impacts breast cancer progression due to HNRNPC deficiency. Indeed, an ectopic PDLIM5 expression in HNRNPC-deficient cells was sufficient to reduce the lung metastatic burden to the levels similar to control cells with intact HNRNPC expression (Fig. 5f), in line with HNRNPC acting upstream of PDLIM5 to control breast cancer metastasis.

Cancer cells exploit multiple phenotypic routes towards metastasis, including modulating their proliferative activity, invasiveness, migration, as well as the capacity to survive in isolation. To test if HNRNPC and PDLIM5 contribute to any of these phenotypes, we performed proliferation, colony formation, migration and invasion assays with control, and HNRNPC- or PDLIM5-perturbed cells. In agreement with our lung colonization assays in vivo (Figs. 4 and 5), HNRNPC and PDLIM5 KDs caused an increase, while HNRNPC overexpression (OE) caused a decrease, in cell migration and invasive capacity in vitro (Extended Data Fig. 5h,i). These observations contrasted with the reduced proliferation and colony formation of HNRNPC-perturbed cells (Extended Data Fig. 5j–l), consistently with primary tumour growth rates of HNRNPC KD cells in vivo (Fig. 4c). Similar to our in vivo data (Fig. 5f), ectopic PDLIM5 expression rescued the proliferation and colony formation defects of HNRNPC KD cells (Extended Data Fig. 5m).

### The HNRNPC–PABPC4 regulatory axis is linked with clinical outcomes

To confirm that our findings in xenograft models are generalizable to human disease, we performed survival analysis on publicly available datasets from breast cancer patients. We found that lower HNRNPC expression in breast cancer tumours was significantly associated with lower overall, disease-free and distant metastasis-free survival, both in individual cohorts and in meta-analyses (Fig. 6a–c and Extended Data Fig. 6a–c). We detected that HNRNPC expression was negatively

associated with tumour stage (Fig. 6d) and presence of metastasis (Fig. 6e), but not with tumour subtype (Extended Data Fig. 6d). Notably, HNRNPC expression remained a significant covariate in a Cox proportional hazards model even after controlling for other known prognostic metrics, such as tumour stage or received treatment (Extended Data Fig. 6e).

As we found HNRNPC to act upstream of a metastasis-suppressive translational programme, we identified a set of HNRNPC mRNA targets, translationally repressed and undergoing proximal-to-distal poly(A) site switching in MDA-LM2 cells to define a translational HNRNPC target signature. We observed that, in proteomic datasets from patients with breast cancer (CPTAC), lower protein levels of the HNRNPC signature, as an aggregate, were significantly associated with lower overall and progression-free survival (Extended Data Fig. 6f,g). In line with PABPC4 acting together with HNRNPC, lower PABPC4 expression was associated with worse prognostic metrics and disease progression in breast cancer patient cohorts (Fig. 6f,g and Extended Data Fig. 6h). Furthermore, in agreement with PDLIM5 being a functional effector downstream of the HNRNPC–PABPC4 axis, PDLIM5 expression was also associated with survival of patients with breast cancer (Fig. 6h and Extended Data Fig. 6i,j).

Finally, we asked whether the impact of the HNRNPC–PABPC4 deficiency in highly metastatic cells could be reversed as a potential therapeutic strategy to prevent metastasis. Recently, a target agnostic chemical screen was used to identify small molecules that impact APA or transcription termination[35]. We chose to test T4, a drug that was reported to induce distal-to-proximal poly(A) site switch[35], which is the opposite of the observed 3′ UTR lengthening of HNRNPC targets in MDA-LM2 cells. We first confirmed that treating MDA-MB-231 cells with 5 μM T4 for 6 h induced predominantly distal-to-proximal poly(A) site switching, as assessed by 3′ end RNA-seq (Extended Data Fig. 6k). Importantly, we observed that T4 treatment can reverse the impact of HNRNPC KD on APA site selection (Extended Data Fig. 6l,m). These results suggested that T4 could potentially counteract the 3′ UTR lengthening of HNRNPC targets, and compromise the pro-metastatic programme instigated by HNRNPC deficiency. To test this possibility, we first performed dose–response measurements of T4 in MDA-LM2 cells (Extended Data Fig. 6n), and treated these cells with the 20% of maximal inhibitory concentration (IC20) of the drug (3 μM) or vehicle control for 6 h. The effect of T4 treatment on the APA remained stable for 24 h, and significant for up to 72 h post drug withdrawal (Extended Data Fig. 6o). We then performed lung colonization assays to measure changes in the metastatic capacity of cells treated in vitro with T4. As shown in Fig. 6i, T4-treated MDA-LM2 cells showed significantly reduced metastatic capacity as compared with vehicle-treated cells. Finally, we assessed if systemic treatment with T4 could prevent metastatic lung colonization. Indeed, intraperitoneal injection of T4 significantly impaired the lung colonization by MDA-LM2 cells as compared with vehicle control treatment (Fig. 6j). These observations indicate that reversing the regulatory consequences of HNRNPC deficiency can restore the metastasis-suppressive activity of its target regulon.

## Discussion

In this study, we show that increased metastatic potential in breast cancer cell lines and PDXs is accompanied by a broad remodelling of the translational landscape, as demonstrated by genome-wide TE measurements derived from Ribo-seq. Using unbiased computational approaches, we discovered that translationally downregulated mRNAs in highly metastatic breast cancer cell lines and PDXs showed enriched poly(U) motifs in their 3′ UTRs. We identified a link between HNRNPC binding to these sequence elements and translational control of the bound transcript, showing that this mechanism relies on control of the APA.

We found that HNRNPC deficiency resulted in increased usage of distal poly(A) sites of target transcripts, leading to 3′ UTR lengthening of HNRNPC targets, consistent with previous reports[28,29]. It was

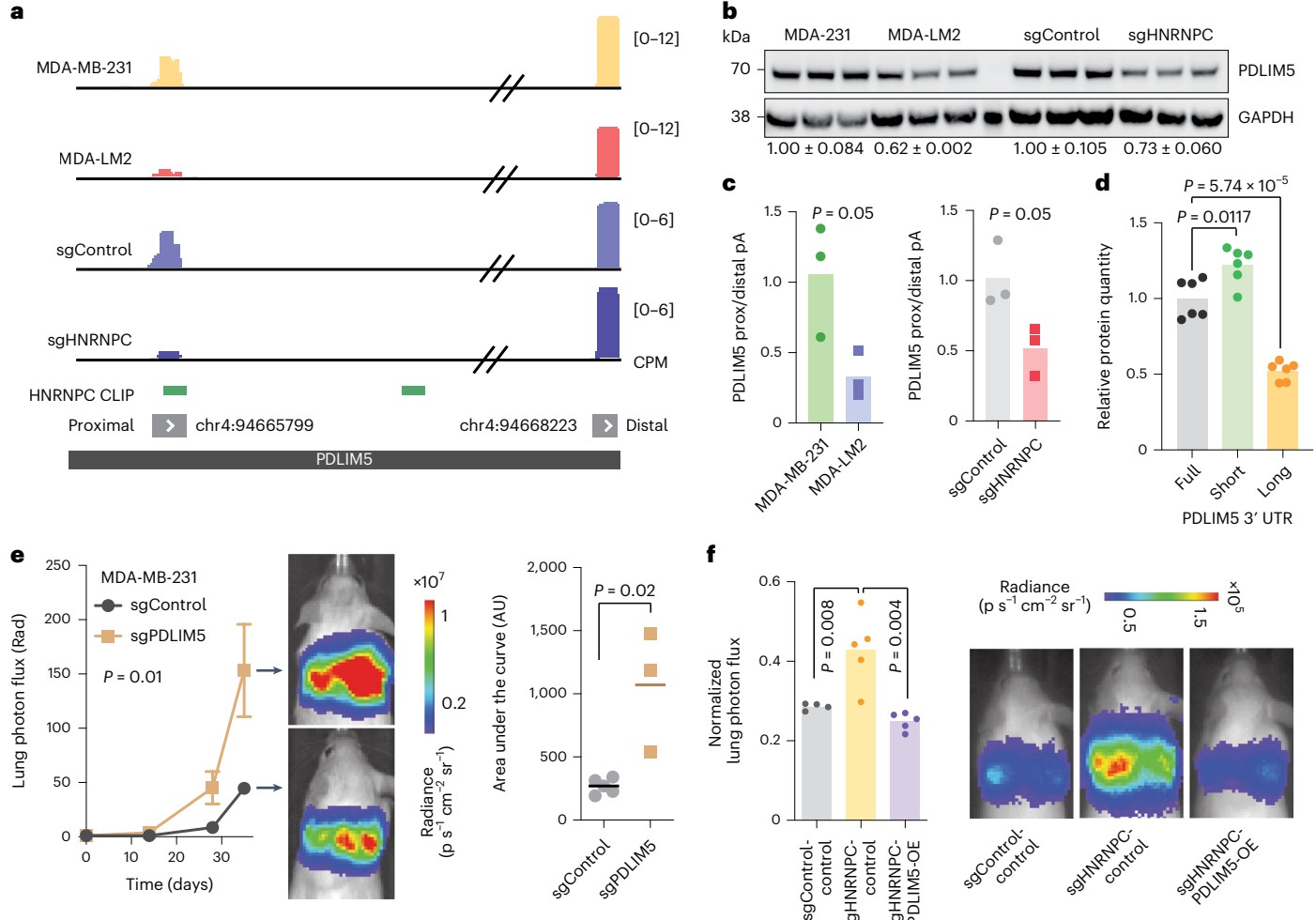

**Fig. 5 | PDLIM5 acts downstream of HNRNPC to suppress breast cancer metastasis. a**, The schematics of PDLIM5 3′ UTR, showing proximal and distal poly(A) sites, HNRNPC binding sites as determined by CLIP-seq, and 3′-end RNA-seq results in MDA-MB-231 and MDA-LM2, as well as sgControl and sgHNRNPC-expressing cells. CPM, counts per million. **b**, Western blot analysis of PDLIM5 protein in MDA-MB-231 and MDA-LM2, as well as sgControl and sgHNRNPC-expressing cells. Relative PDLIM5 quantity normalized to GAPDH is indicated below, along standard errors. **c**, Quantification of relative PDLIM5 proximal to distal poly(A) site usage in MDA-MB-231 and MDA-LM2 cells (left) or sgControl and sgHNRNPC cells (right), as determined by isoform-specific RT–qPCR. $n = 3$ biological replicates. $P$ values calculated using one-tailed Mann–Whitney $U$ test. **d**, Relative reporter FLAG-PDLIM5 protein quantity (normalized to GAPDH),

expressing full-length, short and long PDLIM5 3′ UTRs, as illustrated in Extended Data Fig. 5f. $n = 6$ biological replicates. $P$ value calculated using two-tailed unpaired $t$-test. **e**, MDA-MB-231 cells stably expressing sgPDLIM5 or sgControl were injected via tail vein into NSG mice. Bioluminescence was measured at the indicated times (mean values are shown, error bars indicating standard error of the mean; $P$ value calculated using two-way analysis of variance); area under the curve was measured at the final timepoint ($P$ value calculated using one-tailed Mann–Whitney $U$ test). $n = 3$ and 5 mice per cohort. **f**, MDA-MB-231 cells stably expressing sgControl or sgHNRNPC and mCherry (control) or PDLIM5 (PDLIM5-OE) were injected via tail vein into NSG mice. Bioluminescence was measured at the final timepoint ($P$ value calculated using one-tailed Mann–Whitney $U$ test). $n = 4$–5 mice per cohort.

suggested that HNRNPC masks strong distal poly(A) sites, thereby promoting usage of weaker proximal sites[28]. In line with this, we found that canonical poly(A) signals were in close proximity to HNRNPC-bound poly(U) motifs. We could counteract the proximal-to-distal poly(A) site switch caused by HNRNPC deficiency with T4, a small molecule, promoting the distal-to-proximal switch[35]. While the mechanism of action of T4 is not completely clear, it alters the expression levels of multiple splicing and cleavage and polyadenylation factors[35], emphasizing the interplay between the two pathways.

We also showed that PABPC4 bound HNRNPC targets in vivo and interacted with HNRNPC in controlling APA. PABPC4 is a nucleus–cytoplasm shuttling factor, and is known to have context-dependent functions, overlapping those of other poly(A) binding proteins[30]. PABPC4 is critical for the differentiation of erythroid cells, via an interplay between AU-rich elements in 3′ UTR of target mRNAs and the shortening of poly(A) tails[36]. Poly(A) tail shortening is a well-known mechanism

in promoting mRNA decay and downregulating translation[37]. It is possible that reduced PABPC4 expression in highly metastatic cells contributes to the lower TE of joint HNRNPC–PABPC4 targets via poly(A) tail shortening.

Our data suggest that the long form 3′ UTRs of HNRNPC target mRNAs harbour a greater number of miRNA binding sites and thus are more susceptible to translational repression via argonaute-mediated RNA interference[8,33], although we cannot exclude other RBPs participating in this mechanism. While some studies suggest that miRNAs have a limited impact on global translational repression and destabilization of APA targets[38,39], our data highlight a case where a subset of mRNAs—HNRNPC targets with 3′ UTR lengthening in highly metastatic cells—undergo AGO2-dependent translational repression. Furthermore, HNRNPC targets inherently contain a poly(U) stretch in their 3′ UTR, which might favour interaction with other poly(U) binding RBPs and lead to translational repression.

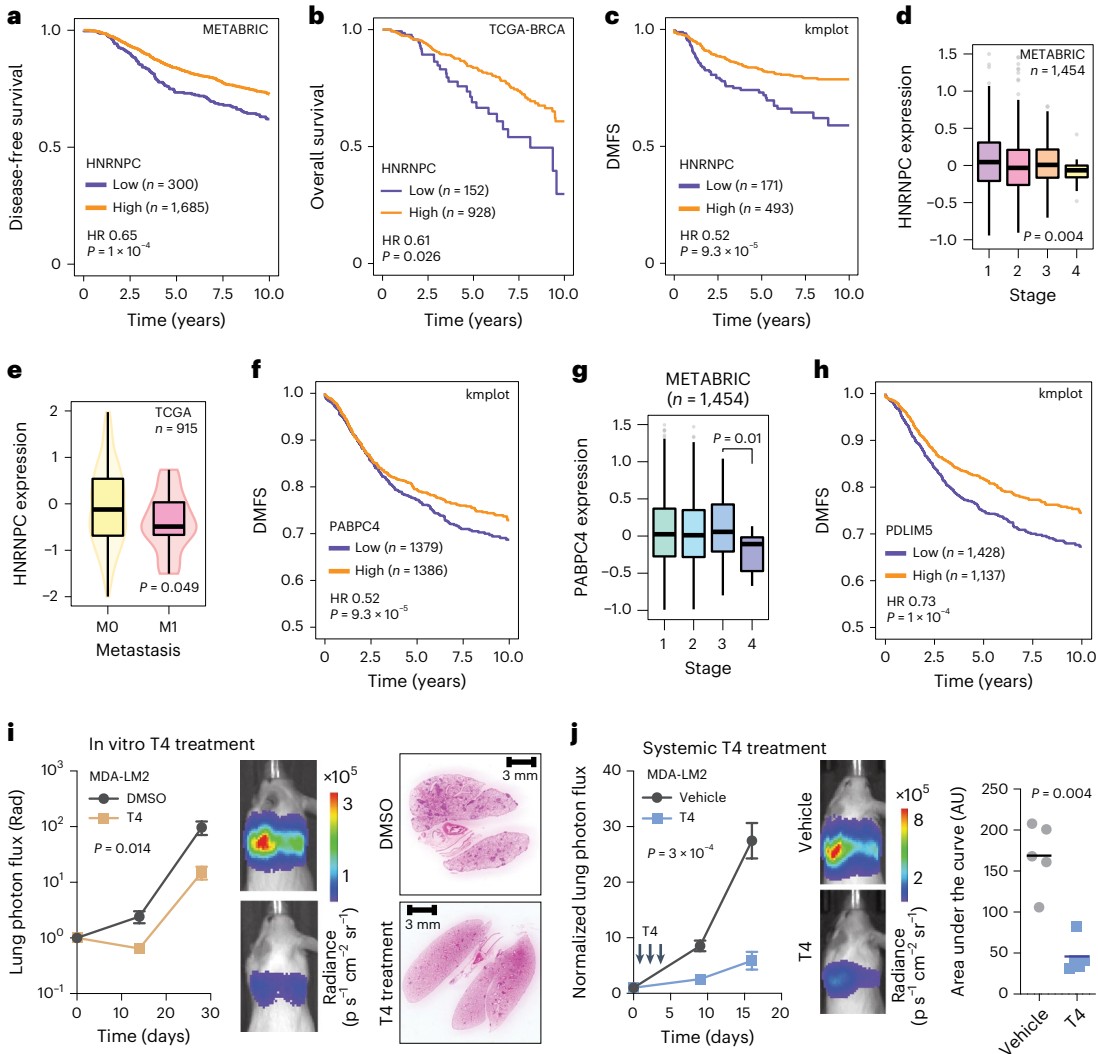

**Fig. 6 | HNRNPC expression is associated with clinical outcomes in patients with breast cancer. a**, Kaplan–Meier survival curve showing association between tumour HNRNPC levels and disease-free survival in the METABRIC cohort. **b**, Kaplan–Meier survival curve showing association between tumour HNRNPC levels and overall survival in the TCGA-BRCA cohort. **c**, Kaplan–Meier survival curve showing association between tumour HNRNPC levels and distant metastasis-free survival (DMFS) in a collection of breast cancer patient cohorts. Hazard ratios (HR) and $P$ values (calculated using log-rank test) are shown (**a**–**c**). **d**, HNRNPC mRNA levels across breast cancer tissue stages I–IV in the METABRIC cohort. $P$ value calculated using one-way analysis of variance (ANOVA). Box centre reports the median value, the boundaries—the quartiles and the whiskers—the 10th and 90th percentiles. **e**, HNRNPC mRNA levels in non-metastatic (M0) and metastatic (M1) breast tumours in the TCGA-BRCA cohort. $P$ value calculated using two-tailed Mann–Whitney $U$ test. Box plot characteristics as in **d**. **f**, Kaplan–Meier survival curve showing association between tumour PABPC4 levels and distant metastasis-free survival (DMFS) in a collection of breast cancer patient cohorts. Hazard ratios (HR) and $P$ values

(calculated using log-rank test) are shown. **g**, PABPC4 mRNA levels across breast cancer tissue stages I–IV in the METABRIC cohort. $P$ value calculated using one-way ANOVA. Box plot characteristics as in **d**. **h**, Kaplan–Meier survival curve showing association between tumour PDLIM5 levels and distant metastasis-free survival (DMFS) in a collection of breast cancer patient cohorts. Hazard ratios (HR) and $P$ values (calculated using log-rank test) are shown. **i**, MDA-LM2 cells treated with T4 or vehicle control (DMSO) at 3 µM for 6 h were injected via tail vein into NSG mice. Bioluminescence was measured at the indicated times (mean values are shown, error bars indicating standard error of the mean (s.e.m.); $P$ value calculated using two-way ANOVA). Lung sections were stained with H&E (representative images shown). $n = 4$–5 mice per cohort. **j**, NSG mice were intravenously injected with MDA-LM2, and intraperitoneally injected with 10 mg kg$^{-1}$ T4 or vehicle control for three consecutive days, starting on the day of cancer cell injection. Bioluminescence was measured at the indicated times (mean values are shown, error bars indicating s.e.m.; $P$ value calculated using two-way ANOVA); area under the curve was measured at the final timepoint ($P$ value calculated using one-tailed Mann–Whitney $U$ test). $n = 4$–5 mice per cohort.

We identified PDLIM5 as a downstream target of HNRNPC and showed that PDLIM5 KD phenocopied the pro-metastatic pheno-type of HNRNPC-depleted cells. PDLIM5 (PDZ and LIM domain 5) is a member of cytoskeleton-associated protein family, implicated in cell–cell, cell–extracellular matrix interactions and cell migra-tion[34]. It also participates in the mechanosensing cascade via YAP/TAZ signalling[40]. PDLIM5 is phosphorylated by AMPK, and this mod-ulates its function in cell migration[41]. Interestingly, these cellular

and molecular functions were enriched among the downregulated proteins in HNRNPC-depleted cells.

We have uncovered an intricate gene regulatory programme at the intersection of APA and translational control mediated by HNRNPC and PABPC4 that plays a metastasis suppressing role in breast cancer. Our clinical association analyses suggest that HNRNPC expression, along with that of its regulon, could be used as a prog-nostic metric for disease progression. We also provide evidence that

HNRNPC-low tumours could benefit from therapeutic strategies targeting APA.

## Online content

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

## Methods

This study complies with all relevant ethical regulations that are approved by the University of California, San Francisco (UCSF) Institutional Review Board and Institutional Animal Care and Use Committee (IACUC, approval number AN179718).

### Cell culture

All cells were cultured in a 37 °C 5% $CO_2$ humidified incubator. The MDA-MB-231 (ATCC HTB-26) breast cancer cell line, its highly metastatic derivative MDA-LM2 (ref. 15), and 293T cells (ATCC CRL-3216) were cultured in Dulbecco's modified Eagle medium high-glucose medium supplemented with 10% foetal bovine serum (FBS), glucose (4.5 g l$^{-1}$), L-glutamine (4 mM), sodium pyruvate (1 mM), penicillin (100 units ml$^{-1}$), streptomycin (100 µg ml$^{-1}$) and amphotericin B (1 µg ml$^{-1}$) (Gibco). The HCC1806-LM2 cell line (an in vivo selected highly lung metastatic derivative of the HCC1806 breast cancer line (ATCC CRL-2335)) was cultured in RPMI-1640 medium supplemented with 10% FBS, glucose (2 g l$^{-1}$), L-glutamine (2 mM), 25 mM HEPES, penicillin (100 units ml$^{-1}$), streptomycin (100 µg ml$^{-1}$) and amphotericin B (1 µg ml$^{-1}$) (Gibco). All cell lines were routinely screened for mycoplasma with a PCR-based assay and tested negative.

### Cell proliferation assays

At day zero, $10^3$ cells per well were seeded in 96-well plates, in six biological replicates. One, three and five days later the cell proliferation was assessed using CellTiter-Glo Luminescent Cell Viability Assay (Promega). An exponential model was then used to fit a growth rate for each sample ($\ln(N_{t-1}/N_1) = rt$ where $t$ is measured in days), and an unpaired two-sided $t$-test was used to test for significant variations.

### Colony formation assays

Five-hundred cells were seeded in six-well plates, in biological triplicates. After the colonies reached a desired size (1–2 weeks later), the cells were paraformaldehyde (PFA)-fixed, stained with aqueous 0.1% (w/v) crystal violet solution and destained with several washes in de-ionized water. The colonies were manually counted, and an unpaired two-sided $t$-test was used to test for significant variations.

### Cell migration and Matrigel invasion assays

The cells were serum starved in the medium containing 0.2% FBS for 18 h, then detached and counted. A total of $4 × 10^5$ cells were seeded onto transwell inserts (containing 8-µm-pore-size polyester membrane, Sterlitech 9328012) in four biological replicates, in 500 µl starving medium. For invasion assays, the transwells were pre-coated with 80 µl 0.15 mg ml$^{-1}$ Matrigel (Corning) in starving medium, for 2 h at 37 °C. For migration assays, the transwell inserts were left untreated. The transwells were put in 24-well plates, and complete medium (containing 10% FBS) was added, 500 µl per well. Twenty-four hours later the cells from the inside of transwells were removed by scraping, and the transwells and wells were washed and PFA-fixed. The cells that had migrated through the membrane were then permeabilized in 0.1% Triton X solution in 1× phosphate-buffered saline (PBS) and DAPI-stained. The membranes were cut, mounted on glass slides and imaged on an inverted fluorescence microscope in UCSF Nikon Imaging Center. The invaded cells were manually counted, and two-tailed Mann–Whitney $U$ test was used to test for significant variations.

### T4 treatment

T4 (Enamine EN300-7536403 or Sigma SML2299) was dissolved in dimethyl sulfoxide (DMSO) at 10 mM stock concentration. For 3′-end RNA-seq, cells were treated with 5 µM T4 or an equivalent amount of DMSO for 6 h, before RNA extraction. For dose–response measurements, the cells were treated with T4 at indicated concentrations for 6 h, after which the medium was changed and the cell viability was measured 72 h later using CellTiter-Glo Luminescent Cell Viability Assay (Promega) in six biological replicates. For in vivo metastasis assays, MDA-LM2 cells were treated with an IC20 concentration of T4 (3 µM) or DMSO control for 6 h, and then immediately collected for tail vein injections. For time course experiments, MDA-LM2 cells were treated with 3 µM T4 or DMSO control for 6 h, then washed and incubated in drug-free medium until RNA extraction 24 h or 72 h later.

### Gene KD and OE

MDA-MB-231, MDA-LM2 and HCC1806-LM2 cells expressing dCas9-KRAB fusion protein were constructed by lentiviral delivery of pMH0006 (Addgene #135448) and fluorescence-activated cell sorting (FACS) isolation of BFP-positive cells.

The lentiviral constructs were co-transfected with pCMV-dR8.91 and pMD2.D plasmids using TransIT-Lenti (Mirus) into 293T cells, following the manufacturer's protocol. Virus was collected 48 h post-transfection and passed through a 0.45 µm filter. Target cells were then transduced overnight with the filtered virus in the presence of 8 µg ml$^{-1}$ polybrene (Millipore).

Guide RNA sequences for CRISPRi-mediated gene KD were cloned into pCRISPRia-v2 (Addgene #84832) via BstXI-BlpI sites (for single guide RNA sequences, see Supplementary Table 2). For double KD experiments, pCRISPRia-v2 plasmid was modified to construct pCRISPRia-v2-Blast, replacing puromycin acetyltransferase by blasticidin deaminase CDSs. After transduction with single guide RNA lentivirus, MDA-MB-231, MDA-LM2 and HCC1806-LM2 CRISPRi cells were selected with 1.5 µg ml$^{-1}$ puromycin or 20 µg ml$^{-1}$ blasticidin (Gibco). KD of target genes was assessed by RT−qPCR as described below. We consistently observed two- to five-fold decrease in target mRNA expression, with the exception of PABPN1. PABPN1 is labelled as common essential gene in DepMap Portal database, and we observed only modest PABPN1 KD efficiency in MDA-MB-231 cells, accompanied by severe impact on cell proliferation.

The EF1α promoter-driven lentiviral reporter has been described[42]. PDLIM5, HNRNPC or mCherry CDSs were synthesized (Integrated DNA Technologies (IDT)) and cloned by restriction digestion (PacI) and Gibson assembly. The constructs were delivered to MDA-MB-231 or MDA-LM2 cells by lentiviral transduction and puromycin selection. The expression of transgenes was evaluated by RT−qPCR as described below. We consistently observed seven- to ten-fold increase in target mRNA expression.

### RNA isolation

Total RNA for RNA-seq and RT−qPCR was isolated using the Zymo QuickRNA isolation kit with in-column DNase treatment per the manufacturer's protocol.

### RT−qPCR

Transcript levels were measured using RT−qPCR by reverse transcribing total RNA to complementary DNA (Maxima H Minus RT, Thermo), then using PerfeCTa SYBR Green SuperMix (QuantaBio) per the manufacturer's instructions. HPRT1 was used as endogenous control (for primer sequences, see Supplementary Table 2).

### Western blotting

Cell lysates were prepared by lysing cells in ice-cold RIPA buffer (25 mM Tris−HCl pH 7.6, 0.15 M NaCl, 1% IGEPAL CA-630, 1% sodium deoxycholate and 0.1% SDS) containing 1× protease inhibitors (Thermo Scientific). Lysate was cleared by centrifugation at 20,000g for 10 min at 4 °C. Samples were denatured for 10 min at 70 °C in 1× LDS loading buffer (Invitrogen) and 50 mM dithiothreitol (DTT). Proteins were separated by SDS−PAGE using 4–12% Bis-Tris NuPAGE gels, transferred to nitrocellulose (Millipore), blocked using 5% bovine serum albumin and probed using target-specific antibodies. Bound antibodies were detected using horseradish peroxidase (HRP)-conjugated secondary antibodies (anti-mouse IgG, light chain specific, HRP (Cell Signaling 91196S,

1/5,000) or anti-rabbit IgG, conformation specific, HRP (Cell Signaling 5127S, 1/5,000)) and ECL substrate (Pierce) or infra-red dye-conjugated secondary antibodies (anti-mouse IgG, IRDye 800CW (Li-Cor 926-32210, 1/10,000) or anti-rabbit IgG, IRDye 680RD (Li-Cor 926-68071, 1/10,000)) according to the manufacturer's instructions. Primary antibodies: β-tubulin (Proteintech 66240-1-Ig, 1/10,000), HNRNPC (Santa Cruz sc-32308, 1/1,000), PABPC4 (Proteintech 14960-1-AP, 1/1,000), PABPN1 (Proteintech 66807-1-Ig, 1/1,000), PDLIM5 (Proteintech 10530-1-AP, 1/1,000), TXNRD1 (Proteintech 11117-1-AP, 1/3,000), PGK1 (Proteintech 17811-1-AP, 1/5,000), GAPDH (Proteintech 60004-1-Ig, 1/10,000) and FLAG tag (Proteintech 66008-1-Ig, 1/1,000).

### Subcellular fractionation

Cells were lysed in ice-cold cytoplasmic lysis buffer (20 mM Tris pH 7.5, 0.15 M NaCl, 5 mM MgCl$_2$, 1 mM DTT, 0.15% Igepal CA-630 and RiboLock 40 U ml$^{-1}$) containing 1× protease inhibitors (Thermo Scientific) for 5 min on ice. The lysates were then centrifuged on a sucrose cushion (20 mM Tris pH 7.5, 0.15 M NaCl and 25% sucrose) at 16,000$g$ for 10 min at 4 °C. Top layer lysate was taken as cytoplasmic fraction, and used for RNA extraction with Zymo QuickRNA isolation kit with in-column DNase treatment per the manufacturer's protocol. The nuclear fraction (pellet) was washed in Nuclei Wash buffer (1× PBS, 1 mM ethylenediaminetetraacetic acid (EDTA) and 0.1% Triton) and spun down for 1 min at 1,150$g$ at 4 °C. The pellet was solubilized in ice-cold RIPA buffer with RNase and protease inhibitors, and the nuclear lysates were used for RNA extraction as described above.

### Reporter assays

The bidirectional CMV promoter-driven lentiviral reporter, expressing eGFP and PuroR-T2A-mCherry fusion has been described[43]. The eGFP open reading frame was replaced with FLAG-BFP CDS by restriction digestion (EcoRV-NheI) and Gibson cloning. A region encompassing two alternative poly(A) sites of KAT14 gene (hg38 chr20:18,187,963-18,188,401) was cloned downstream of the BFP via MluI-PacI sites. KAT14 was chosen as it satisfied two criteria: (1) the distance between the alternative poly(A) sites allowing an amplicon compatible with Illumina sequencing; (2) the usage of both poly(A) sites in MDA-MB-231 cells detectable by RT–PCR. A library of sequences (150 bp long), containing CLIP-seq-derived HNRNPC binding sites in 3′ UTRs or nucleotide-content-matched shuffled sequences (for sequences, see Supplementary Table 3), was cloned upstream of the proximal poly(A) site via MluI digestion and Gibson assembly. The reporter library was delivered to MDA-MB-231 cells by lentiviral transduction and puromycin selection. The resulting cells were then transfected with control or HNRNPC-targeting small interfering RNAs (IDT) using Lipofectamine 2000 (Thermo Scientific). Seventy-two hours post-transfection the cells were analysed by flow cytometry, determining BFP/mCherry ratio. Total RNA was extracted from a portion of transfected cells, and the remaining cells were FACS sorted into two bins, representing the top and bottom 25% of the BFP/mCherry ratio. The RNA extracted from the total and sorted cells was used to amplify the reporter RNA by RT–PCR. In brief, an anchored oligo dT primer containing a unique molecular identifier (UMI) was used for reverse transcription, and a BFP-specific primer for PCR (for primer sequences, see Supplementary Table 2). The amplicon libraries were sequenced on a paired-end Illumina sequencing run, allowing matching of the poly(A) site chosen (proximal or distal) with the library insert upstream of the proximal poly(A) site. The reads matching to each library element and poly(A) site used were compared with APAlog.

The EF1α promoter-driven lentiviral reporter has been described[42]. The FLAG-PDLIM5 CDS (synthesized by IDT) and PCR-amplified PDLIM5 3′ UTR fragments (for primer sequences, see Supplementary Table 2) were cloned by restriction digestion (PacI) and Gibson assembly. The reporters were delivered to MDA-MB-231 cells by lentiviral transduction and puromycin selection. The reporter protein expression was determined by western blotting.

### Ribosome profiling

Ribosome profiling was performed as previously described[44] with following modifications. Snap-frozen PDX tumours were cryoground into powder on dry ice, and then resuspended in ice-cold lysis buffer. The RNA concentration in the lysates was determined with the Qubit RNA HS kit (Thermo).

Monosomes were isolated using MicroSpin S-400 HR (Cytiva) columns, pre-equilibrated with 3 ml polysome buffer per column. One-hundred microlitres of digested lysate was loaded per column (two columns were used per 200 μl sample) and centrifuged 2 min at 600$g$. The RNA from the flow-through was isolated using the Zymo RNA Clean and Concentrator-25 kit. In parallel, total RNA from undigested lysates were isolated using the same kit.

Libraries were sequenced on a SE50 run on Illumina HiSeq4000 instrument at UCSF Center for Advanced Technologies.

To process the reads, the Ribo-seq reads were first trimmed using cutadapt (v2.3) to remove the linker sequence AGATCGGAAGAGCAC. The fastx_barcode_splitter script from the Fastx toolkit (v0.0.13) was then used to split the samples based on their barcodes. Since the reads contain UMIs, they were collapsed to retain only unique reads. The UMIs were then removed from the beginning and end of each read (two and five nucleotides, respectively) and appended to the name of each read. Bowtie2 (v2.3.5) was then used to remove reads that map to ribosomal RNAs and tRNAs, and the remainder of reads were then aligned to mRNAs (we used the isoform with the longest CDS for each gene as the representative). Subsequent to alignment, umitools (v0.3.3) was used to deduplicate reads.

### RNA-seq and 3′-end RNA-seq

RNA-seq libraries (used for calculating TEs) were prepared using SMARTer Stranded Total RNA-Seq Kit v2 Pico Input Mammalian (Takara), with 50 ng total RNA as input. The 3′-end RNA-seq libraries (used for determining poly(A) site usage) were prepared using Quant-Seq 3′ mRNA-Seq Library Prep Kit REV (Lexogen), with 500 ng total RNA as input. Libraries were sequenced as SE50 runs on Illumina HiSeq4000 instrument at UCSF Center for Advanced Technologies.

To compare changes in 3′ UTR usage and poly(A) site selection, we first annotated unique 3′ ends of transcripts using Gencode annotations (v33). Salmon (v0.14.1) was then used to count the number of reads that match each of the annotated ends. The normalized abundances were then tabulated, and APAlog (see below) was used to perform pairwise comparisons between proximal and distal poly(A) sites between conditions.

To assess whether HNRNPC binding was associated with the observed changes in logAPAR values, proximal sites within 500 nt of annotated HNRNPC peaks (based on CLIP-seq datasets) were annotated. The behaviour of these HNRNPC-associated proximal sites was then compared with the background using Wilcoxon rank sum test. Alternatively, logAPAR values were binned into equally populated bins, and the enrichment/depletion patterns of HNRNPC-associated proximal sites were assessed as previously described[23].

### HNRNPC CLIP-seq

CLIP-seq for endogenous HNRNPC in MDA-MB-231 cells was performed using irCLIP[45], with the following modifications. The cells were crosslinked with 400 mJ cm$^{-2}$ 254 nm UV. Cells were lysed in CLIP lysis buffer (1× PBS, 0.1% SDS, 0.5% sodium deoxycholate and 0.5% IGEPAL CA-630) supplemented with 1× protease inhibitors (Thermo) and SUPERaseIN (Thermo), then treated with DNase I (Promega) for 5 min at 37 °C. Lysate was clarified by spinning at 21,000$g$ at 4 °C for 15 min. RNA–protein complexes were then immunoprecipitated from the clarified lysate using protein G Dynabeads (Thermo) conjugated to

anti-HNRNPC (Santa Cruz sc-32308) for 2 h at 4 °C. Beads were washed sequentially with high-stringency buffer, high-salt buffer and low-salt buffer. RNA–protein complexes were then nuclease treated on-bead with RNase A (Thermo), and then ligated to the irCLIP adaptor using T4 RNA ligase (NEB) overnight at 16 °C. RNA–protein complexes were then eluted from beads, resolved on a 4–12% Bis-Tris NuPAGE gel, transferred to nitrocellulose, then imaged using an Odyssey Fc instrument (Licor). Regions of interest were excised from the membrane, and the RNA was isolated by Proteinase K digestion followed by pulldown with oligo d(T) magnetic beads (Thermo). The resulting RNA was then reverse transcribed using Superscript IV RT (Invitrogen) and a barcoded RT primer, purified using MyOne C1 Dynabeads (Invitrogen) and then circularized using CircLigase II (Epicentre). Two rounds of PCR were then performed to first amplify the library using adaptor-specific primers and to add sequences compatible with Illumina sequencing instruments. The libraries were then sequenced as SE50 runs on Illumina HiSeq4000 instrument at UCSF Center for Advanced Technologies.

The CTK package (v1.1.13, CLIP toolkit[46]) was used to annotate peaks from CLIP-seq data. Reads were first collapsed using their UMIs. UMI-tools package was then used to extract the UMI followed by quality trimming (-q 15) and linker removal using cutadapt. BWA (v0.7.17) was then used to align reads to the genome (hg38). CTK scripts were then used to remove PCR duplicates, parse alignments and call peaks using a valley-seeking algorithm (with multi-testing correction). The boundaries of the resulting peaks were combined across multiple independent CLIP experiments, and their union with a previously published HNRNPC iCLIP (E-MTAB-1371) was used to define a comprehensive HNRNPC binding across the transcriptome. To identify motifs, sequences across annotated peak were extracted; control sequences were generated by scrambling the real sequences while maintaining dinucleotide sequence content. FIRE[17] was then used to find enriched sequence motifs.

## PABPC4 PAPERCLIP and TIA1, ELAVL1 CLIP-seq

The MDA-MB-231 cells were crosslinked with 400 mJ cm$^{-2}$ 254 nm UV. Cells were lysed in CLIP lysis buffer (1× PBS, 0.1% SDS, 0.5% sodium deoxycholate and 0.5% IGEPAL CA-630) supplemented with 1× protease inhibitors (Thermo) and SUPERaseIN (Thermo), then treated with DNase I (Promega) for 5 min at 37 °C. Lysates were then split in half and separately treated with medium and low dilutions of RNaseA and RNaseI (Thermo; 1/3,000 RNaseA and 1/100 RNaseI, and 1/15,000 RNaseA and 1/500 RNaseI, respectively). Lysates were then clarified by spinning at 21,000 g at 4 °C for 15 min. Clarified lysates were pooled, and RNA–protein complexes were then immunoprecipitated using protein A/G beads (Pierce) conjugated to anti-PABPC4 (Proteintech 14960-1-AP), or anti-TIA1 (Proteintech 12133-2-AP), or anti-ELAVL1 (Proteintech 11910-1-AP) for 2 h at 4 °C. Beads were then washed sequentially with low-salt buffer, high-salt buffer and PNK buffer. Protein-bound RNAs were end-repaired on beads using T4 PNK (NEB) and 3′-end labelled with azide-dUTP using yeast poly(A) polymerase (Jena). The protein–RNA complexes were labelled with IRDye800-DBCO conjugates (LiCor). The protein–RNA complexes were then eluted from beads, resolved on a 4–12% Bis-Tris NuPAGE gel, transferred to nitrocellulose and imaged using an Odyssey Fc instrument (LiCor). Regions of interest were excised from the membrane, and the RNA was isolated by Proteinase K digestion and phenol/chloroform extraction. Eluted RNA was used for library preparation using SMARTer smRNA-Seq Kit (Takara), with following modifications. The poly(A) tailing step was omitted, and reverse transcription was performed with a custom RT primer (Supplementary Table 2). The library PCR was performed with index forward (i5) primers and universal reverse (P7) primer (Supplementary Table 2). The libraries were purified using Zymo Select-a-Size beads and sequenced as a SE50 run on Illumina HiSeq4000 instrument at UCSF Center for Advanced Technologies.

## TMT–MS

The cell lysates were prepared, digested and labelled using TMT10plex Isobaric Mass Tagging Kit (Thermo), as per the manufacturer's instructions. The labelling reactions were cleaned up and fractionated using Pierce High pH Reversed-Phase Peptide Fractionation Kit (Thermo).

Peptides were analysed on a Thermo Fisher Orbitrap Fusion Lumos Tribid MS system equipped with an Easy nLC 1200 ultrahigh-pressure liquid chromatography system interfaced via a Nanospray Flex nanoelectrospray source. Samples were injected on a C18 reverse phase column (25 cm × 75 μm packed with ReprosilPur C18 AQ 1.9 μm particles). Peptides were separated by a gradient from 5% to 32% acetonitrile in 0.02% heptafluorobutyric acid over 120 min at a flow rate of 300 nl min$^{-1}$. Spectra were continuously acquired in a data-dependent manner throughout the gradient, acquiring a full scan in the Orbitrap (at 120,000 resolution with an AGC target of 400,000 and a maximum injection time of 50 ms) followed by ten MS/MS scans on the most abundant ions in 3 s in the dual linear ion trap (turbo scan type with an intensity threshold of 5,000, CID collision energy of 35%, AGC target of 10,000, maximum injection time of 30 ms and isolation width of 0.7 $m/z$). Singly and unassigned charge states were rejected. Dynamic exclusion was enabled with a repeat count of 1, an exclusion duration of 20 s and an exclusion mass width of ±10 p.p.m. Data were collected using the MS3 method[47] for obtaining TMT tag ratios with MS3 scans collected in the orbitrap at a resolution of 60,000, HCD collision energy of 65% and a scan range of 100–500.

Protein identification and quantification were done with Integrated Proteomics Pipeline (IP2, Integrated Proteomics Applications) using ProLuCID/Sequest, DTASelect2 and Census[48,49]. Tandem mass spectra were extracted into ms1, ms2 and ms3 files from raw files using RawExtractor[50] and were searched against the UniProt human protein database plus sequences of common contaminants, concatenated to a decoy database in which the sequence for each entry in the original database was reversed[51]. Search space included all fully tryptic peptide candidates with no missed cleavage restrictions. Carbamidomethylation (+57.02146) of cysteine was considered a static modification; TMT tag masses, as given in the TMT kit product sheet, were also considered static modifications. We required one peptide per protein and both tryptic termini for each peptide identification. The ProLuCID search results were assembled and filtered using the DTASelect program with a peptide false discovery rate (FDR) of 0.001 for single peptides and a peptide FDR of 0.005 for additional peptides for the same protein. Under such filtering conditions, the estimated FDR was between zero and 0.06 for the datasets used. Quantitative analysis on MS3-based MultiNotch TMT data was analysed with Census 2 in IP2 platform[47,52,53]. As TMT reagents are not 100% pure, we referred to the Thermo Fisher Scientific TMT product data sheet to obtain purity values for each tag and normalized reporter ion intensities. While identification reports best hit for each peptide, Census extracted all PSMs that can be harnessed to increase accuracy from reporter ion intensity variance. Extracted reporter ions were further normalized by using total intensity in each channel to correct sample amount error.

## Co-IP–MS

MDA-MB-231 and MDA-LM2 cells (10 × 10$^6$ per replicate) were washed with ice-cold 1× PBS and lysed in nuclei lysis buffer (100 mM Tris–HCl pH 7.5, 0.5% SDS and 1 mM EDTA) containing 1× protease inhibitors (Thermo Scientific) on ice for 10 min. The lysates were then diluted with four volumes of IP dilution buffer (62.5 mM Tris–HCl pH 7.5, 187.5 mM NaCl, 0.625% Triton X-100 and 1 mM EDTA) with protease inhibitors and passed through a 25 G needle several times. The lysates were cleared 10 min at 21,000 g at +4 °C and used for IP.

For co-IP–MS analysis, HNRNPC antibody was covalently bound to the magnetic beads. For this, HNRNPC antibody (Santa Cruz sc-32308) or mouse IgG (Jackson 015-000-003) was first purified using Protein A/G beads (Thermo). Briefly, 3 μg of antibody were bound to 15 μl

Protein A/G beads (per IP replicate) in Modified Coupling buffer (20 mM sodium phosphate pH 7.2, 315 mM NaCl, 0.1 mM EDTA, 0.1% IGEPAL CA-630 and 0.5% glycerol) and incubated 15 min at room temperature. Then the beads were washed twice in modified coupling buffer and once in coupling buffer (20 mM sodium phosphate pH 7.2 and 300 mM NaCl), and the antibody was eluted in 0.1 M sodium citrate buffer (pH 2.5) for 5 min at room temperature. After neutralization with 1/10 volume of 1 M sodium phosphate buffer (pH 8), the antibody was coupled to M270 Epoxy Dynabeads (Thermo Scientific) in ammonium sulfate buffer (0.1 M sodium phosphate pH 7.4 and 1.2 M ammonium sulfate, final concentration) overnight at 37 °C. Before usage, the antibody conjugated beads were washed four times in 1× PBS, once in 1× PBS supplemented with 0.5% Tween-20 and resuspended in 1× PBS.

Protein complexes were immunoprecipitated with antibody-conjugated beads for 2 h at 4 °C, washed three times in wash buffer (15 mM Tris–HCl pH 7.5, 150 mM NaCl and 0.1% Triton X-100) and eluted in 1× NuPage LDS sample buffer with 0.1 M DTT for 10 min at 70 °C. Eluates were then subjected to alkylation, detergent removal, and trypsin digestion using Filter Aided Sample Preparation protocol[54], followed by desalting using StageTips[55]. Desalted peptides were subsequently lyophilized by vacuum centrifugation, resuspended in 7 µl of A* buffer (2% acetonitrile, 0.5% acetic acid and 0.1% trifluoroacetic acid in water), and analysed on a Q-Exactive plus Orbitrap mass spectrometer coupled with a nanoflow ultimate 3000 RSL nano HPLC platform (Thermo Fisher), as described before[56]. Briefly, 6 µl of each peptide sample was resolved at 250 nl min$^{-1}$ flow rate on an Easy-Spray 50 cm × 75 µm RSLC C18 column (Thermo Fisher), using a 123 min gradient of 3% to 35% of buffer B (0.1% formic acid in acetonitrile) against buffer A (0.1% formic acid in water), followed by online infusion into the mass spectrometer by electrospray (1.95 kV, 255C). The mass spectrometer was operated in data-dependent positive mode. A TOP15 method in which each MS scan is followed by 15 MS/MS scans was applied. The scans were acquired at 375–1,500 $m/z$ range, with a resolution of 70,000 (MS) and 17,500 (MS/MS). A 30 s dynamic exclusion was applied. MaxQuant (v1.6.3.3) was used for all MS search and protein quantifications. All downstream MS data analysis was performed using Perseus (v1.6.2.3).

For co-IP/western blot analysis, when indicated, the lysates were pre-treated with RNaseA (10 µg RNaseA per 1 mg lysate, 10 min on ice) and incubated with HNRNPC antibody or mouse IgG overnight at +4 °C. The protein complexes were then immunoprecipitated with Protein A/G beads for 2 h at +4 °C, washed three times with wash buffer (15 mM Tris–HCl pH 7.5, 150 mM NaCl and 0.1% Triton X-100) and eluted in 1× NuPage LDS sample buffer with 0.1 M DTT for 10 min at 70 °C.

**Metastatic colonization assay**

Mice were housed in accordance with UCSF IACUC protocol (approval number AN179718) in humidity- and temperature-controlled rooms on a 12 h light–dark cycle with free access to food and water. Seven- to 12-week-old age-matched female NSG mice (Jackson Labs, 005557) were used for lung colonization assays. For this assay, cancer cells constitutively expressing luciferase were suspended in 100 µl PBS and then injected via tail vein (2.5 × 10$^4$ for MDA-LM2, 5 × 10$^4$ for MDA-MB-231 and 1 × 10$^5$ for HCC1806-LM2). Each cohort contained four to five mice, which in NSG background is enough to observe a more than two-fold difference with 90% confidence. Mice were randomly assigned into cohorts. Cancer cell growth was monitored in vivo at the indicated times by retro-orbital injection of 100 µl of 15 mg ml$^{-1}$ luciferin (PerkinElmer) dissolved in 1× PBS, and then measuring the resulting bioluminescence with an IVIS instrument and Living Image software (PerkinElmer). Mice were killed before the normalized photon flux in the lung region reached 5 × 10$^8$; this limit was not exceeded. For systemic T4 treatment, the animals were intraperitoneally injected with the drug (10 mg kg$^{-1}$) or a vehicle control (95% corn oil and 5% DMSO) for three consecutive days[57], starting on the day of cancer cell intravenous injection. For spontaneous metastasis assays, 1 × 10$^5$ MDA-MB-231 cells

mixed with 25 µl of Matrigel (Corning) were injected into mammary fat pads of NSG mice. Tumour volume was assessed by caliper measurements and resected 4 weeks after cancer cell injection. Tumours were resected before the tumour volume reached 1,000 mm$^3$ (or 1.5 cm in diameter); this limit was not exceeded. Lung colonization was detected using in vivo bioluminescence as described above.

**Histology**

For gross macroscopic metastatic nodule visualization, mouse lungs (from each cohort) were extracted at the endpoint of each experiment, and 5-µm-thick lung tissue sections were haematoxylin and eosin (H&E) stained. The number of macroscopic nodules was then recorded for each section. An unpaired $t$-test was used to test for significant variations.

**PDXs**

Primary tumours of established triple-negative breast cancer PDX models (HCI-001, HCI-002, HCI-010 and STG139) were generated in NSG mice as described before[18,19]. Original human tissues for generating PDX models were received as de-identified samples, and all subjects provided written informed consent. Medical reports were obtained without personally identifiable information. The UCSF IACUC reviewed and approved all animal experiments. The metastatic potential of the PDXs was determined when primary tumours reached 2.5 cm in diameter[20]. For histological analysis, the middle and postcaval lobes of the right lung were fixed in 4% PFA overnight and processed for paraffin embedding. Lung sections were stained with H&E using standard protocols and manually analysed for the presence of metastasis using a Leica DMR microscope. For ribosome profiling, the tumours were collected at the size of 1.0 cm diameter and snap frozen immediately. Tumours were stored at −80 °C until further processed as described above.

**Computational tools**

**Ribolog.** Unlike differential gene expression analysis using RNA-seq data, which involves comparing two or more count numbers, modelling changes in TE requires comparing ratios between conditions (TE corresponds to a ratio between ribosome protected mRNA footprint and mRNA abundance counts). The main outcome of interest in ribosome profiling, TER, is the ratio of these two TEs. Since the introduction of ribosome profiling, several analytical packages have been developed that largely inherit the assumptions of prior methods originally designed for RNA-seq data analysis[58]. A closer evaluation of the underlying assumptions used in many of these tools, for example, negative binomial distribution of read counts, revealed that reliable estimation of parameters such as overdispersion required many more biological replicates than are commonly generated in studies of translational control[59,60]. Moreover, the ratio of two NB variables does not follow any known statistical distribution; therefore, inference on TER using parametric significance tests remains a challenge. We thus sought to devise a new analytical framework for reliable comparison of TEs across conditions with fewer a priori assumptions. The resulting method, which we have named Ribolog, relies on logistic regression to model individual Ribo-seq and RNA-seq reads to estimate logTER (that is, log fold change in TE) and its associated $P$ value across the coding transcriptome.

Before entering the Ribolog pipeline, RNA and RPF fastq files are pre-processed, as described above. Sorted and indexed bam files are imported into Ribolog, and mapped reads are assigned to specific codons using functions borrowed from the R package riboWaltz[61].

**Stalling bias detection and correction.** Suboptimal codons, RNA secondary structures and activation of RBP binding sites may stall translation at certain codons and produce peaks of RPF reads that stand out against the CDS background. If not removed, stalling reads

will be counted in with other RPF reads and lead to an overestimation of TE. This may lead to false inferences because stalling reads signify locally obstructed translation and should not be misconstrued as a sign of overall increase in translation rate. We developed a new metric, the Consistent Excess of Loess Predictions (CELP) bias coefficient to measure the strength of stalling bias at each codon. First, we smooth out the observed codon counts along the transcript for each sample using the loess function to produce loess predicted counts. Then, we calculate the CELP bias coefficient and the bias-corrected read count as:

$$b_{ij} = \left( \prod_K \frac{y^l_{ijk}}{M^l_{jk}} \right)^{\frac{1}{k}}$$

$$y^{l,c}_{ijk} = \frac{y^l_{ijk}}{b_{ij}}$$

$$y^{l,C}_{jk} = \sum_i y^{l,c}_{ijk}$$

where:

$b_{ij}$: CELP stalling bias coefficient at codon $i$, gene $j$

$y^l_{ijk}$: loess predicted (smoothed) read count at codon $i$, gene $j$, sample $k$

$M^l_{jk}$: median of non-zero loess-smoothed counts in gene $j$, sample $k$

$y^{l,c}_{ijk}$: bias-corrected read count at codon $i$, gene $j$, sample $k$

$y^{l,C}_{jk}$: bias-corrected RPF read count of gene $j$, sample $k$.

The reasons for using loess-smoothed counts and not raw counts in the above calculations are three-fold: (1) In our experience with multiple ribosome profiling datasets, we have observed that stalling peaks often appear in the same approximate position, but not necessarily the same exact codon, even among replicates of a single biological sample. (2) Some of the factors that impede translation, for example, RBP binding or RNA secondary structures, affect several adjacent codons, not a single codon. (3) Calculation of P-site offset and assignment of RPF reads to specific codons carries a degree of uncertainty, because the distance of read ends from start or stop codon, which is used to estimate P-site offset, is always a distribution, not a single value, even for reads of the same length. It is therefore beneficial to borrow information from neighbouring codons for detection of stalling events. The radius of this neighbourhood—which determines the loess 'span' parameter—can be changed by the user (default: 5). Median of loess-smoothed non-zero counts ($M^l_{jk}$) represents background CDS translation level, and the ratio $\frac{y^l_{ijk}}{M^l_{jk}}$ shows excess or depletion of reads at any codon position compared with the background that is relative peak height. The geometric mean of this ratio among samples produces the bias coefficient for that position. If the goal of the study is to investigate local patterns of stalling between groups of samples, group-specific bias coefficients should be calculated. CELP coefficients or summary statistics derived from them can be then regressed against any position-specific (for example, RBP binding site or codon type), transcript-specific (for example, length or existence of known upstream open reading frames) or group-specific (for example, wild type versus tRNA KD cell line) factors to infer their effects on stalling. On the other hand, if CELP is primarily used to debias RPF counts to allow an unbiased TER test, all samples in the dataset can be pooled together in the calculation of bias coefficients.

**TER test.** Both RNA and RPF libraries are mapped to the same reference transcriptome as described in the previous section. RNA read counts per transcript are calculated directly from the bam files. RPF read counts are either obtained in the same way or run through CELP debiasing first to smooth out local non-uniformities (described above in detail). RNA and RPF 'transcript × sample' count matrices are normalized separately for library size variation using the median-of-ratios method[62]. We model TE as the odds of retrieving two different sequencing read types from a sample: RPF versus RNA. In this scenario, we hypothetically pool all the reads from an experiment, and then extract a read from this pool. The odds of extracting an RPF versus an RNA read from this pool yields a probabilistic estimation of TE. We compare TER by testing the effect of model covariates on TE, that is, the odds ratio of RPF/RNA between groups or per unit change of continuous predictors. In the very simple case of comparing only two non-replicated samples, a significance test on TER could be performed using a chi-square or Fisher's exact test on a 2 × 2 contingency table with sample name acting as the exposure (independent variable) and the read type (RPF or RNA) as the response (dependent variable). Since most biological experiments are replicated and involve multiple sample groups, we generalize the test in a logistic regression setting:

$$\log TE = \log \left( \frac{RPF}{RNA} \right) = \alpha + \sum_i \beta_i X_i$$

where:

RPF: normalized (and optionally debiased) RPF read count

RNA: normalized RNA read count

$\alpha$: intercept

$X_i$: predictor (independent variable) $i$

$\beta_i$: regression coefficient for predictor (independent variable) $i$.

The test is run separately for each transcript. Independent variables could be categorical, for example, group labels, or continuous to represent a molecular measurement from the sample for example tRNA concentrations or a codon optimality score. This formulation of TER accommodates complex experimental designs with any number of groups or replicates described by any number of attributes (covariates). It can incorporate interaction terms, batch effect indicators or other confounding variables. A $P$ value is reported for each regression coefficient indicating the significance of its effect ('effect' here is defined as a regression coefficient being different from 0, or the corresponding TER being different from 1). The effect sizes (logTER) and $\log_{10}(P$ values) are plotted together to produce the familiar volcano plot. The expected TER of a transcript between two samples differing in one or multiple attributes can be estimated by substituting the obtained regression coefficients in the equation below:

$$TER = \frac{TE_2}{TE_1} = \exp \left\{ \sum_i \beta_i (X_{i,2} - X_{i,1}) \right\}$$

For detailed instructions to install the package, prepare the input data, run the tests, and interpret and plot the results, visit https://github.com/goodarzilab/Ribolog. Additional modules for quality control, empirical null significance testing to reduce false positives, meta-analysis of ribosome profiling data and so on are also available from the GitHub page. We used the data simulated by the authors of Xtail to benchmark Ribolog against four other commonly used tools (Xtail, Riborex, RiboDiff and Anota2seq). The results are available at https://github.com/goodarzilab/Ribolog/blob/master/benchmarks/ribolog_benchmarks.pdf.

**APAlog.** The RNA reads used to compare poly(A) site usage could originate from a regular RNA-seq or a specialized 3' UTR sequencing protocol. In either case, normalized counts of reads mapped to each poly(A) site are used by APAlog to assess the extent and pattern of differential poly(A) site usage via multinomial logistic regression:

Article

$$\log\left(\frac{\text{Alt.site}}{\text{Ref.site}}\right) = \alpha + \sum_i \beta_i X_i$$

where:

Alt.site: alternative poly(A) site normalized read count
Ref.site: reference poly(A) site normalized read count
$\alpha$: intercept
$X_i$: predictor (independent variable) $i$
$\beta_i$: regression coefficient for predictor (independent variable) $i$.

APAlog automatically sets the poly(A) site of each transcript that comes first alphabetically to reference. The user can specify which poly(A) site to serve as reference by adjusting the poly(A) site names in the count matrix. APAlog can be run in three modes: (1) Overall transcript-wise test: a deviance test is performed between the fitted model with covariates and the null (intercept-only) model. This test identifies transcripts that show differential poly(A) site selection among samples but does not specify which poly(A) sites or covariates contribute to the difference. This mode facilitates the quick scanning of a large multi-group dataset to flag putative targets of regulation. Moreover, by performing exactly one test per transcript, it avoids complications of multiple testing correction among transcripts with unequal number of poly(A) sites. (2) Alternatives versus reference test: One poly(A) site per transcript is marked as the reference site, and all others (one or more) are tested against it. This mode is suitable for specific applications such as testing 3′ UTR length variation when one poly(A) site, in this case the most proximal one, can be set to reference and all others compared with it. (3) Pairwise test: this test compares all pairs of poly(A) sites per transcript and provides the highest-resolution view of poly(A) site selection regulation. It is also the best choice if a reference or canonical poly(A) site cannot be logically assigned.

For detailed instructions to install the package, prepare the input data, run the tests and interpret the results, visit https://github.com/goodarzilab/APAlog.

## Statistics and reproducibility
For in vivo experiments, mice were distributed into cohorts with five mice per cohort, which in NSG background is enough to observe a more than two-fold difference with 90% confidence. For other experiments, no statistical methods were used to calculate sample size.

No data were excluded from the analyses.

Cell migration/invasion assays were performed in four biological replicates. Co-IP–MS, TMT–MS, RT–qPCR and western blot experiments were performed in biological triplicates. Sequencing-based experiments (Ribo-seq, RNA-seq, CLIP-seq, PAPERCLIP and massively parallel reporter assays (MPRA)) were performed in biological duplicates.

Mice for in vivo experiments were randomly assigned into cohorts. For other experiments, no randomization was performed.

For cell migration/invasion assays, the person counting the colonies was blinded for the experimental conditions. For other experiments, the data were acquired and analysed by the same person.

Data distribution was assumed to be normal, but this was not formally tested.

## Reporting summary
Further information on research design is available in the Nature Portfolio Reporting Summary linked to this article.

## Data availability
All sequencing data have been deposited in the GEO database under accession GSE186647. Proteomics data have been deposited in the PRIDE database under accession PXD029560. A previously published HNRNPC iCLIP dataset (E-MTAB-1371) was used in this study. The human breast cancer data were derived from the TCGA (at Genomic Data Commons, https://gdc.cancer.gov). The METABRIC dataset was obtained from cBioPortal (https://www.cbioportal.org). The CPTAC breast cancer dataset was obtained from Proteomics Data Commons (https://proteomic.datacommons.cancer.gov/pdc/). All other data supporting the findings of this study are available from the corresponding author on reasonable request. Source data are provided with this paper.

## Code availability
For detailed instructions on how to install Ribolog package, prepare the input data, run the tests, and interpret and plot the results, visit https://github.com/goodarzilab/Ribolog. For detailed instructions on how to install APAlog package, prepare the input data, run the tests and interpret the results, visit https://github.com/goodarzilab/APAlog.

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

## Acknowledgements

We acknowledge the UCSF Center for Advanced Technology (CAT) for high-throughput sequencing and other genomic analyses. We thank S. F. Tavazoie for the gift of the HCC1806 and HCC1806-LM2 cell lines. We thank the Preclinical Therapeutics core as well as the Laboratory Animal Resource Center (LARC) at UCSF. We acknowledge support from our colleagues at the Helen Diller Family Comprehensive Cancer Center and the Breast Oncology Program. Funding**:** This work was supported by grants from the NIH (R00CA194077 and R01CA240984) and ACS (130920-RSG-17-114-01-RMC) to H.G. This research was also supported by funding from the UCSF Helen Diller Family Comprehensive Cancer Center Breast Oncology Program (the content is solely the responsibility of the authors). This study was supported in part by UCSF Laboratory for Cell Analysis shared resource facility through a grant from NIH (P30CA082103). This work used the Vincent J. Proteomics/Mass Spectrometry Laboratory at UC Berkeley, supported in part by NIH S10 Instrumentation Grant S10RR025622. L.F. was supported by NIH training grant T32CA108462-15. A.N. was supported by DoD PRCRP Horizon Award W81XWH-19-1-0594. H.A. was supported by the NIH Ruth L. Kirschstein National Research Service Award F32GM133118. S.Z. was supported by an HHMI medical research fellowship. M.D. and F.M. were supported by an MRC career development award to F.M. (MR/P009417/1). D.M. was supported by an MD fellowship from the Boehringer Ingelheim Fonds. J.W. was supported by an EMBO long-term post-doctoral fellowship (EMBO ALTF 159-2017). A.G. and J.W. were supported by U01 CA199315, Mark Foundation and CDMRP DoD Breakthrough Award W81XWH-16-1-0603. H.G. and A.G. were supported by the Atwater Foundation. The funders had no role in study design, data collection and analysis, decision to publish or preparation of the manuscript. Finally, we acknowledge the support of our late colleague Z. Werb for the PDX studies reported here.

## Author contributions

H.G. and A.G. conceptualized the study. A.N. performed Ribo-seq, RNA-seq, CLIP-seq, PAPERCLIP, co-IP, TMT labelling, western blotting, qPCR and reporter assay experiments. L.F. performed CLIP-seq. J.W. performed PDX transplantation experiments. H.A. developed the Ribolog and APAlog statistical methods and R packages. H.A. and S.M. set up the Ribolog and APAlog GitHub repositories and maintain the R packages. H.G., H.A. and S.M. performed the Ribolog benchmark study. A.N., K.G., D.M., B.C., S.Z., T.J., K.Y., P.N. and N.S. generated cell lines and performed in vivo metastasis experiments. A.N., P.N., T.J. and K.Y. performed T4 treatment experiments. A.N., T.J. and K.Y. performed cell proliferation, colony formation, cell migration and invasion assays. M.D. and F.M. performed MS and analysis. H.-W.H. contributed to mouse experiments. H.G. analysed the Ribo-seq, RNA-seq data, TCGA data and clinical data. A.N., H.A., L.F., A.G. and H.G. wrote the manuscript. H.G. supervised all research.

## Competing interests

The authors declare no competing interests.

## Additional information

**Extended data** is available for this paper at https://doi.org/10.1038/s41556-023-01141-9.

**Correspondence and requests for materials** should be addressed to Hani Goodarzi.

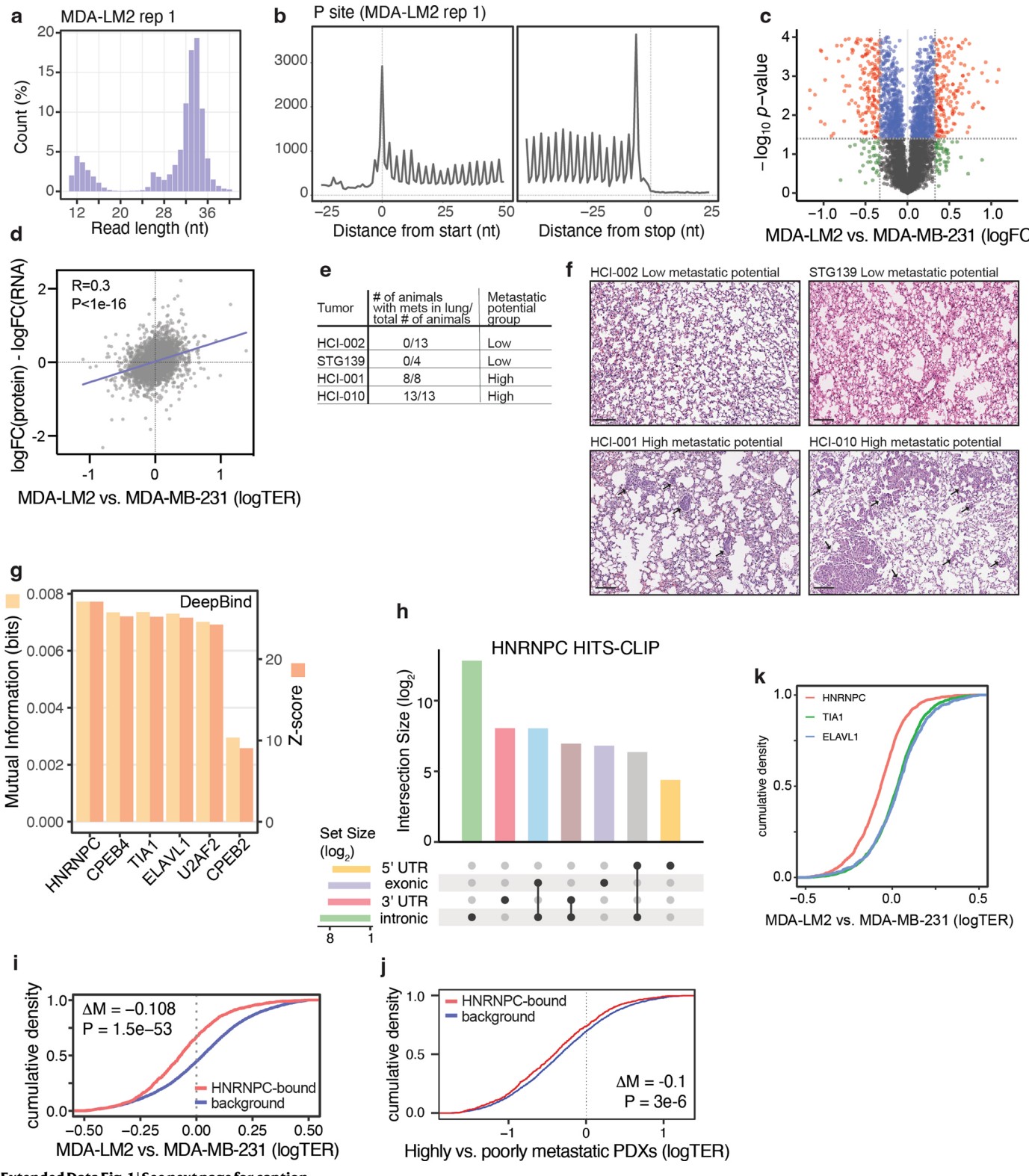

**Extended Data Fig. 1 | See next page for caption.**

**Extended Data Fig. 1 | HNRNPC target mRNAs are translationally repressed in highly metastatic breast cancer cells and PDXs. (a)** Length distribution of ribosome protected footprints (RPFs) as determined by Ribo-seq. A representative data sample is shown. **(b)** Distribution of RPFs, aligned on an inferred ribosome P-site, on a metagene, centered around translation start (left) or stop (right) site. A representative data sample is shown. **(c)** Volcano plot illustrating the changes in protein abundance in MDA-LM2 compared to MDA-MB-231 cells, as determined by TMT-MS analysis. The data points are colored according to thresholds in effect size (logFC ± 0.33) and significance ($p < 0.05$, two-tailed $t$-test). **(d)** The distribution of changes in TEs (as determined by Ribo-seq) and in protein abundance (as determined by TMT-MS and normalized by RNA expression obtained from RNA-seq), in MDA-LM2 compared to MDA-MB-231 cells. Pearson R and associated $p$-value are shown. **(e)** The comparison of the metastatic capacity of breast cancer PDXs used in this study. **(f)** Representative images ($n$ = 4-13 mice per cohort, as in (e)) of H&E stained mouse lung sections transplanted with breast cancer PDXs. The metastatic foci are indicated by black arrows. Scale bar = 100 μm. **(g)** Mutual information (MI) values and associated $z$-scores from the DeepBind algorithm, showing the prediction of poly(U) binding protein targets among translationally repressed mRNAs in MDA-LM2 compared to MDA-MB-231 cells. **(h)** Upset plot showing the distribution and overlap of HNRNPC peaks within genomic features, as determined by CLIP-seq. **(i)** Cumulative density plot of translation efficiency ratios (TER) comparing MDA-LM2 to MDA-MB-231 cells, for HNRNPC 3′ UTR target and non-target mRNAs. **(j)** Cumulative density plot of translation efficiency ratios (TER) comparing highly and poorly metastatic breast cancer PDXs, for HNRNPC 3′ UTR target and non-target mRNAs. Median difference (ΔM) and $p$-value (calculated using two-tailed Mann-Whitney $U$-test) are shown. **(k)** Cumulative density plot of translation efficiency ratios (TER) comparing MDA-LM2 to MDA-MB-231 cells, for HNRNPC, ELAVL1 and TIA1 3′ UTR target mRNAs, as determined by CLIP-seq.

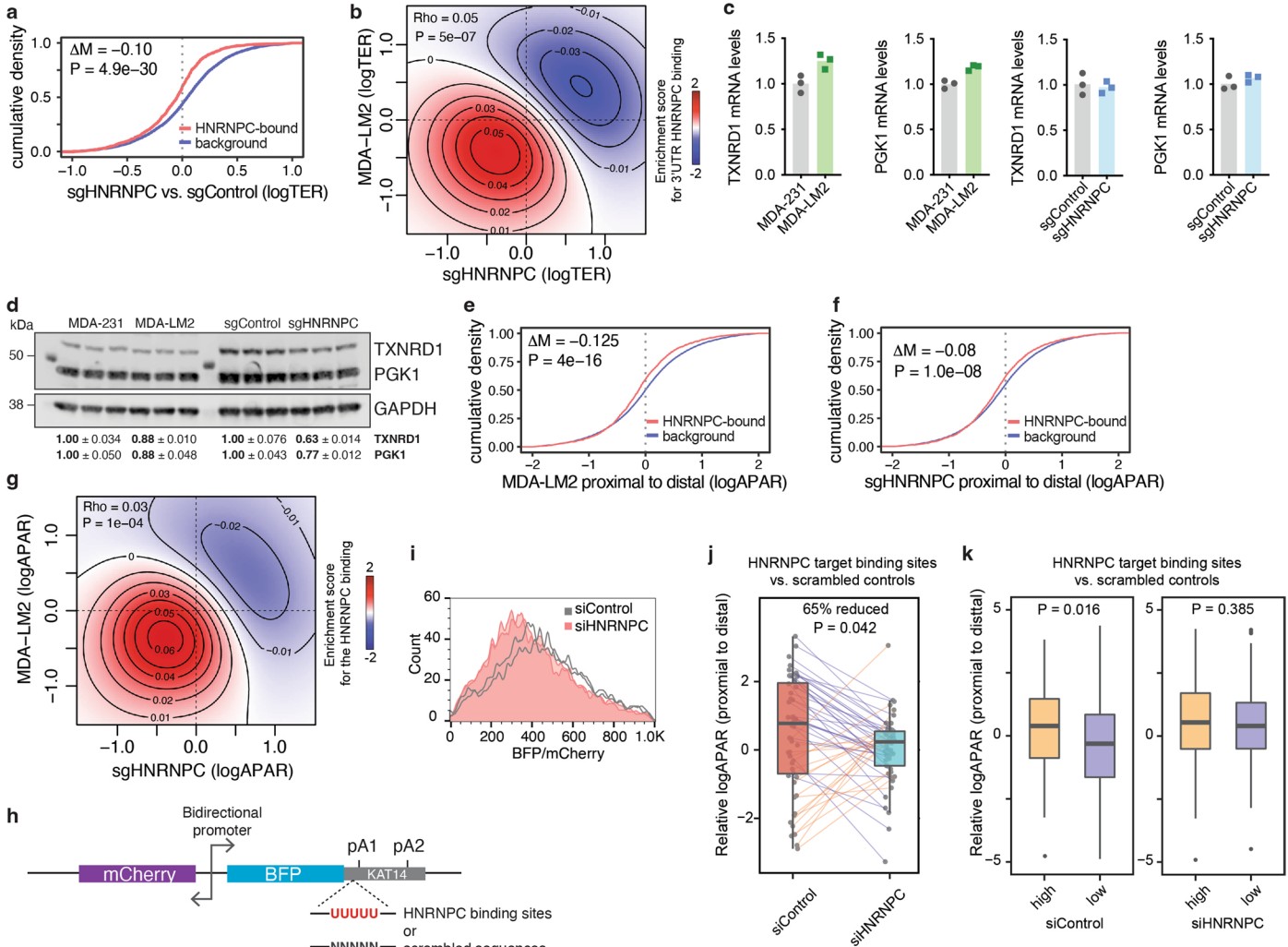

**Extended Data Fig. 2 | HNRNPC binding impacts the translation and alternative polyadenylation of its targets. (a)** Cumulative density plot of translation efficiency ratios (TER) comparing sgHNRNPC to sgControl MDA-MB-231 cells, for HNRNPC 3′ UTR target and non-target mRNAs. Median difference (ΔM) and *p*-value (calculated using two-tailed Mann-Whitney *U*-test) are shown. **(b)** Two-dimensional heatmap showing significant logTER correlation of translationally repressed mRNAs in MDA-LM2 and HNRNPC knockdown (sgHNRNPC) cells. For comparison, the Spearman correlation coefficient and the associated *p*-value are shown across all genes. **(c)** TXNRD1 and PGK1 mRNA levels in MDA-MB-231 and MDA-LM2, as well as sgControl and sgHNRNPC-expressing cells, as determined by RNA-seq. **(d)** Western blot analysis of TXNRD1 and PGK1 proteins in MDA-MB-231 and MDA-LM2, as well as sgControl and sgHNRNPC-expressing cells. Relative protein quantity normalized to GAPDH is indicated below, along SE. The western blot was performed once in biological triplicates to confirm the TMT-MS results. **(e)** Cumulative density plot of alternative polyadenylation ratios (logAPAR) comparing MDA-LM2 to MDA-MB-231 cells, for HNRNPC 3′ UTR target and non-target mRNAs; statistics as in (a). **(f)** Cumulative density plot of logAPAR comparing sgHNRNPC to sgControl cells, for HNRNPC

3′ UTR target and non-target mRNAs; statistics as in (a). **(g)** Two-dimensional heatmap showing significant logAPAR correlation of proximal to distal poly(A) site switch in MDA-LM2 and HNRNPC knockdown (sgHNRNPC) cells; statistics as in (b). **(h)** Bidirectional promoter reporter schematics, used for massively parallel reporter assays (MPRA). **(i)** BFP/mCherry ratio of reporter-expressing MDA-MB-231 cells, transfected with control or HNRNPC-targeting siRNAs, as detected by flow cytometry. **(j)** Box plot illustrating the relative reporter logAPAR, comparing transcripts with HNRNPC-binding sites versus matched scrambled controls, in control and HNRNPC KD cells. *n* = 2 biological replicates. *p*-value calculated using one-tailed Wilcoxon signed-rank test. Box center reports the median value, the boundaries - the quartiles, and the whiskers - the 10 and 90 percentiles. **(k)** Box plot illustrating the relative reporter logAPAR, comparing transcripts with HNRNPC-binding sites versus matched scrambled controls, in control and HNRNPC KD cells, stratified by reporter protein expression (25% high versus 25% low BFP/mCherry ratio, as determined and sorted by flow cytometry). *n* = 2 biological replicates. *p*-value calculated using one-tailed *t*-test. Box plot characteristics as in (j).

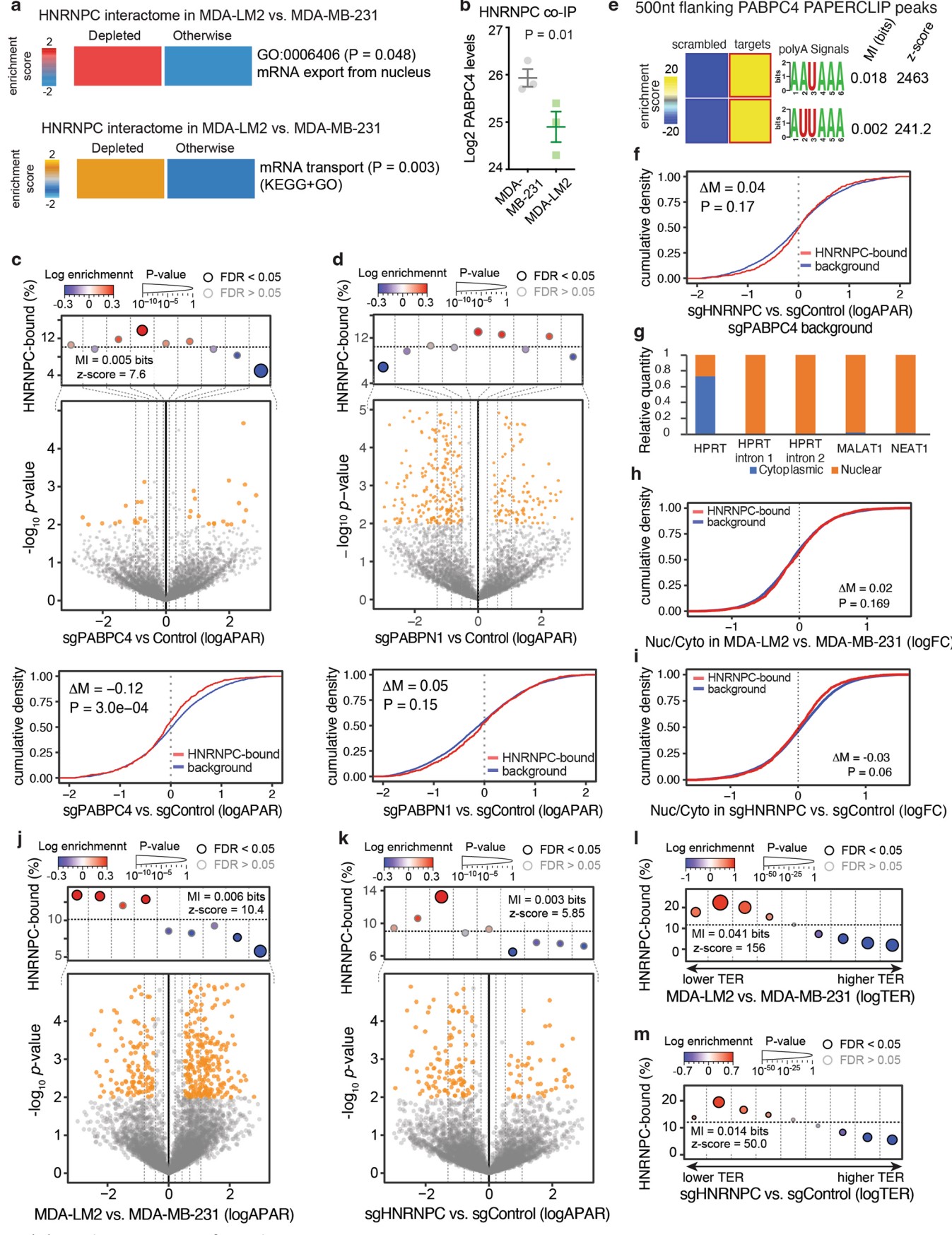

**Extended Data Fig. 3 | See next page for caption.**

**Extended Data Fig. 3 | PABPC4 acts in concert with HNRNPC to control alternative polyadenylation and directs its target mRNAs to AGO2-dependent translational repression. (a)** Significant depletion of selected gene ontology (GO) terms in the HNRNPC interactome in MDA-LM2 compared to MDA-MB-231 cells, as determined by coIP-MS. Also reported are the associated empirical *p*-values from permutation tests. **(b)** Enrichment of PABPC4 in HNRNPC coIP-MS data in MDA-MB-231 and MDA-LM2 cells. *n* = 3 biological replicates. Individual data points and mean values are shown, error bars indicating SEM; *p*-value calculated using parametric linear model controlling for input abundances. **(c-d)** Comparison of alternative polyadenylation ratio (logAPAR) in PABPC4 knockdown (sgPABPC4) (c) or PABPN1 knockdown (sgPABPN1) (d) and control (sgControl) cells, as in Fig. 2d. Below, cumulative density plots as in Extended Data Fig. 2e are shown. **(e)** Heatmaps showing the enrichment of canonical poly(A) signals in the vicinity of PABPC4 binding peaks (as determined by PAPER-CLIP). The bolded border denotes a statistically significant enrichment (hypergeometric test, corrected *p* < 0.05; red). MI values and associated *z*-scores are shown. **(f)** Cumulative density plot of logAPAR comparing sgHNRNPC/sgPABPC4 (double knockdown) to sgControl/sgPABPC4 cells, for HNRNPC 3′ UTR target and non-target mRNAs, as in (c). **(g)** Relative RNA quantity in cytoplasmic versus nuclear fraction, as determined by RTqPCR. **(h-i)** Cumulative density plot of logAPAR comparing MDA-MB-231 and MDA-LM2 (h) or MDA-MB-231 sgControl and sgHNRNPC (i) cells, in nuclear versus cytoplasmic fractions, for HNRNPC 3′ UTR target and non-target mRNAs, as in (c). **(j-k)** Comparison of logAPAR from cytoplasmic RNAs, in MDA-LM2 and MDA-MB-231 (j) or MDA-MB-231 sgControl and sgHNRNPC (k) cells, as in (c). **(l-m)** Enrichment patterns of HNRNPC target mRNAs in the cytoplasm as a function of logTER between MDA-LM2 and MDA-MB-231 (l) or MDA-MB-231 sgControl and sgHNRNPC (m) cells, as in Fig. 1d.

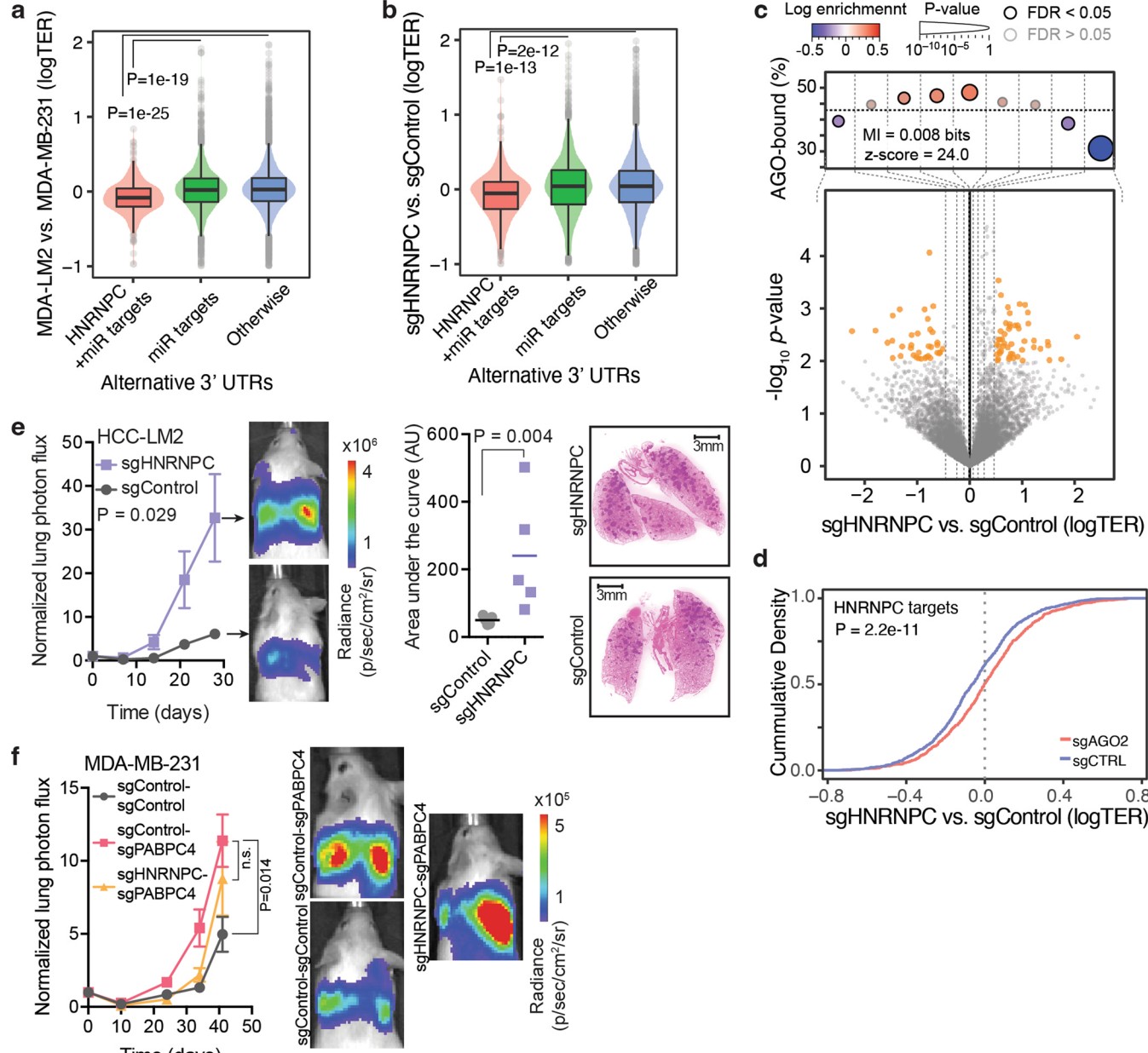

**Extended Data Fig. 4 | HNRNPC levels impact *in vivo* metastatic colonization of breast cancer cells. (a)** Violin plots showing the distribution of translation efficiency ratios (logTER) comparing MDA-LM2 to MDA-MB-231 cells among the miRNA target, joint HNRNPC and miRNA target, and non-target mRNA 3′ UTRs. $n = 2$ biological replicates. *p*-values calculated using two-tailed Mann-Whitney *U*-test. Box center reports the median value, the boundaries - the quartiles, and the whiskers - the 10 and 90 percentiles. **(b)** Violin plots showing the distribution of translation efficiency ratios (logTER) comparing sgHNRNPC to sgControl cells among the miRNA target, joint HNRNPC and miRNA target, and non-target mRNA 3′ UTRs. $n = 2$ biological replicates. *p*-values calculated using two-tailed Mann-Whitney *U*-test. Box plot characteristics as in (a). **(c)** Bottom: Volcano plot showing distribution of changes in translation efficiency ratio (logTER) in sgHNRNPC compared to sgControl cells. Top: Enrichment of the AGO2 targets as a function of logTER between sgHNRNPC and sgControl cells; statistics

as in Fig. 2a. **(d)** Cumulative density plot of logTER (HNRNPC 3′ UTR targets) comparing sgHNRNPC to sgControl cells, in AGO2 knockdown (sgAGO2) and control (sgControl) conditions; statistics as in Extended Data Fig. 2a. **(e)** HCC1806-LM2 cells stably expressing sgHNRNPC or sgControl were injected via tail vein into NSG mice. Bioluminescence was measured at the indicated times (mean values are shown, error bars indicating SEM; *p*-value calculated using two-way ANOVA); area under the curve was measured at the final time point (*p*-value calculated using one-tailed Mann-Whitney *U*-test). Lung sections were stained with H&E (representative images shown). $n = 4$-5 mice per cohort. **(f)** MDA-MB-231 cells stably expressing sgControl, sgPABPC4 or sgPABPC4/sgHNRNPC were injected via tail vein into NSG mice. Bioluminescence was measured at the indicated times (mean values are shown, error bars indicating SEM; *p*-value calculated using two-way ANOVA). $n = 4$-5 mice per cohort.

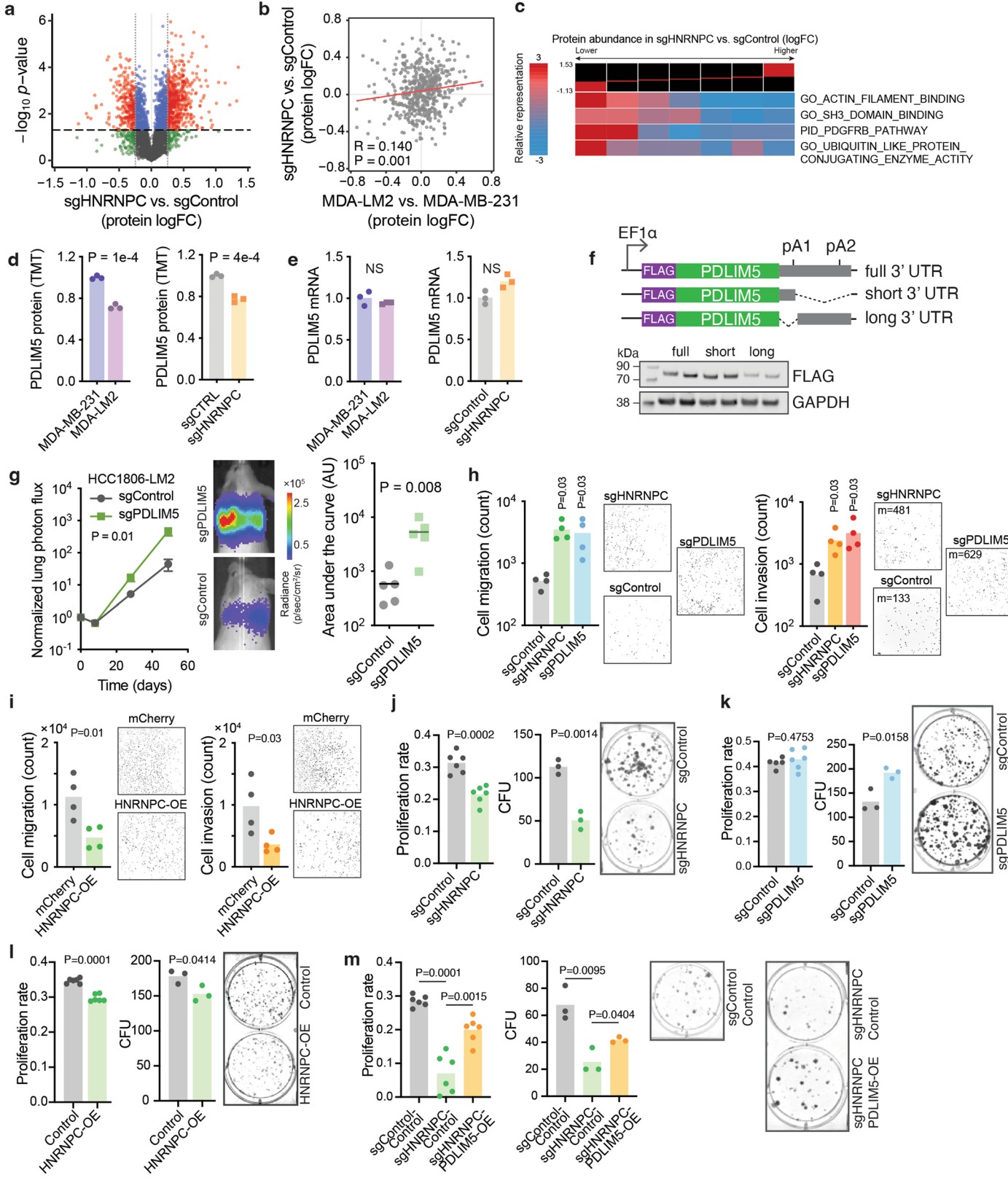

**Extended Data Fig. 5 | See next page for caption.**

**Extended Data Fig. 5 | PDLIM5 acts downstream of HNRNPC to suppress breast cancer metastasis. (a)** Volcano plot illustrating the changes in protein abundance in HNRNPC knockdown (sgHNRNPC) compared to control (sgControl) MDA-MB-231 cells, as determined by TMT-MS analysis. The data points are colored according to thresholds in effect size (logFC ± 0.25) and significance ($p < 0.05$, two-tailed $t$-test). **(b)** The distribution of changes in protein abundance in MDA-LM2 vs. MDA-MB-231 cells and sgHNRNPC vs. sgControl cells, as determined by TMT-MS. Pearson R and associated $p$-value are shown. **(c)** Gene-set enrichment analysis of the data depicted in (a). **(d)** Quantification of PDLIM5 protein expression in MDA-MB-231 and MDA-LM2 (left), or sgControl and sgHNRNPC (right) cells, as determined by TMT-MS. $n = 3$ biological replicates. $p$-values calculated using one-tailed Student's $t$-test. **(e)** Quantification of relative PDLIM5 mRNA expression (normalized to HPRT) in MDA-MB-231 and MDA-LM2

cells (left) or sgControl and sgHNRNPC cells (right), as determined by RTqPCR. $n = 3$ biological replicates. **(f)** Reporter schematics of testing PDLIM5 3′ UTR variants. Representative western blot image of two independent experiments is shown below. **(g)** HCC1806-LM2 cells stably expressing sgPDLIM5 or sgControl were injected via tail vein into NSG mice. Bioluminescence was measured at the indicated times (mean values are shown, error bars indicating SEM; $p$-value calculated using two-way ANOVA); area under the curve was measured at the final time point ($p$-value calculated using one-tailed Mann-Whitney $U$-test). $n = 4$-5 mice per cohort. **(h-i)** Cell migration (left) and cell invasion (right) measurements of MDA-MB-231 cells. $n = 4$ biological replicates. $p$-values calculated using two-tailed Mann-Whitney $U$-test. **(j-m)** Proliferation rates (left) and colony forming units (CFU) (right) of MDA-MB-231 cells. $n = 6$ (proliferation rate) or 3 (CFU) biological replicates. $p$-values calculated using two-tailed unpaired $t$-test.

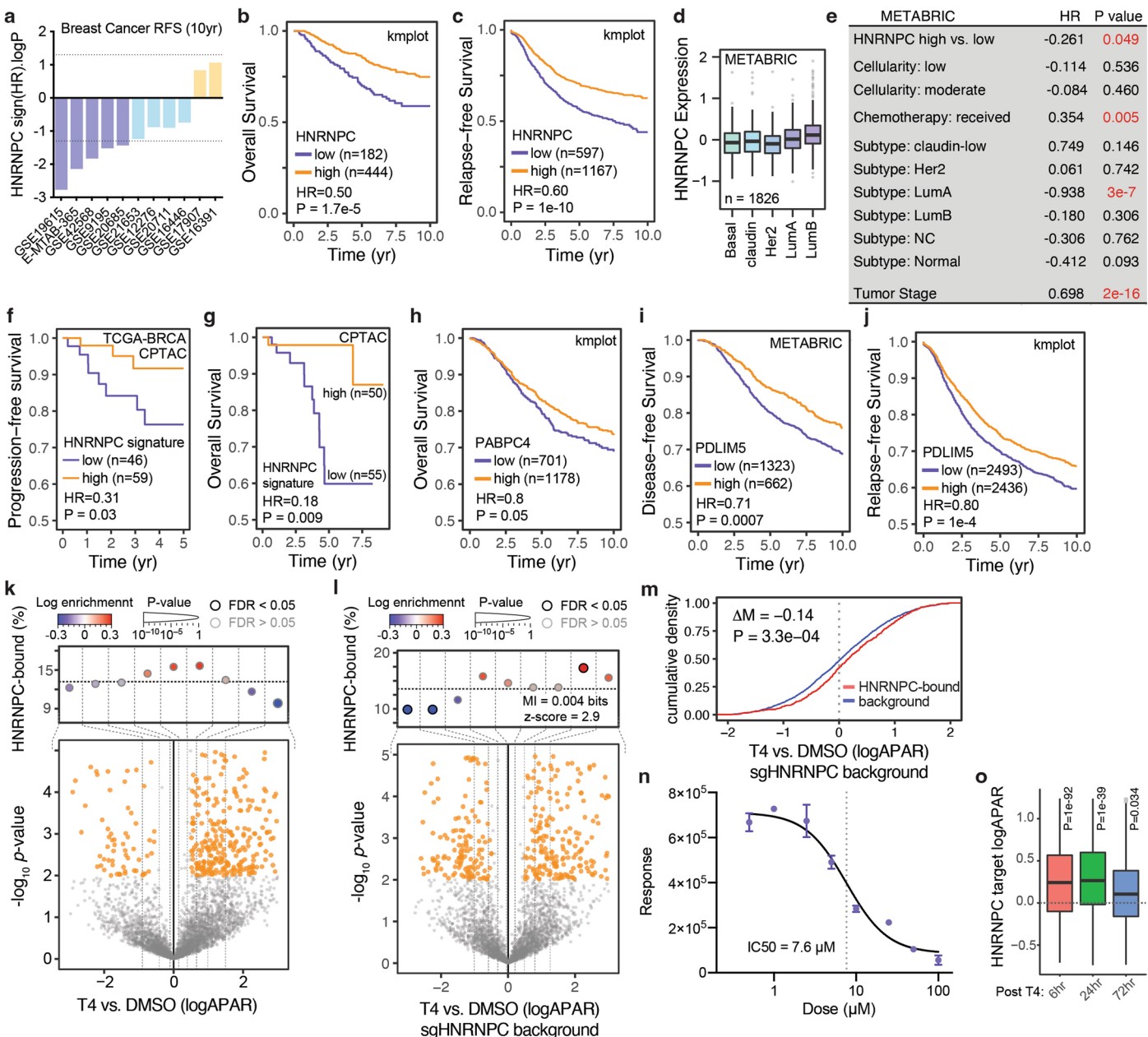

**Extended Data Fig. 6 | HNRNPC expression is associated with clinical outcomes in breast cancer patients. (a)** Distribution of 10-year relapse-free survival *p*-values (two-sided log rank test results reported as −log *p* for positive association and log *p* for negative) of the association of HNRNPC expression and clinical outcome in the listed 10 breast cancer datasets. Violet bars show associations that pass the statistical threshold (−log *p* < -1.3, FDR-corrected two-sided log-rank test (FDR < 0.1)), blue bars are trending negative, and yellow bars are trending positive. The statistical threshold was adjusted as 10/number of datasets. **(b-c)** Kaplan-Meier survival curves showing association between tumor HNRNPC levels and overall (b) or relapse-free (c) survival in a collection of breast cancer patient cohorts. Hazard ratios (HR) and *p*-values (calculated using log-rank test) are shown. **(d)** HNRNPC mRNA levels across breast tumor subtypes in the METABRIC cohort. Box center reports the median value, the boundaries - the quartiles, and the whiskers - the 10 and 90 percentiles. **(e)** Multivariate survival analysis (Cox proportionate-hazards model) of breast cancer patients in the METABRIC cohort with HNRNPC expression as one of the factors. *P* < 0.05 are highlighted in red. LumA, luminal A; LumB, luminal B;

NC, not classified. **(f-g)** Kaplan-Meier survival curve showing association between tumor HNRNPC signature protein levels and progression-free (f) or overall (g) survival in the TCGA-BRCA CPTAC cohort. **(h-j)** Kaplan-Meier survival curve showing association between tumor PABPC4 (h) or PDLIM5 (i-j) levels and overall (h) or disease-free (i-j) survival in a collection of breast cancer patient cohorts. **(k-l)** Comparison of alternative polyadenylation ratio (logAPAR) in T4 and DMSO-treated MDA-MB-231 (k) or HNRNPC knockdown (sgHNRNPC) (l) cells, as in Fig. 2d. **(m)** Cumulative density plot of logAPAR comparing T4- to DMSO-treated HNRNPC knockdown (sgHNRNPC) cells, as in Extended Data Fig. 2e. **(n)** Dose-response measurements for 6-hour T4 treatment and corresponding cell viability, determined 72 hours post-treatment. *n* = 6 biological replicates. Mean values are plotted, error bars indicating SD. **(o)** Box plots illustrating the changes in alternative polyadenylation (logAPAR) of HNRNPC targets between T4-treated (sampled at indicated time points) and untreated MDA-LM2 cells. *n* = 3 biological replicates. *p*-values calculated using two-tailed Wilcoxon signed-rank test. Box plot characteristics as in (d).

# Reporting Summary

## Statistics

For all statistical analyses, confirm that the following items are present in the figure legend, table legend, main text, or Methods section.

| n/a | Confirmed | |
|---|---|---|
| ☐ | ☒ | The exact sample size (*n*) for each experimental group/condition, given as a discrete number and unit of measurement |
| ☒ | ☐ | A statement on whether measurements were taken from distinct samples or whether the same sample was measured repeatedly |
| ☐ | ☒ | The statistical test(s) used AND whether they are one- or two-sided *Only common tests should be described solely by name; describe more complex techniques in the Methods section.* |
| ☐ | ☒ | A description of all covariates tested |
| ☐ | ☒ | A description of any assumptions or corrections, such as tests of normality and adjustment for multiple comparisons |
| ☐ | ☒ | A full description of the statistical parameters including central tendency (e.g. means) or other basic estimates (e.g. regression coefficient) AND variation (e.g. standard deviation) or associated estimates of uncertainty (e.g. confidence intervals) |
| ☐ | ☒ | For null hypothesis testing, the test statistic (e.g. *F*, *t*, *r*) with confidence intervals, effect sizes, degrees of freedom and *P* value noted *Give P values as exact values whenever suitable.* |
| ☒ | ☐ | For Bayesian analysis, information on the choice of priors and Markov chain Monte Carlo settings |
| ☒ | ☐ | For hierarchical and complex designs, identification of the appropriate level for tests and full reporting of outcomes |
| ☐ | ☒ | Estimates of effect sizes (e.g. Cohen's *d*, Pearson's *r*), indicating how they were calculated |

*Our web collection on statistics for biologists contains articles on many of the points above.*

## Software and code

Policy information about availability of computer code

| Data collection | No software was used for data collection. |
|---|---|
| Data analysis | Sequencing data was analyzed using custom R and Python scripts, using these tools: cutadapt (v2.3), FASTX-Toolkit (v0.0.13), umitools (v0.3.3), bowtie2 (v2.3.5), salmon (v0.14.1), ctk (v1.1.13), bwa (v0.7.17). The Ribolog and APAlog have been deposited to GitHub (https://github.com/goodarzilab/Ribolog, https://github.com/goodarzilab/APAlog). MaxQuant (v1.6.3.3) and Perseus (v1.6.2.3) were used for mass spectrometry data analysis. Living Image (v4.7.3) was used to acquire in vivo imaging data with IVIS instrument (Perkin Elmer). |

For manuscripts utilizing custom algorithms or software that are central to the research but not yet described in published literature, software must be made available to editors and reviewers. We strongly encourage code deposition in a community repository (e.g. GitHub). See the Nature Portfolio guidelines for submitting code & software for further information.

## Data

Policy information about availability of data

All manuscripts must include a data availability statement. This statement should provide the following information, where applicable:
- Accession codes, unique identifiers, or web links for publicly available datasets
- A description of any restrictions on data availability
- For clinical datasets or third party data, please ensure that the statement adheres to our policy

All sequencing data have been deposited in the GEO database under accession GSE186647. Proteomics data has been deposited in the PRIDE database under accession PXD029560. A previously published HNRNPC iCLIP dataset (E-MTAB-1371) was used in this study. The human breast cancer data were derived from the TCGA (at Genomic Data Commons, https://gdc.cancer.gov). The METABRIC data set was obtained from cBioPortal (https://www.cbioportal.org). The CPTAC breast

# Field-specific reporting

Please select the one below that is the best fit for your research. If you are not sure, read the appropriate sections before making your selection.

☒ Life sciences   ☐ Behavioural & social sciences   ☐ Ecological, evolutionary & environmental sciences

For a reference copy of the document with all sections, see nature.com/documents/nr-reporting-summary-flat.pdf

# Life sciences study design

All studies must disclose on these points even when the disclosure is negative.

| | |
|---|---|
| Sample size | Based on our previous work (Goodarzi et al., Nature, 2014; Goodarzi et al., Cell, 2015; Goodarzi et al., Cell, 2016), for in vivo experiments, mice were distributed into cohorts with 4-5 mice per cohort, which in NSG background is enough to observe a >2-fold difference with 90% confidence. For example, at t=33 days, average normalized lung photon flux from colonized CN-LM1a breast cancer cells is 150 with s.d. of 66. Based on this distribution, with a cohort size of n=5 in each arm, we can detect a difference of ~100% in size with 90% power. Similarly, in MDA-LM2 cells in the same study, at t=33 days, average normalized signal was recorded as 291 and s.d. of 104, which suggests a cohort size of n=4. For other experiments, no statistical methods were used to calculate sample size. |
| Data exclusions | No data were excluded from the analysis. |
| Replication | Cell migration/invasion assays were performed in 4 biological replicates. CoIP-MS, TMT-MS, RTqPCR, and western blot experiments were performed in biological triplicates. Sequencing-based experiments (Ribo-seq, RNA-seq, CLIP-seq, PAPERCLIP, MPRA) were performed in biological duplicates. |
| Randomization | Mice for in vivo experiments were randomly assigned into cohorts. For other experiments (molecular biology), no randomization was performed. |
| Blinding | For cell migration/invasion assays, the person counting the colonies was blinded for the experimental conditions. For other experiments, the data was acquired and analyzed by the same person and the blinding was not deemed necessary. |

# Reporting for specific materials, systems and methods

We require information from authors about some types of materials, experimental systems and methods used in many studies. Here, indicate whether each material, system or method listed is relevant to your study. If you are not sure if a list item applies to your research, read the appropriate section before selecting a response.

### Materials & experimental systems

| n/a | Involved in the study |
|---|---|
| ☐ | ☒ Antibodies |
| ☐ | ☒ Eukaryotic cell lines |
| ☒ | ☐ Palaeontology and archaeology |
| ☐ | ☒ Animals and other organisms |
| ☒ | ☐ Human research participants |
| ☒ | ☐ Clinical data |
| ☒ | ☐ Dual use research of concern |

### Methods

| n/a | Involved in the study |
|---|---|
| ☒ | ☐ ChIP-seq |
| ☐ | ☒ Flow cytometry |
| ☒ | ☐ MRI-based neuroimaging |

## Antibodies

| | |
|---|---|
| Antibodies used | anti-beta-tubulin (Proteintech 66240-1-Ig) (western blot loading control)<br>anti-GAPDH (Proteintech 60004-1-Ig) (western blot loading control)<br>anti-HNRNPC (Santa Cruz sc-32308) (western blot, IP, CLIP)<br>Mouse IgG (Jackson 015-000-003) (IP control)<br>anti-PABPC4 (Proteintech 14960-1-AP) (western blot, PAPERCLIP)<br>anti-TIA1 (Proteintech 12133-2-AP) (CLIP)<br>anti-ELAVL1 (Proteintech 11910-1-AP) (CLIP)<br>anti-PABPN1 (Proteintech 66807-1-Ig) (western blot)<br>anti-PDLIM5 (Proteintech 10530-1-AP) (western blot)<br>anti-TXNRD1 (Proteintech 11117-1-AP) (western blot)<br>anti-PGK1 (Proteintech 17811-1-AP) (western blot)<br>anti-FLAG tag (Proteintech 66008-1-Ig) (western blot)<br>anti-rabbit IgG, conformation specific, HRP (Cell Signaling 5127S) (western blot) |

anti-mouse IgG, Light chain specific, HRP (Cell Signaling 91196S) (western blot)
anti-rabbit IgG, IRDye 680RD (Li-Cor 926-68071) (western blot)
anti-mouse IgG, IRDye 800CW (Li-Cor 926-32210) (western blot)

| | |
|---|---|
| Validation | All primary and secondary antibodies used in this study are commercially available and have been validated for used applications in human cells by the manufacturers. We have used recommended antibody dilutions for western blot experiments, and detected bands of expected molecular weight, as described by the manufacturers. Prior to CLIP-seq library preparation, the antibodies were tested in IP-western blot experiment, although the antibodies used were validated for IP by the manufacturers. |

## Eukaryotic cell lines

Policy information about cell lines

| | |
|---|---|
| Cell line source(s) | Human cell lines used in this study are available from ATCC:<br>MDA-MB-231 (ATCC HTB-26)<br>HEK293T  (ATCC CRL-3216)<br>HCC1806 (ATCC CRL-2335)<br>MDA-LM2 cell line, the lung metastatic derivative of MDA-MB-231, has been described (ref. 17 in the manuscript), and was a gift from Dr. Joan Massagué.<br>HCC1806-LM2c cell line, the lung metastatic derivative of HCC1806, has been described (Passarelli et al., Nat Cell Bio, 2022), and was a gift from Dr. Sohail Tavazoie. |
| Authentication | The cell lines were authenticated using STR profiling. |
| Mycoplasma contamination | All cell lines have been routinely tested for mycoplasma contamination by a qPCR based assay and tested negative. |
| Commonly misidentified lines<br>(See ICLAC register) | No commonly misidentified cell lines were used in this study. |

## Animals and other organisms

Policy information about studies involving animals; ARRIVE guidelines recommended for reporting animal research

| | |
|---|---|
| Laboratory animals | Mice were housed in accordance with UCSF IACUC protocol in humidity- and temperature-controlled rooms on a 12 hour light-dark cycle with free access to food and water. Seven- to twelve-week-old age-matched female NOD scid gamma mice (NSG, Jackson Labs, 005557) were used in this study. |
| Wild animals | The study did not involve wild animals. |
| Field-collected samples | The study did not involve samples collected from the field. |
| Ethics oversight | This study complies with all relevant ethical regulations that are approved by UCSF Institutional Review Board (IRB) and Institutional Animal Care and Use Committee (IACUC, approval number AN179718). |

Note that full information on the approval of the study protocol must also be provided in the manuscript.

## Flow Cytometry

### Plots

Confirm that:

☒ The axis labels state the marker and fluorochrome used (e.g. CD4-FITC).

☒ The axis scales are clearly visible. Include numbers along axes only for bottom left plot of group (a 'group' is an analysis of identical markers).

☒ All plots are contour plots with outliers or pseudocolor plots.

☒ A numerical value for number of cells or percentage (with statistics) is provided.

### Methodology

| | |
|---|---|
| Sample preparation | MDA-MB-231 cells were trypsinized, washed in 1x PBS, and resuspended in FACS buffer (1x PBS, 1% FBS, 1 mM EDTA) for analysis/sorting. |
| Instrument | The cells were sorted using FACSAria II (BD Biosciences) cell sorter, equipped with 360, 405, 488, 561 and 633 nm lasers. |
| Software | FACSDiva (BD Biosciences) was used to monitor data acquisition. FlowJo (BD Biosciences) was used for post-acquisition data analysis. |
| Cell population abundance | The cells were sorted based on the BFP/mCherry ratio, and 25% top and bottom fractions were collected for analysis. 300k (siCTRL-1), 350k (siCTRL-2), 400k (siHNRNPC-1) and 500k (siHNRNPC-2) cells per bin were sorted for RNA extraction. The purity of the cells post-sorting was not assessed. |

Gating strategy   The cells were gated by FSC-A/SSC-A parameters, the single cells were enriched by FSC-A/FSC-W gating, the mCherry-positive cells were gated by comparing the library-expressing cells with untransduced MDA-MB-231 cells. In each sample, mCherry-positive cells were sorted based on the BFP/mCherry ratio into two bins, corresponding to 25% top and bottom fractions.

☒ Tick this box to confirm that a figure exemplifying the gating strategy is provided in the Supplementary Information.

