## [Peer Review File · Nature Cell Biology]

Peer Review Information

Journal: Nature Cell Biology

Manuscript Title: An mRNA processing pathway suppresses metastasis by governing translational control from the nucleus

Corresponding author name(s): Hani Goodarzi

Editorial Notes:

Reviewer Comments & Decisions:

Decision Letter, initial version:
--

*Please delete the link to your author homepage if you wish to forward this email to co-authors.

Dear Dr Goodarzi,

Your manuscript, "An mRNA processing pathway suppresses metastasis by governing translational control from the nucleus", has now been seen by 3 referees, who are experts in alternative polyadenylation (referee 1); translation (referee 2); and breast cancer and translation (referee 3). As you will see from their comments (attached below) they find this work of potential interest, but have raised substantial concerns, which in our view would need to be addressed with considerable revisions before we can consider publication in Nature Cell Biology.

Nature Cell Biology editors discuss the referee reports in detail within the editorial team, including the chief editor, to identify key referee points that should be addressed with priority, and requests that are overruled as being beyond the scope of the current study. To guide the scope of the revisions, I have listed these points below. We are committed to providing a fair and constructive peer-review process, so please feel free to contact me if you would like to discuss any of the referee comments further.

In particular, it would be essential to:

a) address the concerns regarding the Ribo-seq data, as noted by:

Referee 1:

2. The authors appear to have used total RNA to normalize ribo-seq data. This approach would miss regulation of nuclear export. They should use cytoplasmic RNA for normalization.

Referee 2:

1. In discussing the new "ribolog" tool for Ribo-seq analysis, it is claimed that it has "as few a priori assumptions as possible" and "does not assume a negative binomial (NB) distribution of read counts" (p. 4, ll. 8 - 18). However, logistic regression implicitly assumes a standard binomial error model, which is strictly no more conservative than the NB read count model used in many RNA-Seq tools, and in practice will be less conservative. In particular, the NB models incorporates an overdispersion term, while ribolog effectively assumes that this term is zero. Lower variance estimates then lead to stronger statistical significances.

2. As a related point, the discussion of the Ribolog method (p. 17) mentions problems that arise in Ribo-seq data analysis but does not credit the fact that existing tools offer clear solutions to these problems.

First, "variance sharing" underlies many Ribo-seq and RNA-seq analysis tools, and addresses the challenge in estimating overdispersion parameters with a limited number of replicates (Anders & Huber, *Genome Biol* 2010; Robinson & Smyth, *Bioinformatics* 2010). In the rare case of heavily replicated sequencing data, "The best analysis scheme for our data [726 individual RNA-Seq samples] was to first normalize using the DESeq method and apply a generalized linear model assuming a negative binomial distribution using either edgeR or DESeq software" (Lin & Harbison, *BMC Genomics* 2016).

Second, a linear modeling framework can directly estimate transcriptional and translational changes jointly without considering the "ratio of two NB variables" (Xiao & Yang, *Nat Commun.* 2016).

3. In light of these points, and given the existence of software packages that can perform NB GLM analysis of matched Ribo-seq and RNA-seq samples such as Xtail (cited above) and others, it seems worth comparing ribolog results with established tools.

4. Do Ribo-seq expression change measurements agree better with proteomics than RNA-seq? RNA-seq correlations ($R = 0.5$) and "logTER" correlations ($R = 0.3$) are reported, but not the uncorrected Ribo-seq, which should account for all mRNA changes and for translational changes as well, and thus correlate better with proteomics.

Referee 3:

Specifically, since many of the changes and differences presented in this manuscript are modest in magnitude/significance, it is essential to have sufficient confidence in the quality of the ribosome profiling data, and the validity of the method of analysis. In the absence of peer-review and benchmarking for the new software suite presented herein, it is critical to include PCA plots and p-value/FDR distributions for all data sets. Validation of the results using published benchmarked methods (RUST, RiboDiff, Anot2seq, etc.) would also be preferable, although the authors may wish to do this in a separate manuscript focusing on the Ribolog software package prior to publication of these findings.

1) Page 4, line 14-15

The authors claim that their new method Ribolog has advantages to existing methods. This may be the case, however, their method has not undergone peer-review or systematic benchmarking. For this reason, it is not possible to determine if it performs better or worse than the existing standards. Such comparisons, or a paper that does this, should be included to support the method.

7) Page 19, TER test

If it is understood correctly, the estimation of translation efficiency is based on an odds ratio obtained by pooling all reads (both RFP and RNA) for a given gene, and then sampling reads from this pool. If this is the case, then there seems to be a high danger of masking of outliers/outlier dominance using this method, since all biological variability is lost. For example, if reads from one replicate are considerably higher for a particular gene than in others, this information is lost when pooling but will of course still affect the rate of sampling. Furthermore, since it is not possible to determine outliers with only 2 biological replicates, the potential for false positives seems very high.

In terms of the quality of the ribosome profiling datasets, this is not adequately address. In Fig 2e there are a very large number of genes that have large fold changes but do not reach any threshold of significance, even considering p-values and not FDRs were used. This would suggest that there are substantial issues with the reproducibility of the data. In light of this, it is essential to provide PCA plots for all the various data sets presented herein, as well as p-value and FDR distributions for the outputs of all Ribolog TER test analyses.

The Ribolog method is not the topic of the paper. However, it is difficult to claim its superiority over other published methods that have undergone peer-review and systematic benchmarking. In the absence of peer-review and benchmarking for Ribolog, it would be advisable to also demonstrate the quality of the data presented here using an established tool like RUST (O'Connor 2016). In addition, confidence in the validity of the results could be increased if similar results were obtained using published methods such as RiboDiff (Zhong 2017) or Anot2seq (Oertlin 2019).

b) strengthen the link of the components of the working model to the in vivo cancer metastatic phenotype, as noted by:

Referee 1:

5. The authors should overexpress hnRNPC in MDA-LM2 and then check its metastatic potentials.

4. PDLIM5 is shown to be the downstream of HNRNPC to suppress metastasis. The authors did PDLIM5 knockdown in MDA-MB-231. Ideally, additional overexpression of PDLIM5 in MDA-LM2 could provide more definitive conclusions. In any case, the gene structure, including 3'UTR length and binding sites for hnRNPC, and its APA regulation are poorly presented in the figures.

Referee 3:

14) PDX models should be characterized to a greater extent

In the PDX samples (shown in 1b) there are far more transcripts with reduced translational machinery in the metastatic versus poorly metastatic tumours. This suggests a phenomenon, such as hypoxia, may be associated with the metastatic tumours. These PDX models should thus be better characterized before suggesting that HNRNPC is driving differential effects in these models. Indeed, associations seem much more modest in these models.

15) Models used do not fully test metastasis

The tail vein model is not a metastasis assay, but rather measures extravasation and survival and growth in the lung. The authors, should thus employ spontaneous metastasis models to validate their hypothesis. Given the availability of a PDX models, these should be used throughout, particularly for T4 studies. These latter studies should also be accompanied by a time course to better understand the kinetics of translational change and the extent to which these persist 6 hours after exposure and beyond.

16) Work related to PDLIM5 must be greatly improved

Work related to PDLIM5 (from protein expression to results in animals) is not that convincing. Different translational efficiencies of the isoforms should be determined and mechanisms by which PDLIM5 may affect metastasis should be determined. Moreover, the authors would need to determine the extent to which PDLIM5 overexpression may reverse the effects of HNRNPC knock down. Finally, linkages with AGO2/miRNA should be assessed with cause-effect experimentation.

c) address the concerns regarding the modest correlation and exclude alternative working models, as noted by:

Referee 1:

Also, the possibility of nuclear export regulation was not considered. This is disappointing, especially given that hnRNP C is a nuclear protein and PABPC4 is a shuttling protein. Note that in general transcripts with long 3'UTRs are more likely to be retained in the nucleus, which could lead to low ribo-seq values. In addition, western blot should be used to confirm some ribo-seq data.

Referee 2:

Statistically significant but quite modest correlations underlie the path from HNRNPC binding through alternative 3' end isoforms to Ago2-mediated repression, leaving open the possibility of alternative or indirect connections.

6. It is claimed that, "HNRNPC controls the alternative polyadenylation of its targets" (p. 5, l. 43). However, it would seem that an alternative explanation could be provided by nuclear export, which controls apparent TE (since the nuclear pool of an RNA is untranslated but shows up in mRNA-Seq) and can certainly be influenced by 3' UTRs and by nuclear RBPs. How is it excluded that HNRNPC promotes nuclear export and subsequent degradation of long-3'UTR variants?

Referee 3:

3) Page 5, lines 25-27, Figure S2b

Perhaps the figure has been misunderstood, however, the correlation coefficients between translationally repressed mRNAs in MDA-LM2 and HNRNPC KD cells appear very low, which seems to suggest that the data are actually quite poorly correlated. Thinking in terms of variance explained, with a coefficient of 0.05, this would be 0.25%. This is either quite unconvincing, or overly confusing. If the later, it may be helpful to visualize the overlap by overlaying the significantly translationally suppressed mRNAs from one cell line onto the volcano plot for the other by coloring these points.

d) All other referee concerns pertaining to strengthening existing data, providing controls, methodological details, clarifications and textual changes, should also be addressed.

e) Finally please pay close attention to our guidelines on statistical and methodological reporting (listed below) as failure to do so may delay the reconsideration of the revised manuscript. In particular please provide:

- a Supplementary Table including all numerical source data in Excel format, with data for different figures provided as different sheets within a single Excel file. The file should include source data giving rise to graphical representations and statistical descriptions in the paper and for all instances where the figures

present representative experiments of multiple independent repeats, the source data of all repeats should be provided.

We would be happy to consider a revised manuscript that would satisfactorily address these points, unless a similar paper is published elsewhere, or is accepted for publication in Nature Cell Biology in the meantime.

- ensure that it conforms to our format instructions and publication policies (see below and www.nature.com/nature/authors/).
- provide a point-by-point rebuttal to the full referee reports verbatim, as provided at the end of this letter.
- provide the completed Editorial Policy Checklist (found here <https://www.nature.com/authors/policies/Policy.pdf>), and Reporting Summary (found here <https://www.nature.com/authors/policies/ReportingSummary.pdf>). This is essential for reconsideration of the manuscript and these documents will be available to editors and referees in the event of peer review. For more information see <http://www.nature.com/authors/policies/availability.html> or contact me.

Nature Cell Biology is committed to improving transparency in authorship. As part of our efforts in this direction, we are now requesting that all authors identified as 'corresponding author' on published papers create and link their Open Researcher and Contributor Identifier (ORCID) with their account on the Manuscript Tracking System (MTS), prior to acceptance. ORCID helps the scientific community achieve unambiguous attribution of all scholarly contributions. You can create and link your ORCID from the home page of the MTS by clicking on 'Modify my Springer Nature account'. For more information please visit www.springernature.com/orcid.

[REDACTED]

We would like to receive a revised submission within six months. We would be happy to consider a revision even after this timeframe, however if the resubmission deadline is missed and the paper is eventually published, the submission date will be the date when the revised manuscript was received.

We hope that you will find our referees' comments, and editorial guidance helpful. Please do not hesitate to contact me if there is anything you would like to discuss.

Best wishes,

Jie Wang

Jie Wang, PhD
Senior Editor
Nature Cell Biology

Tel: +44 (0) 207 843 4924
email: jie.wang@nature.com

Reviewers' Comments:

Reviewer #1:

Remarks to the Author:

This manuscript from Navickas et al. reports a function of hnRNPC in regulation of alternative polyadenylation, its interaction with PABPC4, and their relevance to translational regulation in cancer cell metastasis. The authors used a battery of techniques, such as ribo-seq, 3' end seq, CLIP-seq, TMT-MS, CoIP-MS and PDX models, to substantiate their claims. The results largely support their conclusions. However, some critical pieces are missing.

Major:

1. The enrichment of U-rich motifs for translationally repressed transcripts in cells with higher metastatic potentials is quite interesting. This should be validated using a reporter assay. Ideally, regulation of APA, such as in the case of PDLIM5, should also be validated using a reporter assay. The reporter assay can be carried out in MDA-MB-231 vs. MDA-LM2, or one of the cells with knockdown or overexpression of hnRNPC.
2. The authors appear to have used total RNA to normalize ribo-seq data. This approach would miss regulation of nuclear export. They should use cytoplasmic RNA for normalization. Also, the possibility of

nuclear export regulation was not considered. This is disappointing, especially given that hnRNPC is a nuclear protein and PABPC4 is a shuttling protein. Note that in general transcripts with long 3'UTRs are more likely to be retained in the nucleus, which could lead to low ribo-seq values. In addition, western blot should be used to confirm some ribo-seq data.

3. Previous studies have shown PABPC1 knockdown could lead to 3'UTR shortening (Li et al. PLoS Genetics, 2015). The authors should check if PABPC4 knockdown would also lead to PABPC1 downregulation, as they often depend on each other. In addition, the lack of APA regulation in double knockdown of PABPC4 and hnRNPC is intriguing. The authors have not provided a satisfactory answer.

4. The authors should show their 3' sequencing data using genome tracks. The current figures show global patterns, giving nebulous impressions.

5. The authors should overexpress hnRNPC in MDA-LM2 and then check its metastatic potentials.

Minor:

1. In the figure 2c and 2d, the majority of hnRNPC targets contain PASs in the 3' UTR and undergo 3' UTR lengthening in MDA-LM2 compared to MDA-MB-231. Does this subset of HNRNPC targets have functions in cell migration and metastasis? The authors should carry out Gene Ontology analysis on these.

2. The authors seem to indicate that cells with high metastatic potentials express longer 3'UTRs. Given the previously reported connection between 3'UTR size and cell proliferation, could the authors comment on how cell proliferation may be involved in their model?

3. Many hnRNPC-bound targets are also AGO2 targets. U-rich motifs may make miRNA target sites more exposed to AGO2. Do they see miRNA target sites being close to hnRNPC binding sites?

4. PDLIM5 is shown to be the downstream of HNRNPC to suppress metastasis. The authors did PDLIM5 knockdown in MDA-MB-231. Ideally, additional overexpression of PDLIM5 in MDA-LM2 could provide more definitive conclusions. In any case, the gene structure, including 3'UTR length and binding sites for hnRNPC, and its APA regulation are poorly presented in the figures.

5. In supplementary figure 1f, scale bars should be included.

6. Page 4, line 26, 'RNA levels only partially explain the changing in protein levels ($R = 0.5$, $p < 2 \times 10^{-16}$). Which figure is this referring to?

7. In supplementary figure 3, genes regulated by PABPC4 have a higher potential to be HNRNPC targets. Gene ontology results should be presented for those genes. In addition, validation should be carried out for some top hits, using qPCR for example.

8. Figure S5a, the plot shown on the left does not seem to agree with bioluminescence pictures shown on the right.

Reviewer #2:

Remarks to the Author:

This manuscript argues that a metastatic gene expression program results from a shift towards longer 3' UTRs and weaker translation in the affected genes. This shift is driven by lower levels of HNRNPC and PABPC4. The pro-metastatic gene expression program is defined first through Ribo-seq on cell lines representing primary and metastatic tumors. A uridine-rich motif is enriched in transcripts with lower translation under metastatic conditions. Analysis of these targets leads to the nomination of HNRNPC as an RNA-binding protein that recognizes this motif. The association of these HNRNPC binding sites with cleavage and polyadenylation signals motivates 3' end sequencing experiments, which show a change in alternative polyadenylation that is correlated with HNRNPC binding. An RNA-dependent interaction between HNRNPC and the poly-(A) binding protein PABPC4 leads to the observation that PABPC4 knock-down produces 3' end shifts that overlap significantly with those seen in HNRNPC knock-down, and further, a genetic interaction between these two knock-down conditions. This is interpreted to argue that HNRNPC and PABPC4 work together to modulate isoform differences. The reduced translation, then, is argued to result from additional microRNA binding sites introduced into the longer 3' UTRs seen in metastatic cells and in HNRNPC depletion.

The manuscript goes on to argue that HNRNPC restrains metastasis. Knock-down of HNRNPC accelerates engraftment of tumor xenograft metastases, and lower levels of HNRNPC or PABPC4 correlate with worse prognosis in human breast cancers as well. The gene PDLIM5 is identified as a particular target of 3' UTR extension, and it also appears to restrain metastasis in mouse models and correlate with improved prognosis in patients. Notably, a small molecule drug previously found to shift polyadenylation sites can in part reverse the 3' UTR lengthening seen in HNRNPC knock-down.

This study reports a range of observations about RNA processing in primary and metastatic tumors, tying them to effects seen in cancer models. The reported molecular results — translational changes, 3' UTR extension, and changes in HNRNPC — all seem individually well supported, but the causal link between them seems much weaker, though. Statistically significant but quite modest correlations underlie the path from HNRNPC binding through alternative 3' end isoforms to Ago2-mediated repression, leaving open the possibility of alternative or indirect connections.

My specific concerns are

1. In discussing the new "ribolog" tool for Ribo-seq analysis, it is claimed that it has "as few a priori assumptions as possible" and "does not assume a negative binomial (NB) distribution of read counts" (p. 4, ll. 8 - 18). However, logistic regression implicitly assumes a standard binomial error model, which is strictly no more conservative than the NB read count model used in many RNA-Seq tools, and in practice will be less conservative. In particular, the NB models incorporates an overdispersion term, while ribolog effectively assumes that this term is zero. Lower variance estimates then lead to stronger statistical significances.

2. As a related point, the discussion of the Ribolog method (p. 17) mentions problems that arise in Ribo-seq data analysis but does not credit the fact that existing tools offer clear solutions to these problems.

First, "variance sharing" underlies many Ribo-seq and RNA-seq analysis tools, and addresses the challenge in estimating overdispersion parameters with a limited number of replicates (Anders & Huber, *Genome Biol* 2010; Robinson & Smyth, *Bioinformatics* 2010). In the rare case of heavily replicated sequencing data, "The best analysis scheme for our data [726 individual RNA-Seq samples] was to first normalize using the DESeq method and apply a generalized linear model assuming a negative binomial distribution using either edgeR or DESeq software" (Lin & Harbison, *BMC Genomics* 2016).

Second, a linear modeling framework can directly estimate transcriptional and translational changes jointly without considering the "ratio of two NB variables" (Xiao & Yang, *Nat Commun.* 2016).

3. In light of these points, and given the existence of software packages that can perform NB GLM analysis of matched Ribo-seq and RNA-seq samples such as Xtail (cited above) and others, it seems worth comparing ribolog results with established tools.

4. Do Ribo-seq expression change measurements agree better with proteomics than RNA-seq? RNA-seq correlations ($R = 0.5$) and "logTER" correlations ($R = 0.3$) are reported, but not the uncorrected Ribo-seq, which should account for all mRNA changes and for translational changes as well, and thus correlate better with proteomics.

5. Several other RBPs — including TIA1 and ELAVL1/HuR — seem to have almost as much mutual information as HNRNPC (Supplementary Figure 1g).

6. It is claimed that, "HNRNPC controls the alternative polyadenylation of its targets" (p. 5, l. 43). However, it would seem that an alternative explanation could be provided by nuclear export, which controls apparent TE (since the nuclear pool of an RNA is untranslated but shows up in mRNA-Seq) and can certainly be influenced by 3' UTRs and by nuclear RBPs. How is it excluded that HNRNPC promotes nuclear export and subsequent degradation of long-3'UTR variants?

6. HNRNPC and PABPC4 are both pretty well-expressed and broad mRNA-binding proteins, and so an RNA-dependent interaction between them (Figure 3) is not surprising.

7. In the comparison of PABPC4 binding and HNRNPC binding (Figure 3c), what is the total universe of 3' UTRs assessed for binding with both proteins?

8. In Figure 4a, it seems that almost all genes (~16,456 / 17,857 I think?) show AGO2 CLIP? Is the conclusion that >90% of mRNAs experience microRNA-mediated repression?

9. Is the accelerated growth of HNRNPC (Figure 5) and PABPC4 (Supplementary Figure 5) knock-down cells specific to the lung metastasis environment, or do these knock-downs increase growth in culture or in localized xenografts as well?

10. How many transcripts matched the four criteria used to select PDLIM5 (Figure 6a)?

11. As a more minor point, "SH3" is "Src homology 3" not "Scr homology 3" (p. 8, l. 25).

Reviewer #3:

Remarks to the Author:

SUMMARY:

Using ribosome profiling, the authors have performed a transcriptome-wide analysis of translation in parental MDA-231 cells and their metastatic derivatives, as well as PDXs. They identify a subset of mRNAs that are translationally suppressed and harbor binding of HNRNPC and PABPC4, which together coordinate alternative polyadenylation of these transcripts. In metastatic cells, the RBPs are downregulated leading to the selection of downstream polyA signals and polyA lengthening leading to translational repression that is AGO2-dependent. Furthermore, KD of HNRNPC enhances metastasis in vivo, and loss of HNRNPC is associated with poor patient survival. The authors also introduce their new logistic-regression-based methods for analysis of ribosome profiling and alternative polyadenylation data, Ribolog and APALog.

Roles for HNRNPC and PABPC4 in alternative polyadenylation and metastasis have been explored previously. However, the particular mechanism of translational control described herein is novel. However, the rigor of the informatic approaches should be improved, and the metastasis assays should be refined.

Specifically, since many of the changes and differences presented in this manuscript are modest in magnitude/significance, it is essential to have sufficient confidence in the quality of the ribosome profiling data, and the validity of the method of analysis. In the absence of peer-review and benchmarking for the new software suite presented herein, it is critical to include PCA plots and p-value/FDR distributions for all data sets. Validation of the results using published benchmarked methods (RUST, RiboDiff, Anota2seq, etc.) would also be preferable, although the authors may wish to do this in a separate manuscript focusing on the Ribolog software package prior to publication of these findings. Finally, the functional assays should actually look more specifically at metastasis, and the mechanistic links with PDLIM5 must be refined and better validated.

COMMENTS AND SUGGESTED IMPROVEMENTS

1) Page 4, line 14-15

The authors claim that their new method Ribolog has advantages to existing methods. This may be the case, however, their method has not undergone peer-review or systematic benchmarking. For this reason, it is not possible to determine if it performs better or worse than the existing standards. Such comparisons, or a paper that does this, should be included to support the method.

2) Page 5, lines 10-11, page 6 line 2-3

The RNAseq has only two replicates, so commenting on significance is likely not appropriate. Another replicate should be added. Moreover, the authors should show a dot plot to assess the variability between replicates.

3) Page 5, lines 25-27, Figure S2b

Perhaps the figure has been misunderstood, however, the correlation coefficients between translationally repressed mRNAs in MDA-LM2 and HNRNPC KD cells appear very low, which seems to suggest that the data are actually quite poorly correlated. Thinking in terms of variance explained, with a coefficient of 0.05, this would be 0.25%. This is either quite unconvincing, or overly confusing. If the later, it may be helpful to visualize the overlap by overlaying the significantly translationally suppressed mRNAs from one cell line onto the volcano plot for the other by coloring these points.

4) Page 7, Figure S4a-b

The subset of translationally suppressed HNRNPC targets that are not AGO2/miRNA targets are not shown here. This seems like an essential comparison if the claim is that the effect on translation is mediated by miRNA targeting.

5) Page 8, Figure S6b

Again, with an R of 0.140, this means that 1.96% of the variation is explained by HNRNPC activity. This suggests the effect may be for a relatively small subset of genes, but the wording here seems to imply that this is a more generalized effect.

6) Page 8, Figure 6a

The Venn diagram seems to be incomplete as the numbers of genes are not shown for the various groups. Additionally, the size of the circles should reflect the gene numbers. It is also not clear from the text what the “clinical association” dataset is, and where it came from.

7) Page 19, TER test

If it is understood correctly, the estimation of translation efficiency is based on an odds ratio obtained by pooling all reads (both RFP and RNA) for a given gene, and then sampling reads from this pool. If this is the case, then there seems to be a high danger of masking of outliers/outlier dominance using this method, since all biological variability is lost. For example, if reads from one replicate are considerably higher for a

particular gene than in others, this information is lost when pooling but will of course still affect the rate of sampling. Furthermore, since it is not possible to determine outliers with only 2 biological replicates, the potential for false positives seems very high.

In terms of the quality of the ribosome profiling datasets, this is not adequately address. In Fig 2e there are a very large number of genes that have large fold changes but do not reach any threshold of significance, even considering p-values and not FDRs were used. This would suggest that there are substantial issues with the reproducibility of the data. In light of this, it is essential to provide PCA plots for all the various data sets presented herein, as well as p-value and FDR distributions for the outputs of all Ribolog TER test analyses.

The Ribolog method is not the topic of the paper. However, it is difficult to claim its superiority over other published methods that have undergone peer-review and systematic benchmarking. In the absence of peer-review and benchmarking for Ribolog, it would be advisable to also demonstrate the quality of the data presented here using an established tool like RUST (O'Connor 2016). In addition, confidence in the validity of the results could be increased if similar results were obtained using published methods such as RiboDiff (Zhong 2017) or Anota2seq (Oertlin 2019).

8) Volcano plots

Due to multiple-testing, the y-axis on all volcano plots presented should reflect the FDR rather than the p-value.

In general, coloring the HNRNPC targets, or mRNA in the proposed HNRNPC/PABPC4 regulon on the scatter plots would be helpful in seeing the specificity of the effect. As the data is currently presented, it is not possible to see how the same gene is regulated between two plots. The interpretability of the results would be greatly improved by being able to see how this same set of genes behaves across the different KD conditions.

9) CRISPRi

A number of CRISPRi KDs were generated in this study, but no validation was provided to confirm that the KDs were successful. This should at least be shown in a supplement.

10) Motifs

Although scrambled sequences are used by some, even with maintenance of kmers this method is more prone to underestimating the rate of false positives than using the true sequences for a background set.

11) Fig 1d

MI in bits and z-score are likely not very meaningful metrics for most. Fig S1h is more compelling, and a similar cdf plot should be generated for the PDXs.

12) Fig S3a

P-values are presented for the enrichment analyses. An adjusted p-value or FDRs is appropriate to report here. Furthermore, the accompanying heatmap seems misleading considering the lack of significance for the mRNA export from the nucleus GO term.

13) Fig S6e-f

Quantifying all biological replicates from the same gel somewhat defeats the purpose of replicates in terms of controlling for assay variability. It also looks like there may have been a technical issue with the transfer for this membrane.

14) PDX models should be characterized to a greater extent

In the PDX samples (shown in 1b) there are far more transcripts with reduced translational machinery in the metastatic versus poorly metastatic tumours. This suggests a phenomenon, such as hypoxia, may be associated with the metastatic tumours. These PDX models should thus be better characterized before suggesting that HNRNPC is driving differential effects in these models. Indeed, associations seem much more modest in these models.

15) Models used do not fully test metastasis

The tail vein model is not a metastasis assay, but rather measures extravasation and survival and growth in the lung. The authors, should thus employ spontaneous metastasis models to validate their hypothesis. Given the availability of a PDX models, these should be used throughout, particularly for T4 studies. These latter studies should also be accompanied by a time course to better understand the kinetics of translational change and the extent to which these persist 6 hours after exposure and beyond.

16) Work related to PDLIM5 must be greatly improved

Work related to PDLIM5 (from protein expression to results in animals) is not that convincing. Different translational efficiencies of the isoforms should be determined and mechanisms by which PDLIM5 may affect metastasis should be determined. Moreover, the authors would need to determine the extent to which PDLIM5 overexpression may reverse the effects of HNRNPC knock down. Finally, linkages with AGO2/miRNA should be assessed with cause-effect experimentation.

Methods should be written concisely, but should contain all elements necessary to allow interpretation and replication of the results. As a guideline, Methods sections typically do not exceed 3,000 words. The Methods should be divided into subsections listing reagents and techniques. When citing previous methods, accurate references should be provided and any alterations should be noted. Information must be provided about: antibody dilutions, company names, catalogue numbers and clone numbers for monoclonal antibodies; sequences of RNAi and cDNA probes/primers or company names and catalogue numbers if reagents are commercial; cell line names, sources and information on cell line identity and authentication. Animal studies and experiments involving human subjects must be reported in detail, identifying the committees approving the protocols. For studies involving human subjects/samples, a statement must be included confirming that informed consent was obtained. Statistical analyses and information on the reproducibility of experimental results should be provided in a section titled “Statistics and Reproducibility”.

All Nature Cell Biology manuscripts submitted on or after March 21 2016 must include a Data availability statement at the end of the Methods section. For Springer Nature policies on data availability see <http://www.nature.com/authors/policies/availability.html>; for more information on this particular policy see <http://www.nature.com/authors/policies/data/data-availability-statements-data-citations.pdf>. The Data availability statement should include:

- Accession codes for primary datasets (generated during the study under consideration and designated as "primary accessions") and secondary datasets (published datasets reanalysed during the study under consideration, designated as "referenced accessions"). For primary accessions data should be made public to coincide with publication of the manuscript. A list of data types for which submission to community-endorsed public repositories is mandated (including sequence, structure, microarray, deep sequencing data) can be found here <http://www.nature.com/authors/policies/availability.html#data>.
- Unique identifiers (accession codes, DOIs or other unique persistent identifier) and hyperlinks for datasets deposited in an approved repository, but for which data deposition is not mandated (see here for details <http://www.nature.com/sdata/data-policies/repositories>).
- At a minimum, please include a statement confirming that all relevant data are available from the authors, and/or are included with the manuscript (e.g. as source data or supplementary information), listing which data are included (e.g. by figure panels and data types) and mentioning any restrictions on availability.
- If a dataset has a Digital Object Identifier (DOI) as its unique identifier, we strongly encourage including this in the Reference list and citing the dataset in the Methods.

We recommend that you upload the step-by-step protocols used in this manuscript to the Protocol Exchange. More details can found at www.nature.com/protocolexchange/about.

All imaging data should be accompanied by scale bars, which should be defined in the legend. Cropped images of gels/blots are acceptable, but need to be accompanied by size markers, and to retain visible background signal within the linear range (i.e. should not be saturated). The boundaries of panels with low background have to be demarked with black lines. Splicing of panels should only be considered if unavoidable, and must be clearly marked on the figure, and noted in the legend with a statement on whether the samples were obtained and processed simultaneously. Quantitative comparisons between

samples on different gels/blots are discouraged; if this is unavoidable, it should only be performed for samples derived from the same experiment with gels/blots were processed in parallel, which needs to be stated in the legend.

The total number of Supplementary Figures (not including the “unprocessed scans” Supplementary Figure) should not exceed the number of main display items (figures and/or tables (see our Guide to Authors and March 2012 editorial <http://www.nature.com/ncb/authors/submit/index.html#suppinfo>;

<http://www.nature.com/ncb/journal/v14/n3/index.html#ed>). No restrictions apply to Supplementary Tables or Videos, but we advise authors to be selective in including supplemental data.

GUIDELINES FOR EXPERIMENTAL AND STATISTICAL REPORTING

REPORTING REQUIREMENTS – To improve the quality of methods and statistics reporting in our papers we have recently revised the reporting checklist we introduced in 2013. We are now asking all life sciences authors to complete two items: an Editorial Policy Checklist (found here <https://www.nature.com/authors/policies/Policy.pdf>) that verifies compliance with all required editorial policies and a reporting summary (found here <https://www.nature.com/authors/policies/ReportingSummary.pdf>) that collects information on experimental design and reagents. These documents are available to referees to aid the evaluation of the manuscript. Please note that these forms are dynamic ‘smart pdfs’ and must therefore be downloaded and completed in Adobe Reader. We will then flatten them for ease of use by the reviewers. If you would like to reference the guidance text as you complete the template, please access these flattened versions at <http://www.nature.com/authors/policies/availability.html>.

Author Rebuttal to Initial comments

A) address the concerns regarding the Ribo-seq data, as noted by:

Referee Comment A.1:

2. The authors appear to have used total RNA to normalize ribo-seq data. This approach would miss regulation of nuclear export. They should use cytoplasmic RNA for normalization.

Response to Comment A.1:

As you will see below, we have explicitly addressed this point raised by the reviewer through multiple independent approaches. However, we should note that we had previously studied RNA export in these models of metastasis¹, and we had not observed a signal for differential export of the HNRNPC-bound RNAs.

In our initial manuscript, we performed ribosome profiling from the cell lysates prepared with 1% (v/v) Triton X-100 as detergent². We used the same lysate to perform nuclease footprinting for Ribo-seq or extract "total" (i.e. not RNase1-treated) RNA for RNA-seq. That is, we used clarified lysates for RNA extraction and not whole cell lysates.

To determine if our lysates contained nuclear fraction, we performed subcellular fractionation using our initial lysis buffer with 1% (v/v) Triton X-100 as detergent, or 0.15% Igepal CA-630 (NP40 alternative), as described³. We collected the cytoplasmic fraction as a soluble lysate, and nuclei as a pellet centrifuged through sucrose cushion. We prepared nuclear lysates by solubilizing the pellet in RIPA buffer. We then saved half of the lysates for protein analysis, and used the other half for RNA extraction. As shown in **Fig. R1a** and S1g in the revised manuscript, in both lysis buffers, we detected mature HPRT mRNA mostly in the cytoplasmic fraction, while

the intron-containing HPRT pre-mRNA, as well as MALAT1 and NEAT1 ncRNAs (known to be retained in the nucleus), were predominantly nuclear. At the protein level, as shown in **Fig. R1b**, we detected HNRNPC and WTAP predominantly in the nuclear fraction, and GAPDH and beta-tubulin mostly in the cytoplasmic fraction as expected, similarly in both lysis buffers.

Figure R1. (a) Relative proportions of various RNAs in nuclear (orange) and cytoplasmic (blue) fractions, extracted using 1% Triton X-100 (left panel) or 0.15% Igepal CA-630 (right panel) detergents, as determined by RTqPCR. **(b)** Western blot images detecting the proteins in subcellular fractions as described in (a).

Next, we repeated the Ribo-seq experiments, but with the modified lysis buffer, substituting Triton X-100 for 0.15% Igepal CA-630. We performed the experiments in biological triplicates, in MDA-MB-231 and MDA-LM2, as well as control and HNRNPC KD cells. We used cytoplasmic lysates both for nuclease foot printing and undigested RNA extraction. From the same preparations, we also kept nuclei pellets and extracted nuclear RNA. As shown in **Fig. R2** and **Fig. S3l-m** in the revised manuscript, our results from the cytoplasmic fraction recapitulated our initial findings: HNRNPC target RNAs were translationally repressed in highly metastatic or HNRNPC-depleted cells.

Figure R2. (a) Bottom: Volcano plot showing the distribution of changes in translation efficiency ratio (logTER) in highly metastatic MDA-LM2 cells compared to parental MDA-MB-231 cells. Statistically significant (logistic regression, $p < 0.01$) observations are highlighted in orange. Top: Enrichment of the HNRNPC targets as a function of logTER between sgHNRNPC and sgControl cells. mRNAs are divided into equally populated bins according to logTER (dotted vertical lines delineate the bins); the y-axis shows the frequency of the HNRNPC targets that we identified in each bin (dotted horizontal line denotes the average HNRNPC target frequency across all transcripts). Bins with significant enrichment (logistic regression, FDR < 0.05; red) or depletion (blue) of HNRNPC targets are denoted with a black border. Also included are mutual information (MI) values and their associated z-scores. (b) Comparison between sgHNRNPC and sgControl MDA-MB-231 cells, otherwise as in (a).

We also repeated 3'-end RNA-seq experiments aimed at poly(A) site mapping with nuclear and cytoplasmic RNA from MDA-MB-231 and MDA-LM2, as well as control and HNRNPC KD cells, in biological triplicates. Just as with Ribo-seq experiments described above, changes in alternative polyadenylation in the cytoplasmic fraction recapitulated our initial findings: HNRNPC target RNAs underwent 3' UTR lengthening in highly metastatic or HNRNPC-depleted cells (Fig. R3a-b and Fig. S3k-j in the revised manuscript). Moreover, as shown in cumulative density plots in Fig. R3c-d and Fig. S3h-i in the revised manuscript, we detected no significant difference between HNRNPC target and non-target RNAs when comparing nuclear and cytoplasmic fractions for alternative polyadenylation (logAPAR) changes in highly metastatic or

HNRNPC-depleted cells.

Figure R3. (a) Bottom: Cumulative density plot of alternative polyadenylation ratios (logAPAR) comparing MDA-LM2 to MDA-MB-231 cells, for HNRNPC 3' UTR target and non-target mRNAs. Median difference (ΔM) and p value (calculated using Mann-Whitney U -test) are shown. Middle: Volcano plot showing distribution of changes in logAPAR in MDA-LM2 compared to MDA-MB-231 cells. Top: Enrichment of the HNRNPC-bound 3' UTRs as a function of logAPAR between MDA-LM2 and MDA-MB-231

cells; statistics as in Figure R2a. **(b)** Comparison between sgHNRNPC and sgControl MDA-MB-231 cells, otherwise as in (a). **(c)** Cumulative density plot of log fold-change between nuclear and cytoplasmic fractions, comparing logAPAR between MDA-LM2 and MDA-MB-231 cells, for HNRNPC 3' UTR target and non-target mRNAs; statistics as in (a). **(d)** Comparison between sgHNRNPC and sgControl MDA-MB-231 cells, otherwise as in (c).

In sum, this data suggests that HNRNPC-dependent regulation of poly(A) site choice and the resulting translational consequences is not contingent on the regulation of nuclear export.

Referee Comment A.2:

In discussing the new "ribolog" tool for Ribo-seq analysis, it is claimed that it has "as few a priori assumptions as possible" and "does not assume a negative binomial (NB) distribution of read counts" (p. 4, ll. 8 - 18). However, logistic regression implicitly assumes a standard binomial error model, which is strictly no more conservative than the NB read count model used in many RNA-Seq tools, and in practice will be less conservative. In particular, the NB models incorporates an overdispersion term, while ribolog effectively assumes that this term is zero. Lower variance estimates then lead to stronger statistical significances.

Response to comment A.2:

While we have addressed the reviewers' comments around our Ribolog analytical suite, we should emphasize that the key insights derived from our Ribo-seq data is based on gene-set enrichment analysis of differential translation efficiency values. In other words, these analyses are not sensitive to the choice of statistical test, as they do not rely on p-values but rather the ranking of genes based on their logTERs. To demonstrate this, we have included the comparison of logTER values for alternatively poly-adenylated genes that are bound or unbound

by HNRNPC (Fig. S1i and S2a in the revised manuscript) with Xtail, Riborex, and RiboDiff as well (Fig. R4).

Figure R4. Cumulative density plot of translation efficiency ratios (logTER) comparing MDA-LM2 to MDA-MB-231 (Mda-Par, top) or sgHNRNPC to sgControl MDA-MB-231 (bottom) cells, for HNRNPC 3' UTR target and non-target mRNAs, using Xtail (left), Riborex (center) or RiboDiff (right). Median difference (ΔM) and p value (calculated using Mann-Whitney U -test) are shown.

Second, the aforementioned similarity of results is largely expected because the main determinants of logTER in all these tools are simply the normalized RPF and RNA counts. As an example, we have included the pair-wise Pearson correlation matrix of logTER values calculated using Ribolog, Xtail, RiboDiff, and Riborex. These results, which were generated on simulated counts from the Xtail package, show almost identical logTER values across tools except for Anota2seq (Table R1).

	Ribolog	Xtail	Riborex	RiboDiff
Ribolog		0.999	0.998	1.000
Xtail	0.999		1.000	0.999
Riborex	0.998	1.000		0.998
RiboDiff	1.000	0.999	0.998	

Table R1. Pairwise Pearson correlation matrix of logTER values generated on simulated counts from the Xtail package, obtained with Ribolog, Xtail, Riborex, and RiboDiff.

Third, unlike the above-mentioned tools, Ribolog allows for correction of local heterogeneity (stalling) biases in Ribo-seq data by implementing a CELP module. Both simulated data and real-world examples indicate that this correction improves logTER estimates. We found that after applying the CELP correction, residuals calculated as the difference between Ribolog and Xtail logTER values are significantly correlated with changes in protein levels (Fig. R5). This

observation establishes that starting from the same data, Ribolog captures the protein landscape better.

Figure R5. Regression analysis comparing the protein logFC in MDA-LM2 versus MDA-MB-231 cells to the residual logTER from the values calculated using Xtail subtracted from those calculated using Ribolog. Correlation coefficient and the resulting p -value are shown.

Finally, we have made the results of our benchmarking tests available online at https://github.com/goodarzilab/Ribolog/blob/master/benchmarks/ribolog_benchmarks.pdf. These benchmarks establish Ribolog as a strong contender with (i) the highest ROC- and PR-AUC, (ii) one of the lowest CPU times, and (iii) incredibly robust to changes in sequencing depth.

Referee Comment A.3

As a related point, the discussion of the Ribolog method (p. 17) mentions problems that arise in Ribo-seq data analysis but does not credit the fact that existing tools offer clear solutions to these problems.

First, "variance sharing" underlies many Ribo-seq and RNA-seq analysis tools, and addresses the challenge in estimating overdispersion parameters with a limited number of replicates (Anders & Huber, *Genome Biol* 2010; Robinson & Smyth, *Bioinformatics* 2010). In the rare case of heavily replicated sequencing data, "The best analysis scheme for our data [726 individual RNA-Seq samples] was to first normalize using the DESeq method and apply a generalized linear model assuming a negative binomial distribution using either edgeR or DESeq software" (Lin & Harbison, *BMC Genomics* 2016).

Response to comment A.3

As mentioned above, our benchmarking document compares the performance of multiple tools on simulated counts (from the original Xtail publication) and how they compare with Ribolog. And it is clear that Ribolog surpasses existing tools in a number of important metrics. With regards to this comment, the mere existence of proposed solutions does not imply that these

solutions have been rigorously tested in every case and their use is broadly advisable in every situation. The reviewer is correct in their assertion that several solutions have been proposed for variance estimation in count data, including the Bayesian shrinkage method. However, the fact that at low replicate counts of Ribo-seq datasets (often at 2 or 3 per condition) these estimates are far from perfect is not disputed. Furthermore, shrinkage and other variance sharing methods implicitly assume the same relationship between mean and variance for genes of the same expression level, which facilitates the estimation of dispersion parameter but explicitly adds an additional core assumption. Since highly replicated datasets have not been created for Ribo-seq, the veracity of this critical assumption has not been tested. For example, consider this quote from the developers of RiboDiff (available on their github page):
“Similar to many other RNA-Seq based tools, RiboDiff uses negative binomial distribution to

model the read count, which handles larger variation across samples than Poisson. However, if the randomness of count data from certain types of samples are extremely large, limited number of replicates cannot provide a good estimation on dispersion, which ends up with less significant results..." (<https://github.com/ratschlab/RiboDiff/blob/master/MANUAL>).

Negative Binomial has often performed better than the other commonly tested distributions. However, we believe that there may be other distributions that surpass NB. The most highly replicated RNA-seq dataset we are aware of was generated by Gierlinski *et al.*⁴, where a wild type and a *snf2Δ* yeast strain were sequenced *48 times* each. The authors showed that NB fitted the data better than normal or lognormal distributions. To showcase our point, we removed the bad replicates per instructions of the original paper and tested the goodness-of-fit of the data against a gamma distribution, and compared with NB. **Table R2** shows that gamma distribution fits this data better than NB. To be clear, we are not proposing TER testing based on gamma distribution, rather we are providing this observation in the RNA-seq world to highlight the notion that although NB-based methods have been generally successful in the past, alternative distributions and approaches can still be found to further improve our models of sequencing count data.

	Wild type		snf2Δ	
	Not rejected	Rejected	Not rejected	Rejected
Gamma (CVM)	4976	1113	5467	707
Gamma (AD)	4935	1154	5423	751
NB (Chi2)	4888	1202	5369	806

Table R2. Goodness-of-fit of Gamma and NB distributions for the 48-replicate yeast RNA-seq dataset. Values show the number of genes for which the null hypothesis of a given distribution is rejected (or not) based on the best fitted parameters and unadjusted p-values. CVM: Cramer-Von Mises test. AD: Anderson-Darling test. Chi2: Chi-square test.

Referee Comment A.4.

Second, a linear modeling framework can directly estimate transcriptional and translational changes jointly without considering the "ratio of two NB variables" (Xiao & Yang, Nat Commun. 2016).

In light of these points, and given the existence of software packages that can perform NB GLM analysis of matched Ribo-seq and RNA-seq samples such as Xtail (cited above) and others, it seems worth comparing Ribolog results with established tools.

Response to comment A.4.

As mentioned above, our biological results here are not sensitive to the choice of model and tool (**Fig. R4**). We have also provided access to our benchmarks that justify the choice of Ribolog as our current strategy for Ribo-seq data analysis.

To emphasize, the assumptions that the distribution parameters can be estimated effectively with 2 or 3 biological replicates is not a strong one. If the parameter estimates for a single NB distribution are imprecise, the ratio or joint distribution of two NB variables will be so too.

Our benchmarking results on the simulated dataset provided by the authors of Xtail show that Ribolog performs equally well in terms of ROC-, PR-AUC, and F1 measure. Yet Ribolog runs almost two orders of magnitude faster than Xtail and is much more robust to variations in

sequencing coverage

(https://github.com/goodarzilab/Ribolog/blob/master/benchmarks/ribolog_benchmarks.pdf).

Furthermore, Ribolog provides analytical modules for stalling bias detection and correction, meta-analysis, uORF usage and translational readthrough (please see <https://github.com/goodarzilab/Ribolog>). And, by implementing a regression-based model allows the accommodation of complex experimental designs with multiple predictors, interaction terms, and batch effect (confounding factor) adjustment variables. This makes Ribolog more flexible and more widely applicable than alternative methods such as Xtail, RiboDiff and Riborex – which basically compare two-groups⁵.

Referee Comment A.5

Do Ribo-seq expression change measurements agree better with proteomics than RNA-seq? RNA-seq correlations ($R = 0.5$) and "logTER" correlations ($R = 0.3$) are reported, but not the uncorrected Ribo-seq, which should account for all mRNA changes and for translational changes as well, and thus correlate better with proteomics.

Response to comment A.5

Overall, we agree with the reviewer that this is the expected behavior. With important caveats that proteomics data is skewed towards more abundant proteins and is strongly confounded protein decay changes, which is not captured in Ribo-seq. To answer this question, we used DESeq2 to compare log fold changes for RPF and RNA-seq data separately. Using logFC from RPF counts, we observed a correlation coefficient of $R = 0.53$ (CI = [0.51,0.55]) with mass-spec logFC. In comparison, we observed a correlation coefficient of $R = 0.46$ (CI = [0.43,0.48]) for RNA-seq data. In other words, RPF data captures an additional 7% of variation in proteomic measurements. Consistent with this finding, Duncan and Mata⁶ reported a moderately stronger correlation of protein levels with RPF ($R = 0.82$) than with RNA ($R = 0.68$) counts. Finally, in our initial manuscript we also mention that protein abundance measurements by mass spectrometry corrected my RNA-seq data and logTER values (i.e. RPF sequencing measurements corrected by RNA-seq data) correlate well ($R = 0.3$).

Referee Comment A.6:

Specifically, since many of the changes and differences presented in this manuscript are modest in magnitude/significance, it is essential to have sufficient confidence in the quality of the ribosome profiling data, and the validity of the method of analysis. In the absence of peer-review and benchmarking for the new software suite presented herein, it is critical to include PCA plots and p-value/FDR distributions for all data sets. Validation of the results using published benchmarked methods (RUST, RiboDiff, Anota2seq, etc.) would also be preferable, although the authors may wish to do this in a separate manuscript focusing on the Ribolog software package prior to publication of these findings.

Response to comment A.6

We agree that biological results should ideally be robust to the choice of justifiable analytical models. First, as was demonstrated above, our results remain largely identical across different tools. Below, we have included the comparison of logTER values for alternatively poly-adenylated genes that are bound or unbound by HNRNPC (**Fig. S1i and S2a** in the revised manuscript) with Xtail, Riborex, and RiboDiff as well (**Fig. R4**). Furthermore, while the median effect of post-transcriptional regulatory pathways on their individual targets is modest, the regulon effect observed here is highly stable, robust, and significant.

With regards to data quality, Ribolog in fact provides an extensive module devoted to QC. As included in the revised manuscript, we have now repeated the key experiments twice, which

resulted in identical results. We have also included the PCA plots for our Ribo-seq datasets below (**Fig. R6**).

Figure R6. (a) Principal component analysis (PCA) of Ribo-seq and RNA-seq data in biological triplicates, in MDA-MB-231 and MDA-LM2 cells. **(b)** PCA of Ribo-seq and RNA-seq data in biological triplicates, in sgControl or sgHNRNPC-expressing MDA-MB-231 cells.

Referee Comment A.7

The authors claim that their new method Ribolog has advantages to existing methods. This may be the case, however, their method has not undergone peer-review or systematic benchmarking. For this reason, it is not possible to determine if it performs better or worse than the existing standards. Such comparisons, or a paper that does this, should be included to support the method.

Response to comment A.7

Please refer to the previous comment for the response to this point. In sum, *(i)* we have clarified that our results are not sensitive to the choice analytical tools, *(ii)* we have released a benchmarking document for Ribolog that systematically compares a number of common Ribo-seq packages head-to-head, and *(iii)* some components of Ribolog have been previously published.

Referee Comment A.8

If it is understood correctly, the estimation of translation efficiency is based on an odds ratio obtained by pooling all reads (both RFP and RNA) for a given gene, and then sampling reads from this pool. If this is the case, then there seems to be a high danger of masking of

outliers/outlier dominance using this method, since all biological variability is lost. For example, if reads from one replicate are considerably higher for a particular gene than in others, this information is lost when pooling but will of course still affect the rate of sampling. Furthermore, since it is not possible to determine outliers with only 2 biological replicates, the potential for false positives seems very high.

Response to comment A.8

The reviewer's description of the Ribolog TER test is correct. Regarding outliers: They are a risk to any method, not just ours. Read-level QC and removing outliers should be done prior to TER analysis using any approach. There are several packages that provide tools for this purpose for example by checking the correlation of RNA and RPF reads across replicates and identifying the outliers; we have not duplicated those steps here. Instead, we provide an entire module with Ribo-seq-specific QC tools including PCA (on normalized RNA counts, normalized RPF counts and TE), proportion of null features (non-differentially translated transcripts) and correlograms of equivalent TER tests in replicated datasets. We have also re-done all Ribo-seq datasets in triplicates as well (**Fig. R6**).

In sum: (i) Outliers affect both count-based methods (e.g. Xtail) and read-based methods (e.g. Ribolog). If an outlier is not removed, it could inflate the within-group variance and produce a

false negative result in a count-based model. Our method may indeed be more robust to outliers [need benchmark to confirm] because the effect is mitigated via pooling with “good” replicates. (ii) During pre-processing, we normalize read counts (separately for RNA and RPF) using the median-of-ratios method⁷ which partially reduces outlier effects. (iii) If the outlier behavior manifests itself as unusually high local densities of RPF reads in some transcripts and samples – whether due to a biological reason or a technological artifact – the Ribolog CELP module mitigates the problem, even without the researcher having to explicitly specify the cause.

Referee Comment A.9

In terms of the quality of the ribosome profiling datasets, this is not adequately address. In Fig 2e there are a very large number of genes that have large fold changes but do not reach any threshold of significance, even considering p-values and not FDRs were used. This would suggest that there are substantial issues with the reproducibility of the data. In light of this, it is essential to provide PCA plots for all the various data sets presented herein, as well as p-value and FDR distributions for the outputs of all Ribolog TER test analyses.

Response to comment A.9

We should note that Fig. 2e pointed to by the reviewer is not Ribo-seq data, but rather 3'-end RNA-seq used for comparison of APA. The observed behavior was due to the fact that there are a large number of annotated poly-A sites in the genome, the vast majority of which are rare events and see very low counts. We have now added a filtering step to our analysis that removes pA sites that are largely unused in the data. We also repeated this experiment with three biological replicates and deeper sequencing to further boost the observed counts. These results, included in **Fig. R19** of the reviewer rebuttal and Fig. S3j-k in the revised manuscript, show a consistent volcano plot. Additionally, our biological findings are not contingent on the approach used for APAR significance testing as our analyses use the ranks of logAPAR values, similarly to what has been described above for logTER. For Ribo-seq, we have included PCA plots above (**Fig. R6**). The observed correlation coefficient of correlation between the replicates across all our datasets were ≥ 0.99 .

Referee Comment A.10

The Ribolog method is not the topic of the paper. However, it is difficult to claim its superiority over other published methods that have undergone peer-review and systematic benchmarking. In the absence of peer-review and benchmarking for Ribolog, it would be advisable to also demonstrate the quality of the data presented here using an established tool like RUST (O'Connor 2016). In addition, confidence in the validity of the results could be increased if similar results were obtained using published methods such as RiboDiff (Zhong 2017) or Anot2seq (Oertlin 2019).

Response to comment A.10

As the reviewer has alluded to, Ribolog is indeed not the topic of the paper; regardless, we have now provided access to our Ribolog benchmarks against the most commonly used tools in the field (https://github.com/goodarzilab/Ribolog/blob/master/benchmarks/ribolog_benchmarks.pdf). And as mentioned above, we have in fact tested the sensitivity of our analysis to the choice of the tool used, and in no case did we see a notable change in the effect size, direction, or statistical significance of our observations.

- b) strengthen the link of the components of the working model to the in vivo cancer metastatic phenotype**

Referee Comment B.1:

5. The authors should overexpress hnRNPC in MDA-LM2 and then check its metastatic potentials.

Response to comment B.1

To address this comment, we stably expressed HNRNPC or mCherry from the EF1 α promoter in MDA-LM2 cells, and performed experimental metastasis assays in NSG mice. As shown in **Fig. R7** (and Fig. 5b in the revised manuscript), HNRNPC expression in highly metastatic MDA-LM2 cells resulted in lower lung colonization rates as compared to mCherry-expressing cells.

Figure R7. MDA-LM2 cells stably expressing mCherry (Control) or HNRNPC (HNRNPC-OE) were injected via tail vein into NSG mice. Bioluminescence was measured at the indicated times (p value calculated using two-way ANOVA); area under the curve was measured at the final time point (p value calculated using one-tailed Mann-Whitney U -test). $n = 4-5$ mice per cohort.

Referee Comment B.2

PDLIM5 is shown to be the downstream of HNRNPC to suppress metastasis. The authors did PDLIM5 knockdown in MDA-MB-231. Ideally, additional overexpression of PDLIM5 in MDA-LM2 could provide more definitive conclusions. In any case, the gene structure, including 3'UTR length and binding sites for hnRNPC, and its APA regulation are poorly presented in the figures.

Response to comment B.2

To provide additional evidence that PDLIM5 acts downstream of HNRNPC, we first depleted HNRNPC in MDA-MD-231 cells, and then stably expressed mCherry or PDLIM5 from the EF1 α promoter in these cells. As shown in **Fig. R8a** (and Fig. 6f in the revised manuscript), PDLIM5 overexpression in HNRNPC-deficient cells brought down the metastatic potential *in vivo* to the level of control cells with intact HNRNPC expression. Additionally, we now provide the schematics of PDLIM5 poly(A) sites, along the 3'-end RNA sequencing results and HNRNPC binding sites, as determined by CLIP-seq (**Fig. R8b**, Fig. 6a in the revised manuscript).

Figure R8. (a) MDA-MB-231 cells stably expressing sgControl or sgHNRNPC and mCherry (Control) or PDLIM5 (PDLIM5-OE) were injected via tail vein into NSG mice. Bioluminescence was measured at the final time point (p value calculated using one-tailed Mann-Whitney U -test). $n = 4-5$ mice per cohort. **(b)** The schematics of the PDLIM5 proximal and distal poly(A) sites, HNRNPC binding sites as determined by CLIP-seq, and genome tracks illustrating the 3'-end RNA-seq data.

Referee Comment B.3

PDX models should be characterized to a greater extent

In the PDX samples (shown in 1b) there are far more transcripts with reduced translational machinery in the metastatic versus poorly metastatic tumours. This suggest a phenomenon, such as hypoxia, may be associated with the metastatic tumours. These PDX models should thus be better characterized before suggesting that HNRNPC is driving differential effects in these models. Indeed, associations seem much more modest in these models.

Response to comment B.3

The PDX models used in our study have been recently described in details⁸. It should be emphasized that unlike the MDA-LM2 vs MDA-Par model, which is a matched isogenic comparison, the highly and poorly metastatic PDXs are **not** matched. Given that the inter-patient heterogeneity is expected, it is not surprising that the effect sizes are smaller and the signal is slightly diluted. The observation that the HNRNPC regulon shows a significant signal in cell line models, PDXs, and clinical samples, makes this a highly consistent signal.

Referee Comment B.4

Models used do not fully test metastasis

The tail vein model is not a metastasis assay, but rather measures extravasation and survival and growth in the lung. The authors, should thus employ spontaneous metastasis models to validate their hypothesis. Given the availability of a PDX models, these should be used throughout, particularly for T4 studies. These latter studies should also be accompanied by a time course to better understand the kinetics of translational change and the extent to which these persist 6 hours after exposure and beyond.

Response to comment B.4

While lung colonization assays side-step intravasation, it is long established that intravasation is not a major bottleneck in metastasis (hence the presence of CTCs even in early stage tumors). Therefore, the lung colonization assays are often used as a stand-in for lung metastasis. We nevertheless performed orthotopic (mammary fat pad) injections of control or HNRNPC knockdown cells in NSG mice, monitored primary tumor growth, resected it 4 weeks post

injection, and then followed the lung metastasis. As shown in **Fig. R9** (and Fig. 5c-d in the revised manuscript), while the primary tumor volume was slightly but not significantly decreased in the HNRNPC-deficient cell cohort, HNRNPC knockdown cells showed a marked and significant increase in their metastatic potential.

Figure R9. (a) MDA-MB-231 cells stably expressing sgControl or sgHNRNPC were injected orthotopically (mammary fat pad) into NSG mice. Four weeks after injection the primary tumor was resected and lung bioluminescence was measured at the experiment end-point. Lung sections were stained with H&E (representative images shown). $n = 4-5$ mice per cohort. (b) Tumor volume of the orthotopically injected sgControl or sgHNRNPC cells was measured at Day

28. $n = 7-10$ mice per cohort. P value calculated using one-tailed Mann-Whitney U -test.

We also addressed the details of the T4-mediated inhibition of the lung metastatic colonization by MDA-LM2 cells. As described in our initial manuscript, the 6-hour T4 treatment with the IC20 concentration of MDA-LM2 cells *in vitro* markedly reduced its lung colonization capacity *in vivo*. To gain more insights into the poly(A) site selection kinetics caused by T4, we sampled the MDA-LM2 cells before and after the 6-hour treatment, as well as 24 and 72 hours post 6h treatment. We then performed 3'-end RNA-seq and calculated the changes in alternative polyadenylation using APAlig. As shown in **Fig. R10a** (and Fig. S7o in the revised manuscript), both 6h and 24h post-T4 treatment HNRNPC target mRNAs showed 3' UTR shortening (positive logAPAR values). The effect remained significant up to 72h post treatment, albeit to a lesser extent. Since cancer cell invasion, which is an early event, is the mechanism of metastasis in this case, even a transient effect is sufficient to cause a lasting difference between the two cohorts.

Finally, to support our data from the *in vitro* T4-treated MDA-LM2 cells, we assessed the impact of the systemic T4 treatment on the lung colonization by MDA-LM2 cells *in vivo*. We injected

NGS mice intraperitoneally with T4 or vehicle control for three consecutive days⁹ starting on the day of cancer cell injection. As shown in **Fig. R10b** (and Fig. 7j in the revised manuscript), the systemic T4 treatment significantly reduced lung metastatic burden as compared to vehicle control, supporting our finding that modulating the alternative polyadenylation program could be a therapeutic strategy in metastatic breast cancer.

Figure R10. (a) Box plots illustrating the changes in alternative polyadenylation (logAPAR) of HNRNPC targets between T4-treated (sampled at indicated time points) and untreated MDA-LM2 cells. *P* values calculated using Wilcoxon signed-rank test. (b) MDA-LM2 cells were injected via tail vein into NSG mice. On the day of cancer cell injection and for two additional consecutive days, animals were injected with 10 mg/kg T4 or an equivalent amount of vehicle control. Bioluminescence was measured at the indicated times (*p* value calculated using two-way ANOVA); area under the curve was measured at the final time point (*p* value calculated using one-tailed Mann-Whitney *U*-test). *n* = 4-5 mice per cohort.

Referee Comment B.5

Work related to PDLIM5 must be greatly improved
 Work related to PDLIM5 (from protein expression to results in animals) is not that convincing. Different translational efficiencies of the isoforms should be determined and mechanisms by which PDLIM5 may affect metastasis should be determined. Moreover, the authors would need to determine the extent to which PDLIM5 overexpression may reverse the effects of HNRNPC knock down. Finally, linkages with AGO2/miRNA should be assessed with cause-effect experimentation.

Response to comment B.5

In our initial manuscript, we showed that PDLIM5 mRNA showed reduced translation efficiency (TE) in highly metastatic and HNRNPC KD cells (Ribo-seq data), PDLIM5 protein was reduced in quantity in these conditions (TMT-MS and western blotting data) (Fig. S6d and Fig. 6b in the revised manuscript), and PDLIM5 mRNA underwent 3' UTR lengthening in these conditions (3'-end RNA-seq and RTqPCR data) (Fig. 6a and Fig. 6c in the revised manuscript). Using

experimental metastasis assays in mice we showed that PDLIM5 downregulation increased the lung colonization by MDA-MB-231 cells *in vivo*, as compared to control cells (Fig. 6e in the revised manuscript). We additionally performed experimental metastasis assays with control and PDLIM5 KD HCC1806 cells, which is another commonly used triple receptor negative breast cancer cell line. Just as in MDA-MB-231 cells, the PDLIM5 KD led to increased lung colonization in HCC1806 cells (Fig. R11, and Fig. S6g in the revised manuscript).

Figure R11. HCC1806-LM2 cells expressing shControl or sgPDLIM5 were injected via tail vein into NSG mice. Bioluminescence was measured at the indicated times (p value calculated using two-way ANOVA); area under the curve was measured at the final time point (p value calculated using one-tailed Mann-Whitney U -test). $n = 4$ -5 mice per cohort.

To provide additional evidence that PDLIM5 acts downstream of HNRNPC, we first depleted HNRNPC in MDA-MD-231 cells, and then stably expressed mCherry or PDLIM5 from the EF1 α promoter in these cells. As shown in **Fig. R12** (and Fig. 6f in the revised manuscript), PDLIM5 overexpression in HNRNPC-deficient cells brought down the metastatic potential *in vivo* to the level of control cells with intact HNRNPC expression.

Figure R12. MDA-MB-231 cells stably expressing sgControl or sgHNRNPC and mCherry (Control) or PDLIM5 (PDLIM5-OE) were injected via tail vein into NSG mice. Bioluminescence was measured at the final time point (p value calculated using one-tailed Mann-Whitney U -test). $n = 4$ -5 mice per cohort.

To better understand the mechanisms how the HNRNPC-PDLIM5 axis impacts metastasis, we determined proliferation rates, colony formation potential, and migration/matrigel invasion capacity of cell lines with depleted HNRNPC and PDLIM5, as well as HNRNPC overexpression. In agreement with our lung colonization assays *in vivo*, HNRNPC and PDLIM5 knockdowns caused an increase, while HNRNPC OE—a decrease, in cell migration and invasive capacity *in vitro* (**Fig. R13a** and Fig. S6h-I in the revised manuscript). These observations contrasted with the reduced proliferation and colony formation of HNRNPC-perturbed cells (**Fig. R13b-c** and Fig. S6j-l in the revised manuscript), consistently with primary tumor growth rates of HNRNPC KD cells *in vivo* (Fig. 5c in the revised manuscript). Similar to our *in vivo* data (Fig. 6f in the revised manuscript), ectopic PDLIM5 expression rescued the proliferation and colony formation defects of HNRNPC KD cells (**Fig. R13b-c** and Fig. S6m in the revised manuscript).

Figure R13. (a) Cell migration and cell invasion measurements in MDA-MB-231 cells expressing indicated sgRNAs

and/or overexpression constructs. Representative images are shown next to the bar plots. *P* values calculated using two-tailed unpaired *t*-test. **(b)** Proliferation rate measurements in MDA-MB-231 cells expressing indicated sgRNAs and/or overexpression (OE) constructs. **(c)** Colony formation unit (CFU) measurements in MDA-MB-231 cells expressing indicated sgRNAs and/or overexpression constructs. Representative images are shown below the bar plots.

Finally, to show that the PDLIM5 expression is dependent on its APA site choice, we constructed a reporter, consisting of N-terminally FLAG-tagged PDLIM5 CDS and three versions of 3' trailers: (i) an entire region encompassing both proximal and distal PDLIM5 poly(A) sites (hg38 chr4:94,664,164-94,666,887), labeled *full* 3' UTR; (ii) a region only up to the proximal poly(A) site (hg38 chr4:94,664,164-94,665,481), labeled *short* 3' UTR; (iii) and a region between the proximal and distal poly(A) sites (hg38 chr4:94,665,535-94,666,887), labeled *long* 3' UTR. We then delivered the three reporters to MDA-MB-231 cells by lentiviral transduction and Blasticidin selection, and determined the reporter protein expression by western blotting. As shown in **Fig. R14** (and Fig. 6d and Fig. S6f in the revised manuscript), removing the PDLIM5 3' UTR region beyond the proximal poly(A) site slightly increased the normalized FLAG-PDLIM5 abundance as compared to full 3' UTR. More importantly however, leaving only the PDLIM5 3' UTR region beyond the proximal poly(A) site substantially reduced the reporter protein abundance, confirming that the *cis* regulatory elements present between the proximal and distal poly(A) sites negatively control the PDLIM5 production.

Figure R14. (a) Schematics of the PDLIM5 3' UTR reporters. (b) The reporters illustrated in (a) were stably delivered to MDA-MB-231 cells and the reporter protein quantity was measured by western blotting in cell lysates (N = 6; representative image in biological duplicates is shown). (c) Relative normalized reporter protein quantity (FLAG signal normalized to GAPDH, relative to PDLIM5 *full* 3' UTR) is shown; error bars correspond to standard errors, *p* values calculated using two-tailed unpaired *t*-test.

c) **address the concerns regarding the modest correlation and exclude alternative working models, as noted by:**

Referee Comment C.1

Also, the possibility of nuclear export regulation was not considered. This is disappointing, especially given that hnRNPC is a nuclear protein and PABPC4 is a shuttling protein. Note that in general transcripts with long 3'UTRs are more likely to be retained in the nucleus, which could lead to low ribo-seq values. In addition, western blot should be used to confirm some ribo-seq data.

Response to comment C.1

Indeed, if the HNRNPC-dependent changes in poly(A) site selection impacted the nuclear export, the normalization of the Ribo-seq data with total RNA-seq would overestimate the translational impact. However, we provide several points of evidence this was not the case in our study.

Firstly, and as described above, we used cytoplasmic fraction to perform both nuclease footprinting aimed at ribosome profiling as well as for RNA-seq that we used to normalize Ribo-seq data. Even when we changed detergent concentration from 1% Triton X-100 to 0.15% Igepal CA-630 in the lysis buffer as per subcellular fractionation protocol, our cleared lysates showed comparable presence of cytoplasmic RNAs and proteins. Our new ribosome profiling data completely supports our initial observations.

We also repeated 3'-end RNA-seq experiments aimed at poly(A) site mapping with nuclear and cytoplasmic RNA from MDA-MB-231 and MDA-LM2, as well as control and HNRNPC KD cells, in biological triplicates. As in our initial findings, the cytoplasmic HNRNPC target RNAs underwent 3' UTR lengthening in highly metastatic or HNRNPC-depleted cells (**Fig. R3a-b**). Moreover, as shown in cumulative density plots in **Fig. R3c-d**, we detected no significant difference between HNRNPC target and non-target RNAs when comparing nuclear and cytoplasmic fractions for alternative polyadenylation (logAPAR) changes in highly metastatic or HNRNPC-depleted cells. Additionally, we have previously studied the quality control of mRNA retained in the nucleus in MDA-MB-231 and MDA-LM2 cells, and did not see any impact on the HNRNPC regulon described here¹.

In sum, these experiments suggest that HNRNPC-dependent regulation of poly(A) site choice and the resulting translational consequences are not contingent on the regulation of nuclear export.

We have also performed additional validations of our Ribo-seq data. As shown in **Fig. R15a** (and Fig. S2d in the revised manuscript), additional proteins, including PGK1 and TXNRD1, are downregulated in highly metastatic and in HNRNPC KD cells. Importantly, these observations were not due to the differences in mRNA levels (**Fig. R15b**, Fig. S2c in the revised manuscript).

Figure R15. (a) Cell lysates from MDA-MB-231, MDA-LM2, and MDA-MB-231 cells expressing sgControl or sgHNRNPC were analysed by western blotting in biological triplicates. Relative normalized (to GAPDH) protein quantities and standard errors are indicated below the images. **(b)** Relative mRNA quantities of the indicated transcripts from MDA-MB-231, MDA-LM2, and MDA-MB-231 cells expressing sgControl or sgHNRNPC as determined by RNA-seq (N = 3).

Referee Comment C.2

Statistically significant but quite modest correlations underlie the path from HNRNPC binding through alternative 3' end isoforms to Ago2-mediated repression, leaving open the possibility of alternative or indirect connections.

Response to comment C.2

We agree with the reviewer that the translational repression of longer 3' UTR-bearing mRNAs by miRNAs might be not the only mechanism in action. We thus do not claim that the RNA interference is the sole pathway governing the translational repression of HNRNPC targets. While we failed to identify additional RNA binding proteins that were associated both with the HNRNPC regulon and its reduced translation, we cannot rule out that select RNA binding proteins participate in the repression of specific targets in the regulon. We updated our discussion to emphasize this point.

Referee Comment C.3

It is claimed that, "HNRNPC controls the alternative polyadenylation of its targets" (p. 5, l. 43). However, it would seem that an alternative explanation could be provided by nuclear export, which controls apparent TE (since the nuclear pool of an RNA is untranslated but shows up in mRNA-Seq) and can certainly be influenced by 3' UTRs and by nuclear RBPs. How is it excluded that HNRNPC promotes nuclear export and subsequent degradation of long-3'UTR variants?

Response to comment C.3

To test this alternative possibility, we performed 3'-end RNA-seq from the cytoplasmic and nuclear fraction, in highly and poorly metastatic, as well as control and HNRNPC KD MDA-MB-231 cells. As described above, and as shown in cumulative density plots in **Fig. R16** (and Fig. S3h-i in the revised manuscript), we detected no significant difference between HNRNPC target and non-target RNAs when comparing nuclear and cytoplasmic fractions for alternative polyadenylation (logAPAR) changes in highly metastatic or HNRNPC-depleted cells.

Figure R16. Cumulative density plot of log fold-change between nuclear and cytoplasmic fractions, comparing alternative polyadenylation ratio (logAPAR) between (top panel) MDA-LM2 and MDA-MB-231 cells, or (bottom panel) sgControl and

sgHNRNPC cells, for HNRNPC 3' UTR target and non-target mRNAs. Median difference (ΔM) and p value (calculated using Mann-Whitney U -test) are shown.

Referee Comment C.3

3) Page 5, lines 25-27, Figure S2b

Perhaps the figure has been misunderstood, however, the correlation coefficients between translationally repressed mRNAs in MDA-LM2 and HNRNPC KD cells appear very low, which seems to suggest that the data are actually quite poorly correlated. Thinking in terms of variance explained, with a coefficient of 0.05, this would be 0.25%. This is either quite unconvincing, or overly confusing. If the later, it may be helpful to visualize the overlap by overlaying the significantly translationally suppressed mRNAs from one cell line onto the volcano plot for the other by coloring these points.

Response to comment C.3

The correlation coefficients comparing **general** changes in translation efficiency (logTER) or alternative polyadenylation (logAPAR) across the transcriptome between highly and poorly metastatic and HNRNPC KD and control cells are indeed minimal. In fact, we do not expect the HNRNPC knockdown to recapitulate the ensemble of translational or alternative polyadenylation changes occurring during the metastatic progression in breast cancer. To support this, we and others have found a variety of other pathways contributing to RNA processing¹⁰ and translation control¹¹ in metastasis. This study thus highlights that HNRNPC is a part of an ensemble of mechanisms, emphasizing a new link of HNRNPC deficiency with alternative polyadenylation and translational control in breast cancer metastasis. With this in mind, Fig. S2b and S2e in the original manuscript illustrate that while **general** correlation between the two conditions is minimal (described by ρ), the HNRNPC **target** RNAs are enriched among translationally repressed or longer 3' UTR-bearing transcripts in both conditions (a red peak in the two-dimensional heatmaps). Similarly, the HNRNPC regulon is depleted among translationally activated or shorter 3' UTR-bearing transcripts in both conditions (a blue valley in the heatmaps). Such data visualization allows to compare the continuous variables in both datasets, instead of applying an arbitrary cut off values to one data set and project it on the other¹².

References

1. Fish, L. *et al.* Nuclear TARBP2 Drives Oncogenic Dysregulation of RNA Splicing and Decay. *Mol. Cell* **75**, 967- 981.e9 (2019).
2. McGlincy, N. J. & Ingolia, N. T. Transcriptome-wide measurement of translation by ribosome profiling. *Methods* **126**, 112–129 (2017).
3. Mayer, A. *et al.* Native Elongating Transcript Sequencing Reveals Human Transcriptional Activity at Nucleotide Resolution. *Cell* **161**, 541–554 (2015).
4. Gierliński, M. *et al.* Statistical models for RNA-seq data derived from a two-condition 48-replicate experiment.

Bioinformatics **31**, 3625–3630 (2015).

5. Chothani, S. *et al.* deltaTE: Detection of Translationally Regulated Genes by Integrative Analysis of Ribo-seq and RNA-seq Data. *Current Protocols in Molecular Biology* **129**, (2019).
6. Duncan, C. D. S. & Mata, J. The translational landscape of fission-yeast meiosis and sporulation. *Nat Struct Mol Biol* **21**, 641–647 (2014).
7. Love, M. I., Huber, W. & Anders, S. Moderated estimation of fold change and dispersion for RNA-seq data with DESeq2. *Genome Biol.* **15**, 550 (2014).
8. Winkler, J. *et al.* Dissecting the contributions of tumor heterogeneity on metastasis at single-cell resolution. <http://biorxiv.org/lookup/doi/10.1101/2022.08.04.502697> (2022) doi:10.1101/2022.08.04.502697.
9. Nakamura, P. A. *et al.* Small molecule Photoregulin3 prevents retinal degeneration in the RhoP23H mouse model of retinitis pigmentosa. *eLife* **6**, e30577 (2017).
10. Fish, L. *et al.* A prometastatic splicing program regulated by SNRPA1 interactions with structured RNA elements. *Science* **372**, eabc7531 (2021).
11. Goodarzi, H. *et al.* Modulated Expression of Specific tRNAs Drives Gene Expression and Cancer Progression. *Cell* **165**, 1416–1427 (2016).
12. Alkallas, R., Fish, L., Goodarzi, H. & Najafabadi, H. S. Inference of RNA decay rate from transcriptional profiling highlights the regulatory programs of Alzheimer's disease. *Nat Commun* **8**, 909 (2017).

Reviewer #1:

Remarks to the Author:

This manuscript from Navickas *et al.* reports a function of hnRNPC in regulation of alternative polyadenylation, its interaction with PABPC4, and their relevance to translational regulation in cancer cell metastasis. The authors used a battery of techniques, such as ribo-seq, 3' end seq, CLIP-seq, TMT-MS, CoIP-MS and PDX models, to substantiate their claims. The results largely support their conclusions. However, some critical pieces are missing.

We thank the reviewer for taking the time to read our work and provide thoughtful comments. We are confident that addressing their comments below have helped improve our work.

Major:

Comment 1.1. The enrichment of U-rich motifs for translationally repressed transcripts in cells with higher metastatic potentials is quite interesting. This should be validated using a reporter assay. Ideally, regulation of APA, such as in the case of PDLIM5, should also be validated using a reporter assay. The reporter assay can be carried out in MDA-MB-231 vs. MDA-LM2, or one of the cells with knockdown or overexpression of hnRNPC.

Response 1.1.

We thank the reviewer for this suggestion. To support our observations, we designed a massively parallel reporter assay (MPRA) that allows to monitor both the changes in alternative

polyadenylation and translation. We used a bidirectional promoter dual color reporter, expressing BFP and mCherry proteins. We cloned the region encompassing two alternative poly(A) sites of KAT14 gene (hg17 chr20:18,187,963-18,188,401) downstream of the BFP coding sequence. We chose KAT14 as it satisfied two criteria: (i) the distance between the alternative poly(A) sites allowing an amplicon compatible with Illumina sequencing (this is not the case for PDLIM5, the distance between its poly(A) sites being several kb long); (ii) the usage of both poly(A) sites in MDA-MB-231 cells detectable by RT-PCR. We then cloned a library of sequences (150 bp long), containing HNRNPC binding sites in 3' UTRs (as determined by our CLIP-seq experiments) or nucleotide-content matched shuffled sequences, upstream of the KAT14 proximal poly(A) site in the reporter (**Fig. R1a**, Fig. S2h in the revised manuscript).

We delivered the reporter library to MDA-MB-231 cells by lentiviral infection and puromycin selection. We then transfected the reporter cell pool with siRNAs targeting HNRNPC or a non-targeting control. Seventy-two hours post transfection, we analyzed the cells by flow cytometry: siHNRNPC-transfected cells showed lower BFP/mCherry ratio when compared to siCTRL cell pool (**Fig. R1b**, and Fig. S2i in the revised manuscript). We extracted total RNA from a portion of transfected cells, and used the remaining cells for sorting into two bins, representing the top and bottom 25% of the BFP/mCherry ratio. We used RNA extracted from the total and sorted cells to amplify the reporter RNA by RT-PCR. We used an anchored oligo dT primer containing a UMI for reverse transcription, and a BFP-specific primer for PCR. We sequenced the amplicon libraries on a paired-end Illumina sequencing run, allowing to match the poly(A) site chosen (proximal or distal) with the library insert upstream of the proximal poly(A) site. As shown in the **Fig. R1c** (and Fig. S2j in the revised manuscript), HNRNPC knockdown led to a preferential choice of the distal poly(A) site in reporter mRNAs bearing HNRNPC binding sites when compared to matched shuffled controls. Furthermore, analyzing high and low BFP/mCherry ratio cells separately showed that lower reporter protein expression was significantly associated with longer (lower logAPAR values) reporter 3' UTRs with HNRNPC binding sites as compared to shuffled controls. Importantly, this observation was dependent on HNRNPC (**Fig. R1d**, and Fig. S2k in the revised manuscript).

Figure R1. (a) Massively parallel reporter assay (MPRA) schematics. BFP and mCherry mRNAs were expressed from a bidirectional promoter. KAT14 3' trailer was cloned downstream of BFP, including its proximal and distal poly(A) sites (pA1 and pA2). Upstream of the proximal poly(A) site, HNRNPC binding sites, or nucleotide content controlled scrambled sequenced, were cloned. (b) The comparison of relative reporter alternative polyadenylation ratio (logAPAR) between transcripts bearing HNRNPC binding sites and scrambled controls, in siControl or siHNRNPC transfected cells. (c) The comparison of reporter logAPAR between transcripts bearing HNRNPC binding sites and scrambled controls, showing high and low BFP/mCherry ratio as determined by flow cytometry, in siControl or siHNRNPC transfected cells.

Comment 1.2. The authors appear to have used total RNA to normalize ribo-seq data. This approach would miss regulation of nuclear export. They should use cytoplasmic RNA for normalization. Also, the possibility of nuclear export regulation was not considered. This is disappointing, especially given that hnRNP is a nuclear protein and PABPC4 is a shuttling protein. Note that in general transcripts with long 3'UTRs are more likely to be retained in the nucleus, which could lead to low ribo-seq values. In addition, western blot should be used to confirm some ribo-seq data.

Response 1.2.

We thank the reviewer for this valuable remark. We should note that we had previously studied RNA export in these models of metastasis¹, and we had not observed a signal for differential

export of the HNRNPC-bound RNAs. Indeed, if the HNRNPC-dependent changes in poly(A) site selection impacted the nuclear export, the normalization of the Ribo-seq data with total RNA-seq would overestimate the translational impact. However, we explicitly provide several points of evidence this was not the case in our study.

In our initial manuscript, we performed ribosome profiling from the cell lysates prepared with 1% (v/v) Triton X-100 as detergent². We used the same lysate to perform nuclease footprinting for Ribo-seq or extract “total” (i.e. not RNaseI-treated) RNA for RNA-seq. That is, we used clarified lysates for RNA extraction and not whole cell lysates.

To determine if our lysates contained nuclear fraction, we performed subcellular fractionation using our initial lysis buffer with 1% (v/v) Triton X-100 as detergent, or 0.15% Igepal CA-630 (NP40 alternative), as described³. We collected the cytoplasmic fraction as a soluble lysate, and nuclei as a pellet centrifuged through sucrose cushion. We prepared nuclear lysates by solubilizing the pellet in RIPA buffer. We then saved half of the lysates for protein analysis, and used the other half for RNA extraction. As shown in **Fig. R2a**, in both lysis buffers, we detected mature HPRT mRNA mostly in the cytoplasmic fraction, while the intron-containing HPRT pre-mRNA, as well as MALAT1 and NEAT1 ncRNAs (known to be retained in the nucleus), were predominantly nuclear. At the protein level, as shown in **Fig. R2b**, we detected HNRNPC and

WTAP predominantly in the nuclear fraction, and GAPDH and beta-tubulin mostly in the cytoplasmic fraction as expected, similarly in both lysis buffers.

Figure R2. (a) Relative proportions of various RNAs in nuclear (orange) and cytoplasmic (blue) fractions, extracted using 1% Triton X-100 (left panel) or 0.15% Igepal CA-630 (right panel) detergents, as determined by RTqPCR. **(b)** Western blot images detecting the proteins in subcellular fractions as described in (a).

Next, we repeated the Ribo-seq experiments, but with the modified lysis buffer, substituting Triton X-100 for 0.15% Igepal CA-630. We performed the experiments in biological triplicates, in MDA-MB-231 and MDA-LM2, as well as control and HNRNPC KD cells. We used cytoplasmic lysates both for nuclease foot printing and undigested RNA extraction. From the same preparations, we also kept nuclei pellets and extracted nuclear RNA. As shown in **Fig. R3** and Fig. S3I-m in the revised manuscript, our results from the cytoplasmic fraction recapitulated our initial findings: HNRNPC target RNAs were translationally repressed in highly metastatic or HNRNPC-depleted cells.

Figure R3. (a) Bottom: Volcano plot showing the distribution of changes in translation efficiency ratio (logTER) in highly metastatic MDA-LM2 cells compared to parental MDA-MB-231 cells. Statistically significant (logistic regression, $p < 0.01$) observations are highlighted in orange. Top: Enrichment of the HNRNPC targets as a function of logTER between sgHNRNPC and sgControl cells. mRNAs are divided into equally populated bins according to logTER (dotted vertical lines delineate the bins); the y-axis shows the frequency of the HNRNPC targets that we identified in each bin (dotted horizontal line denotes the average HNRNPC target frequency across all transcripts). Bins with significant enrichment (logistic regression, $FDR < 0.05$; red) or depletion (blue) of HNRNPC targets are denoted with a black border. Also included are mutual information (MI) values and their associated z-scores. **(b)** Comparison between sgHNRNPC and sgControl MDA-MB-231 cells, otherwise as in (a).

We also repeated 3'-end RNA-seq experiments aimed at poly(A) site mapping with nuclear and cytoplasmic RNA from MDA-MB-231 and MDA-LM2, as well as control and HNRNPC KD cells,

in biological triplicates. Just as with Ribo-seq experiments described above, changes in alternative polyadenylation in the cytoplasmic fraction recapitulated our initial findings: HNRNPC target RNAs underwent 3' UTR lengthening in highly metastatic or HNRNPC-depleted cells (Fig. R4a-b and Fig. S3j-k in the revised manuscript). Moreover, as shown in cumulative density plots in Fig. R4c-d and Fig. S3h-i in the revised manuscript, we detected no significant difference between HNRNPC target and non-target RNAs when comparing nuclear and cytoplasmic fractions for alternative polyadenylation (logAPAR) changes in highly metastatic or HNRNPC-depleted cells.

Figure R4. (a) Bottom: Cumulative density plot of alternative polyadenylation ratios (logAPAR) comparing MDA-LM2 to MDA-MB-231 cells, for HNRNPC 3' UTR target and non-target mRNAs. Median difference (ΔM) and p value (calculated using Mann-Whitney U -test) are shown. Middle: Volcano plot showing distribution of changes in logAPAR in MDA-LM2 compared to MDA-MB-231 cells. Top: Enrichment of the HNRNPC-bound 3' UTRs as a function of logAPAR between MDA-LM2 and MDA-MB-231 cells; statistics as in Figure R2a. (b) Comparison between sgHNRNPC and sgControl MDA-MB-231 cells, otherwise as in (a). (c) Cumulative density plot of log fold-change between nuclear and cytoplasmic fractions, comparing logAPAR between MDA-LM2 and MDA-MB-231 cells, for HNRNPC 3' UTR target and non-target mRNAs; statistics as in (a). (d) Comparison between sgHNRNPC and sgControl MDA-MB-231 cells, otherwise as in (c).

In sum, this data suggests that HNRNPC-dependent regulation of poly(A) site choice and the resulting translational consequences is not contingent on the regulation of nuclear export.

Comment 1.3. Previous studies have shown PABPC1 knockdown could lead to 3'UTR shortening (Li et al. PLoS Genetics, 2015). The authors should check if PABPC4 knockdown would also lead to PABPC1 downregulation, as they often depend on each other. In addition, the lack of APA regulation in double knockdown of PABPC4 and hnRNPC is intriguing. The authors have not provided a satisfactory answer.

Response 1.3.

As shown in **Fig. R5**, the PABPC1 was expressed at comparable levels in control and PABPC4 KD cells in our study.

Regarding our data comparing the changes in alternative polyadenylation between control and HNRNPC KD conditions in cells with PABPC4 KD (Fig. 3d in the initial manuscript), we claim that the HNRNPC regulon does not undergo 3' UTR lengthening upon HNRNPC knockdown if PABPC4 is not present (as shown in the heatmap of the top panel of Fig. 3d). As PABPC4 and HNRNPC physically interact (Fig. 3b), and their target mRNAs largely overlap (Fig. 3c), we conclude that HNRNPC-mediated control of alternative polyadenylation is contingent on PABPC4. As shown in Fig. 3d however, multiple alternative polyadenylation events are still present in the double HNRNPC/PABPC4 KD as compared to PABPC4 knockdown only.

Figure R5. PABPC1 mRNA levels in control and PABPC4-deficient MDA-MB-231 cells, as determined by RNA-seq.

P value calculated using DESeq2.

Comment 1.4. The authors should show their 3' sequencing data using genome tracks. The current figures show global patterns, giving nebulous impressions.

Response 1.4.

We now included 3'-end RNA-seq data using genome tracks (Fig. 6a in the revised manuscript).

Comment 1.5. The authors should overexpress hnRNPc in MDA-LM2 and then check its metastatic potentials.

Response 1.5.

We thank the reviewer for this comment. We stably expressed HNRNPC or mCherry from the EF1 α promoter in MDA-LM2 cells, and performed experimental metastasis assays in NSG mice.

As shown in **Fig. R6** (and Fig. 5b in the revised manuscript), HNRNPC expression in highly metastatic MDA-LM2 cells resulted in lower lung colonization rates as compared to mCherry-expressing cells.

Figure R6. MDA-LM2 cells stably expressing mCherry (Control) or HNRNPC (HNRNPC-OE) were injected via tail vein into NSG mice. Bioluminescence was measured at the indicated times (p value calculated using two-way ANOVA); area under the curve was measured at the final time point (p value calculated using one-tailed Mann-Whitney U -test). $n = 4-5$ mice per cohort.

Minor:

Comment 1.6. In the figure 2c and 2d, the majority of hnRNPC targets contain PASs in the 3' UTR and undergo 3' UTR lengthening in MDA-LM2 compared to MDA-MB-231. Does this subset of HNRNPC targets have functions in cell migration and metastasis? The authors should carry out Gene Ontology analysis on these.

Response 1.6.

We thank the reviewer for this question. Indeed, the genes that show 3' UTR lengthening (lower logAPAR values) in highly metastatic or HNRNPC-deficient cells have been associated with breast cancer and metastasis (**Fig. R7**).

Figure R7. iPAGE⁴ pathway enrichment analysis comparing alternative polyadenylation ratio (logAPAR, proximal to distal poly(A) site) between MDA-LM2 and MDA-MB-231 (top panel) or sgHNRNPC and sgControl (bottom panel) cells. All data points were distributed into equally populated bins, represented by the black and red box plots. In each bin, enrichment for indicated pathways or datasets (from the Molecular Signature Database, MSigDB) was calculated and represented by a color coded heatmap. Only relevant pathways or datasets are shown.

Comment 1.7. The authors seem to indicate that cells with high metastatic potentials express longer 3'UTRs. Given the previously reported connection between 3'UTR size and cell proliferation, could the authors comment on how cell proliferation may be involved in their model?

Response 1.7.

We determined proliferation rates, colony formation potential, and migration/matrigel invasion capacity of cell lines with depleted HNRNPC and PDLIM5, as well as HNRNPC overexpression. In agreement with our lung colonization assays *in vivo*, HNRNPC and PDLIM5 knockdowns caused an increase, while HNRNPC OE—a decrease, in cell migration and invasive capacity *in vitro* (Fig. R8a and Fig. S6h-l in the revised manuscript). These observations contrasted with the reduced proliferation and colony formation of HNRNPC-perturbed cells (Fig. R8b-c and Fig. S6j-l in the revised manuscript), consistently with primary tumor growth rates of HNRNPC KD cells *in vivo* (Fig. 5c in the revised manuscript). Similar to our *in vivo* data (Fig. 6f in the revised manuscript), ectopic PDLIM5 expression rescued the proliferation and colony formation defects of HNRNPC KD cells (Fig. R8b-c and Fig. S6m in the revised manuscript).

Figure R8. (a) Cell migration and cell invasion measurements in MDA-MB-231 cells expressing indicated sgRNAs

and/or overexpression constructs. Representative images are shown next to the bar plots. *P* values calculated using two-tailed unpaired *t*-test. **(b)** Proliferation rate measurements in MDA-MB-231 cells expressing indicated sgRNAs and/or overexpression (OE) constructs. **(c)** Colony formation unit (CFU) measurements in MDA-MB-231 cells expressing indicated sgRNAs and/or overexpression constructs. Representative images are shown below the bar plots.

We conclude that the affected HNRNPC regulon primarily contributes to the cell metastatic potential through the invasive and not proliferative mechanism.

Comment 1.8. Many hnRNPC-bound targets are also AGO2 targets. U-rich motifs may make miRNA target sites more exposed to AGO2. Do they see miRNA target sites being close to hnRNPC binding sites?

Response 1.8.

The median distance between HNRNPC 3' UTR sites (as determined by CLIP-seq) and the closest miRNA site (as determined by AGO2 CLIP-seq⁵) is 87 nucleotides, as shown in **Fig. R9**.

Figure R9. Distance distribution between HNRNPC binding sites and AGO2-bound miRNA sites.

Comment 1.9. PDLIM5 is shown to be the downstream of HNRNPC to suppress metastasis. The authors did PDLIM5 knockdown in MDA-MB-231. Ideally, additional overexpression of PDLIM5 in MDA-LM2 could provide more definitive conclusions. In any case, the gene structure, including 3'UTR length and binding sites for hnRNPC, and its APA regulation are poorly presented in the figures.

Response 1.9.

We thank the reviewer for these suggestions. To provide additional evidence that PDLIM5 acts downstream of HNRNPC, we first depleted HNRNPC in MDA-MD-231 cells, and then stably expressed mCherry or PDLIM5 from the EF1 α promoter in these cells. As shown in **Fig. R10a** (and Fig. 6f in the revised manuscript), PDLIM5 overexpression in HNRNPC-deficient cells

brought down the metastatic potential *in vivo* to the level of control cells with intact HNRNPC expression. Additionally, we now provide the schematics of PDLIM5 poly(A) sites, along the 3'-end RNA sequencing results and HNRNPC binding sites, as determined by CLIP-seq (**Fig. R10b**, Fig. 6a in the revised manuscript).

Figure R10. (a) MDA-MB-231 cells stably expressing sgControl or sgHNRNPC and mCherry (Control) or PDLIM5 (PDLIM5-OE) were injected via tail vein into NSG mice. Bioluminescence was measured at the final time point (p value calculated using one-tailed Mann-Whitney U -test). $n = 4-5$ mice per cohort. (b) The schematics of the PDLIM5 proximal and distal poly(A) sites, HNRNPC binding sites as determined by CLIP-seq, and genome tracks illustrating the 3'-end RNA-seq data.

Comment 1.10. In supplementary figure 1f, scale bars should be included.

Response 1.10.

We thank the reviewer for this remark. The scale bars are now described in the figure legend.

Comment 1.11. Page 4, line 26, 'RNA levels only partially explain the changing in protein levels ($R = 0.5$, $p < 2 \times 10^{-16}$). Which figure is this referring to?

Response 1.11.

This data does not refer to a figure in the manuscript.

Comment 1.12. In supplementary figure 3, genes regulated by PABPC4 have a higher potential to be HNRNPC targets. Gene ontology results should be presented for those genes. In addition, validation should be carried out for some top hits, using qPCR for example.

Response 1.12.

We thank the reviewer for this remark. Indeed, the genes with longer 3' UTRs (lower logAPAR values) in PABPC4-deficient cells have been associated with various cancers, including breast cancer, and BMP signaling (**Fig. R11**).

Figure R11. iPAGE⁴ pathway enrichment analysis comparing alternative polyadenylation ratio (logAPAR, proximal to distal poly(A) site) between sgPABPC4 and sgControl cells. All data points were distributed into equally populated bins, represented by the black and red box plots. In each bin, enrichment for indicated pathways or datasets (from the Molecular Signature Database, MSigDB) was calculated and represented by a color coded heatmap. Only relevant pathways or datasets are shown.

Comment 1.13. Figure S5a, the plot shown on the left does not seem to agree with bioluminescence pictures shown on the right.

Response 1.13.

We thank the reviewer for this remark. We now replaced the bioluminescence picture for the sgControl-sgPABPC4 condition that better represents the median value of the cohort.

Reviewer #2:

Remarks to the Author:

This manuscript argues that a metastatic gene expression program results from a shift towards longer 3' UTRs and weaker translation in the affected genes. This shift is driven by lower levels of HNRNPC and PABPC4. The pro-metastatic gene expression program is defined first through Ribo-seq on cell lines representing primary and metastatic tumors. A uridine-rich motif is enriched in transcripts with lower translation under metastatic conditions. Analysis of these targets leads to the nomination of HNRNPC as an RNA-binding protein that recognizes this motif. The association of these HNRNPC binding sites with cleavage and polyadenylation signals motivates 3' end sequencing experiments, which show a change in alternative polyadenylation that is correlated with HNRNPC binding. An RNA-dependent interaction between HNRNPC and the poly-(A) binding protein PABPC4 leads to the observation that PABPC4 knock-down produces 3' end shifts that overlap significantly with those seen in HNRNPC

knock-down, and further, a genetic interaction between these two knock-down conditions. This is interpreted to argue that HNRNPC and PABPC4 work together to modulate isoform differences. The reduced translation, then, is argued to result from additional microRNA binding sites introduced into the longer 3' UTRs seen in metastatic cells and in HNRNPC depletion.

The manuscript goes on to argue that HNRNPC restrains metastasis. Knock-down of HNRNPC accelerates engraftment of tumor xenograft metastases, and lower levels of HNRNPC or PABPC4 correlate with worse prognosis in human breast cancers as well. The gene PDLIM5 is identified as a particular target of 3' UTR extension, and it also appears to restrain metastasis in mouse models and correlate with improved prognosis in patients. Notably, a small molecule drug previously found to shift polyadenylation sites can in part reverse the 3' UTR lengthening seen in HNRNPC knock-down.

This study reports a range of observations about RNA processing in primary and metastatic tumors, tying them to effects seen in cancer models. The reported molecular results — translational changes, 3' UTR extension, and changes in HNRNPC — all seem individually well supported, but the causal link between them seems much weaker, though. Statistically significant but quite modest correlations underlie the path from HNRNPC binding through alternative 3' end isoforms to Ago2-mediated repression, leaving open the possibility of alternative or indirect connections.

We thank the reviewer for their detailed and thoughtful questions. We hope the points we have addressed below remove the perceived weaknesses in the causal links in the pathway we have described.

My specific concerns are

Comment 2.1. In discussing the new "ribolog" tool for Ribo-seq analysis, it is claimed that it has "as few a priori assumptions as possible" and "does not assume a negative binomial (NB) distribution of read counts" (p. 4, ll. 8 - 18). However, logistic regression implicitly assumes a

standard binomial error model, which is strictly no more conservative than the NB read count model used in many RNA-Seq tools, and in practice will be less conservative. In particular, the NB models incorporates an overdispersion term, while ribolog effectively assumes that this term is zero. Lower variance estimates then lead to stronger statistical significances.

Response 2.1.

We thank the reviewer for bringing up this issue. Before addressing the comment, we need to clarify a few points. First, we should emphasize that the key insights derived from our Ribo-seq data is based on gene-set enrichment analysis of differential translation efficiency values. In other words, these analyses are not sensitive to the choice of statistical test, as they do not rely on p-values but rather the ranking of genes based on their logTERs. Below, we have included the comparison of logTER values for alternative poly-adenylated genes that are bound or unbound by HNRNPC (Fig. S1i and S2a in the revised manuscript) with Xtail, Riborex, and RiboDiff as well (Fig. R12).

Figure R12. Cumulative density plot of translation efficiency ratios (logTER) comparing MDA-LM2 to MDA-MB-231 (Mda-Par, top) or sgHNRNPC to sgControl MDA-MB-231 (bottom) cells, for HNRNPC 3' UTR target and non-target mRNAs, using Xtail (left), Riborex (center) or Ribodiff (right). Median difference (ΔM) and p value (calculated using Mann-Whitney U -test) are shown.

Second, the aforementioned similarity of results is largely expected because the main determinants of logTER in all these tools are simply the normalized RPF and RNA counts. As an example, we have included the pair-wise Pearson correlation matrix of logTER values calculated using Ribolog, Xtail, RiboDiff, and Riborex. These results, which were generated on simulated counts from the Xtail package, show almost identical logTER values across tools except for Anot2seq (Table R1).

	Ribolog	Xtail	Riborex	Ribodiff
Ribolog		0.999	0.998	1.000
Xtail	0.999		1.000	0.999
Riborex	0.998	1.000		0.998
Ribodiff	1.000	0.999	0.998	

Table R1. Pairwise Pearson correlation matrix of logTER values generated on simulated counts from the Xtail package, obtained with Ribolog, Xtail, Riborex, and Ribodiff.

Third, Ribolog allows for correction of local heterogeneity (stalling) biases in Ribo-seq data by implementing a CELP module. Both simulated data and real-world examples indicate that this correction improves logTER estimates. We found that after applying the CELP correction, residuals calculated as the difference between Ribolog and Xtail logTER values are significantly correlated with changes in protein levels (**Fig. R13**). This observation establishes that starting from the same data, Ribolog captures the protein landscape better.

Figure R13. Regression analysis comparing the protein logFC in MDA-LM2 versus MDA-MB-231 cells to the residual logTER from the values calculated using Xtail subtracted from those calculated using Ribolog. Correlation coefficient and the resulting p -value are shown.

Finally, we have made the results of our benchmarking tests available online at (https://github.com/goodarzilab/Ribolog/blob/master/benchmarks/ribolog_benchmarks.pdf). These benchmarks establish Ribolog as a strong contender with (i) the highest ROC- and PR-AUC, (ii) one of the lowest CPU times, and (iii) incredibly robust to changes in sequencing depth.

Now, with regards to the reviewer's comment, we wish to emphasize the following:

1. The reviewer is correct in that TER testing in Ribolog is based on the binomial distribution. However, Ribolog goes beyond a simple logistic regression and contains multiple additional modules for stalling correction, replicate reproducibility check and QC, meta-analysis, empirical null testing, etc.
2. The significance and necessity of overdispersion in NB distribution is quite different from binomial distribution. With overdispersion set to zero, NB reduces to a Poisson distribution. Poisson, a popular choice to model count data, is characterized by a single parameter which is equal to its mean and variance. An overdispersion parameter is incorporated in NB to relax this constraint and allow mean and variance to vary more freely. No such requirement for equality of mean and variance exists in the standard binomial distribution in the first place. Therefore, the inclusion of an overdispersion term is not required. However, we should note that in some empirical datasets variance can exceed the predictions of the best fitted binomial distribution; this has motivated the development of overdispersed variations of binomial, e.g. beta-binomial distribution (requires estimating 3 parameters instead of 2). In the case of Ribo-seq comparisons we deemed this unnecessary as the standard binomial distribution allows inequality of mean and variance. Furthermore, Ribolog also takes advantage of empirical nulls for significance testing which greatly reduces the risk of inflated p -values and false positives (also pointed out by the reviewer). This is confirmed by our benchmarking results (https://github.com/goodarzilab/Ribolog/blob/master/benchmarks/ribolog_benchmarks.pdf).
3. In the standard binomial model, all observations are deemed independent of one another. This is a reasonable assumption for Ribo-seq data. In a ribosome profiling experiment, tens to

hundreds of RNA and RPF reads per transcript are sampled from thousands of cells, each containing hundreds to thousands of copies of a (averagely expressed) transcript. Since capturing multiple fragments of the very same RNA molecule is exceedingly rare, the independence assumption is warranted. In conclusion, although the test for the ratio of RPF and RNA is performed using logistic regression

(binomial model), we make no assumptions regarding count distributions of either RNA or RPF reads.

Comment 2.2. As a related point, the discussion of the Ribolog method (p. 17) mentions problems that arise in Ribo-seq data analysis but does not credit the fact that existing tools offer clear solutions to these problems.

First, "variance sharing" underlies many Ribo-seq and RNA-seq analysis tools, and addresses the challenge in estimating overdispersion parameters with a limited number of replicates (Anders & Huber, *Genome Biol* 2010; Robinson & Smyth, *Bioinformatics* 2010). In the rare case of heavily replicated sequencing data, "The best analysis scheme for our data [726 individual RNA-Seq samples] was to first normalize using the DESeq method and apply a generalized linear model assuming a negative binomial distribution using either edgeR or DESeq software" (Lin & Harbison, *BMC Genomics* 2016).

Response 2.2.

We thank the reviewer for raising this point. As mentioned above, our benchmarking document assesses the performance of multiple tools on simulated counts (from the original Xtail publication) and how they compare with Ribolog. And it is clear that Ribolog surpasses existing tools in a number of important metrics. With regards to this comment, the mere existence of proposed solutions does not imply that these solutions have been rigorously tested in every case and their use is broadly advisable in every situation. The reviewer is correct in their assertion that several solutions have been proposed for variance estimation in count data, including Bayesian shrinkage method. However, the fact that at low replicate counts of Ribo-seq datasets (often at 2 or 3 per condition) these estimates are far from perfect is not disputed. Furthermore, shrinkage and other variance sharing methods implicitly assume the same relationship between mean and variance for genes of the same expression level, which facilitates the estimation of dispersion parameter but explicitly adds an additional core assumption. Since highly replicated datasets have not been created for Ribo-seq, the veracity of this critical assumption has not been tested. For example, consider this quote from the developers of RiboDiff (available on their github page):

"Similar to many other RNA-Seq based tools, RiboDiff uses negative binomial distribution to model the read count, which handles larger variation across samples than Poisson. However, if the randomness of count data from certain types of samples are extremely large, limited number of replicates cannot provide a good estimation on dispersion, which ends up with less significant results..." (<https://github.com/ratschlab/RiboDiff/blob/master/MANUAL>).

Negative Binomial has often performed better than the other commonly tested distributions. However, we believe that there may be other distributions that surpass NB. The most highly replicated RNA-seq dataset we are aware of was generated by Gierlinski *et al.*⁶, where a wild type and a *snf2Δ* yeast strain were sequenced 48 times each. The authors showed that NB fitted the data better than normal or lognormal distributions. To showcase our point, we removed the bad replicates per instructions of the original paper and tested the goodness-of-fit of the

data against a gamma distribution, and compared with NB. **Table R2** shows that gamma distribution fits this data better than NB. To be clear, we are not proposing TER testing based on gamma distribution, rather we are providing this observation in the RNA-seq world to highlight the notion that although NB-based methods have been generally successful in the past, alternative distributions and approaches can still be found to further improve our models of sequencing count data.

	Wild type		snf2Δ	
	Not rejected	Rejected	Not rejected	Rejected
Gamma (CVM)	4976	1113	5467	707
Gamma (AD)	4935	1154	5423	751
NB (Chi2)	4888	1202	5369	806

Table R2. Goodness-of-fit of Gamma and NB distributions for the 48-replicate yeast RNA-seq dataset. Values show the number of genes for which the null hypothesis of a given distribution is rejected (or not) based on the best fitted parameters and unadjusted p-values. CVM: Cramer-Von Mises test. AD: Anderson-Darling test. Chi2: Chi-square test.

Second, a linear modeling framework can directly estimate transcriptional and translational changes jointly without considering the "ratio of two NB variables" (Xiao & Yang, Nat Commun. 2016).

We thank the reviewer for this comment; and we agree with them that Xtail indeed provided a solution to this problem. In fact, Xtail has been one of our favorite tools in this space and we have previously published multiple studies using Xtail⁷⁻⁹. However, as stated above, the assumptions that the distribution parameters can be estimated effectively with 2 or 3 biological replicates is not a strong one. If the parameter estimates for a single NB distribution are imprecise, the ratio or joint distribution of two NB variables will be so too.

Our benchmarking results on the simulated dataset provided by the authors of Xtail show that Ribolog performs equally well in terms of ROC-, PR-AUC, and F1 measure. Yet Ribolog runs almost two orders of magnitude faster than Xtail and is much more robust to variations in sequencing coverage (https://github.com/goodarzilab/Ribolog/blob/master/benchmarks/ribolog_benchmarks.pdf). Furthermore, Ribolog provides analytical modules for stalling bias detection and correction, meta-analysis, uORF usage and translational readthrough (please see <https://github.com/goodarzilab/Ribolog>). And, by implementing a regression-based model allows the accommodation of complex experimental designs with multiple predictors, interaction terms, and batch effect (confounding factor) adjustment variables. This makes Ribolog more flexible and more widely applicable than alternative methods such as Xtail, RiboDiff and Riborex – which basically compare two-groups¹⁰.

Comment 2.3. In light of these points, and given the existence of software packages that can perform NB GLM analysis of matched Ribo-seq and RNA-seq samples such as Xtail (cited above) and others, it seems worth comparing ribolog results with established tools.

Response 2.3.

We thank the reviewer and we are in complete agreement with them. As mentioned above, our biological results here are not sensitive to the choice of model and tool (**Fig. R12**). We have also provided access to our benchmarks that justify the choice of Ribolog as our current

strategy for Ribo-seq data analysis.

Comment 2.4. Do Ribo-seq expression change measurements agree better with proteomics than RNA-seq? RNA-seq correlations ($R = 0.5$) and "logTER" correlations ($R = 0.3$) are reported, but not the uncorrected Ribo-seq, which should account for all mRNA changes and for translational changes as well, and thus correlate better with proteomics.

Response 2.4.

Overall, we agree with the reviewer that this is the expected behavior. With important caveats that proteomics data is skewed towards more abundant proteins and is strongly confounded by protein decay changes, which is not captured in Ribo-seq. To answer this question, we used DESeq2 to compare log fold changes for RPF and RNA-seq data separately. Using logFC from RPF counts, we observed a correlation coefficient of $R = 0.53$ ($CI = [0.51, 0.55]$) with mass-spec logFC. In comparison, we observed a correlation coefficient of $R = 0.46$ ($CI = [0.43, 0.48]$) for RNA-seq data. In other words, RPF data captures an additional 7% of variation in proteomic measurements. Consistent with this finding, Duncan and Mata¹¹ reported a moderately stronger correlation of protein levels with RPF ($R = 0.82$) than with RNA ($R = 0.68$) counts. Finally, in our initial manuscript we also mention that protein abundance measurements by mass spectrometry corrected my RNA-seq data and logTER values (i.e. RPF sequencing measurements corrected by RNA-seq data) correlate well ($R = 0.3$).

Comment 2.5. Several other RBPs — including TIA1 and ELAVL1/HuR — seem to have almost as much mutual information as HNRNPC (Supplementary Figure 1g).

Response 2.5.

We thank the reviewer for this comment. We performed TIA1 and ELAVL1 CLIP-seq in MDA-MB-231 cells and determined mRNA targets of these RNA binding proteins. As shown in **Fig. R14** and **Fig. S1k** in the revised manuscript, HNRNPC targets showed substantially higher translational repression (i.e. lower logTER values) in highly metastatic or HNRNPC-deficient cells than TIA1 or ELAVL1 targets.

Figure R14. Cumulative density plots of translation efficiency ratios (logTER) comparing MDA-LM2 and MDA-MB-231 (left) or sgHNRNPC and sgControl (right) cells, for HNRNPC, TIA1 and ELAVL1 target transcripts, as determined by CLIP-seq.

Comment 2.6. It is claimed that, "HNRNPC controls the alternative polyadenylation of its targets" (p. 5, l. 43). However, it would seem that an alternative explanation could be provided by nuclear export, which controls apparent TE (since the nuclear pool of an RNA is untranslated but shows up in mRNA-Seq) and can certainly be influenced by 3' UTRs and by nuclear RBPs. How is it excluded that HNRNPC promotes nuclear export and subsequent degradation of long-3'UTR variants?

Response 2.6.

To test this alternative possibility, we performed 3'-end RNA-seq from the cytoplasmic and nuclear fraction, in highly and poorly metastatic, as well as control and HNRNPC KD MDA-MB-231 cells. As described above, and as shown in cumulative density plots in **Fig. R15** (and Fig. S3h-i in the revised manuscript), we detected no significant difference between HNRNPC target

and non-target RNAs when comparing nuclear and cytoplasmic fractions for alternative polyadenylation (logAPAR) changes in highly metastatic or HNRNPC-depleted cells.

Figure R15. Cumulative density plot of log fold-change between nuclear and cytoplasmic fractions, comparing alternative polyadenylation ratio (logAPAR) between (top panel) MDA-LM2 and MDA-MB-231 cells, or (bottom panel) sgControl and sgHNRNPC cells, for HNRNPC 3' UTR target and non-target mRNAs. Median difference (ΔM) and p value (calculated using Mann-Whitney U -test) are shown.

Comment 2.7. HNRNPC and PABPC4 are both pretty well-expressed and broad mRNA-binding proteins, and so an RNA-dependent interaction between them (Figure 3) is not surprising.

Response 2.7.

We agree with the reviewer on this point. However, our data (HNRNPC and PABPC4 CLIP-seq and genetic epistasis experiments) suggests that this interaction is functional in controlling poly(A) site selection. Moreover, as outlined above, our additional data shows that other well-

expressed and broad mRNA-binding proteins, such as TIA1 and ELAVL1, despite sharing the poly(U) binding preference with HNRNPC, do not contribute the same way to the translational repression in highly metastatic cells.

Comment 2.8. In the comparison of PABPC4 binding and HNRNPC binding (Figure 3c), what is the total universe of 3' UTRs assessed for binding with both proteins?

Response 2.8.

The total number of transcripts in this comparison is 17,857 (APA genes), we have added this information to Fig. 3c in the revised manuscript.

Comment 2.9. In Figure 4a, it seems that almost all genes (~16,456 / 17,857 I think?) show AGO2 CLIP? Is the conclusion that >90% of mRNAs experience microRNA-mediated repression?

Response 2.9.

We thank the reviewer for this remark, the Fig. 4a in the original manuscript incorrectly reported the number of AGO2 targets due to a typographical error. We corrected it in the revised manuscript, the total number of AGO2 targets is 4,945 out of 17,857 genes showing APA events.

Comment 2.10. Is the accelerated growth of HNRNPC (Figure 5) and PABPC4 (Supplementary Figure 5) knock-down cells specific to the lung metastasis environment, or do these knock-downs increase growth in culture or in localized xenografts as well?

Response 2.10.

As described above (**Fig. R8**), our additional data show that HNRNPC KD increases cell migratory and invasive capacity of MDA-MB-231 cells, which is in agreement with our observed *in vivo* phenotypes. This is despite the fact that HNRNPC depletion reduced proliferation and colony formation rate *in vitro*, as well as a slightly lower primary tumor growth rate *in vivo* (orthotopic injection model). Therefore, the metastasis phenotype cannot be explained by increased rate of growth and proliferation.

Comment 2.11. How many transcripts matched the four criteria used to select PDLIM5 (Figure 6a)?

Response 2.11.

We thank the reviewer for this question. We overlapped the strongly translationally repressed transcripts common in highly metastatic and HNRNPC-deficient cells (N = 16), and transcripts with longer 3' UTR in these conditions (N = 30), which identified PDLIM5 as a single prospective target (at the chosen effect size and significance thresholds). It also was in the list of common HNRNPC and PABPC4 binding targets (N = 1263), and was significantly associated with survival metrics in breast cancer patient cohorts. To help clarify this, we included this information to the main text and removed the Fig. 6a in the initial manuscript.

Comment 2.12. As a more minor point, "SH3" is "Src homology 3" not "Scr homology 3" (p. 8, l. 25).

Response 2.12.

We thank the reviewer for this remark, it is now corrected in the revised manuscript.

Reviewer #3:

Remarks to the Author:

SUMMARY:

Using ribosome profiling, the authors have performed a transcriptome-wide analysis of translation in parental MDA-231 cells and their metastatic derivatives, as well as PDXs. They identify a subset of mRNAs that are translationally suppressed and harbor binding of HNRNPC and PABPC4, which together coordinate alternative polyadenylation of these transcripts. In metastatic cells, the RBPs are downregulated leading to the selection of downstream polyA signals and polyA lengthening leading to translational repression that is AGO2-dependent. Furthermore, KD of HNRNPC enhances metastasis in vivo, and loss of HNRNPC is associated with poor patient survival. The authors also introduce their new logistic-regression-based methods for analysis of ribosome profiling and alternative polyadenylation data, Ribolog and APAllog.

Roles for HNRNPC and PABPC4 in alternative polyadenylation and metastasis have been explored previously. However, the particular mechanism of translational control described herein is novel. However, the rigor of the informatic approaches should be improved, and the metastasis assays should be refined.

We thank the reviewer for their thoughtful comments, which have helped improve our manuscript, and we hope we have addressed their concerns below.

Comment. Specifically, since many of the changes and differences presented in this manuscript are modest in magnitude/significance, it is essential to have sufficient confidence in the quality of the ribosome profiling data, and the validity of the method of analysis. In the absence of peer-review and benchmarking for the new software suite presented herein, it is critical to include PCA plots and p-value/FDR distributions for all data sets. Validation of the results using published benchmarked methods (RUST, RiboDiff, Anotas2seq, etc.) would also be preferable, although the authors may wish to do this in a separate manuscript focusing on the Ribolog software package prior to publication of these findings. Finally, the functional assays should actually look more specifically at metastasis, and the mechanistic links with PDLIM5 must be refined and better validated.

Response

We thank the reviewer for raising this point. We agree that biological results should ideally be robust to the choice of justifiable analytical models. First, we should emphasize that the key insights derived from our Ribo-seq data is based on gene-set enrichment analysis of differential translation efficiency values (TER estimates, not their test p-values). Therefore, our results remain largely identical across different tools. Below, we have included the comparison of logTER values for alternatively poly-adenylated genes that are bound or unbound by HNRNPC

(Fig. S1i and S2a in the revised manuscript) with Xtail, Riborex, and RiboDiff as well (**Fig. R16**). Furthermore, while the median effect of post-transcriptional regulatory pathways on their individual targets is modest, the regulon effect observed here is highly stable, robust, and significant.

Figure R16. Cumulative density plot of translation efficiency ratios (logTER) comparing MDA-LM2 to MDA-MB-231 (Mda-Par, top) or sgHNRNPC to sgControl MDA-MB-231 (bottom) cells, for HNRNPC 3' UTR target and non-target mRNAs, using Xtail (left), Riborex (center) or Ribodiff (right). Median difference (ΔM) and p value (calculated using Mann-Whitney U -test) are shown.

To further emphasize the similarity of logTER values from different packages, we have included the pair-wise Pearson correlation matrix of logTER values calculated using Ribolog, Xtail, RiboDiff, and Riborex. These results, which were generated on simulated counts from the Xtail package, shows almost identical logTER values across tools, except for Anota2seq (**Table R3**).

	Ribolog	Xtail	Riborex	Ribodiff
Ribolog				
Xtail	0.999			
Riborex	0.998	1.000		
Ribodiff	1.000	0.999	0.998	

Table R3. Pairwise Pearson correlation matrix of logTER values generated on simulated counts from the Xtail package, obtained with Ribolog, Xtail, Riborex, and Ribodiff.

Third, Ribolog allows for correction of local heterogeneity (stalling) biases in Ribo-seq data by implementing a CELP module. Both simulated data and real-world examples indicate that this correction improves logTER estimates. We found that after applying the CELP correction, residuals calculated as the difference between Ribolog and Xtail logTER values are significantly correlated with changes in protein levels (**Fig. R17**). This observation establishes that starting from the same data, Ribolog captures the protein landscape better.

Figure R17. Regression analysis comparing the protein logFC in MDA-LM2 versus MDA-MB-231 cells to the residual logTER from the values calculated using Xtail subtracted from those calculated using Ribolog. Correlation coefficient and the resulting p -value are shown.

Finally, as the reviewer mentioned, we are in the process of writing up a manuscript to be published shortly which will include expanded benchmarking results along with more theoretical discussion, a full description of module functionalities and use case scenarios. We have nevertheless made the results of our benchmarking tests available online at (https://github.com/goodarzilab/Ribolog/blob/master/benchmarks/ribolog_benchmarks.pdf). These benchmarks establish Ribolog as a strong contender with (i) the highest ROC- and PR-AUC, (ii) one of the lowest CPU times, and (iii) incredibly robust to changes in sequencing depth. Furthermore, contrary to most alternative tools such as Xtail, RiboDiff and Riborex which basically compare two-groups¹⁰, Ribolog can accommodate complex experimental designs with multiple predictors, interaction terms, and batch effect (confounding factor) adjustment variables. Notably, we have tested our method using the datasets simulated by the authors of Xtail with its underlying assumptions. Our observations indicate that Ribolog outperforms existing tools on several key metrics. It should also be highlighted that some components of Ribolog have been already published in peer reviewed publications^{8,9,12}.

With regards to data quality, Ribolog in fact provides an extensive module devoted to QC. As included in the revised manuscript, we have now repeated the key experiments twice, which resulted in identical results. We have also included the PCA plots for our Ribo-seq datasets below (**Fig. R18**).

Figure R18. (a) Principal component analysis (PCA) of Ribo-seq and RNA-seq data in biological triplicates, in MDA- MB-231 and MDA-LM2 cells. **(b)** PCA of Ribo-seq and RNA-seq data in biological triplicates, in sgControl or sgHNRNPC-expressing MDA-MB-231 cells.

COMMENTS AND SUGGESTED IMPROVEMENTS

Comment 3.1. Page 4, line 14-15

The authors claim that their new method Ribolog has advantages to existing methods. This may be the case, however, their method has not undergone peer-review or systematic

benchmarking. For this reason, it is not possible to determine if it performs better or worse than the existing standards. Such comparisons, or a paper that does this, should be included to support the method.

Response 3.1.

Please refer to the previous comment for the response to this point. In sum, (i) we have clarified that our results are not sensitive to the choice analytical tools, (ii) we have released a benchmarking document for Ribolog that systematically compares a number of common Ribo-seq packages head-to-head, and (iii) some components of Ribolog have been previously published.

Comment 3.2. Page 5, lines 10-11, page 6 line 2-3

The RNAseq has only two replicates, so commenting on significance is likely not appropriate. Another replicate should be added. Moreover, the authors should show a dot plot to assess the variability between replicates.

Response 3.2.

We thank the reviewer for these suggestions. First, we have now repeated all experiments in biological triplicates, and as shown in the PCA plots below (**Fig. R19**), these replicates are very consistent ($R \geq 0.99$ between all pair-wise replicates across all datasets produced). As commented above, our new results are entirely consistent with our initial observations.

Figure R19. (a) Principal component analysis (PCA) of Ribo-seq and RNA-seq data in biological triplicates, in MDA-MB-231 and MDA-LM2 cells. **(b)** PCA of Ribo-seq and RNA-seq data in biological triplicates, in sgControl or sgHHRNPC-expressing MDA-MB-231 cells.

Comment 3.3. Page 5, lines 25-27, Figure S2b

Perhaps the figure has been misunderstood, however, the correlation coefficients between translationally repressed mRNAs in MDA-LM2 and HNRNPC KD cells appear very low, which seems to suggest that the data are actually quite poorly correlated. Thinking in terms of variance explained, with a coefficient of 0.05, this would be 0.25%. This is either quite unconvincing, or overly confusing. If the later, it may be helpful to visualize the overlap by overlaying the significantly translationally suppressed mRNAs from one cell line onto the volcano plot for the other by coloring these points.

Response 3.3.

We thank the reviewer for this comment. The correlation coefficients comparing **general** changes in translation efficiency (logTER) or alternative polyadenylation (logAPAR) between highly and poorly metastatic and HNRNPC KD and control cells are indeed minimal. In fact, we do not expect the HNRNPC knockdown to recapitulate the ensemble of translational or alternative polyadenylation changes occurring during the metastatic progression in breast cancer. To support this, we and others have found a variety of other pathways contributing to the translation control in metastasis¹³. This study thus highlights that HNRNPC is a part of an

ensemble of mechanisms, emphasizing a new link of HNRNPC deficiency with alternative polyadenylation and translational control in breast cancer metastasis. With this in mind, Fig. S2b and S2e in the original manuscript illustrate that while (and despite) **general** correlation between the two conditions is minimal, the HNRNPC **target** RNAs are enriched among translationally repressed or longer 3' UTR-bearing transcripts in **both** conditions (a red-shift in the two-dimensional density heatmaps). Similarly, the HNRNPC regulon is depleted among translationally activated or shorter 3' UTR-bearing transcripts in both conditions (a blue-shift in the heatmaps). Such data visualization allows us to demonstrate that **the same HNRNPC targets** are translationally changing between the two datasets by comparing the continuous variables in both datasets, instead of applying an arbitrary cut off values to one data set and project it on the other¹⁴.

Comment 3.4. Page 7, Figure S4a-b

The subset of translationally suppressed HNRNPC targets that are not AGO2/miRNA targets are not shown here. This seems like an essential comparison if the claim is that the effect on translation is mediated by miRNA targeting.

Response 3.4.

We thank the reviewer for this question. As shown in **Fig. R20**, HNRNPC targets that are also miRNA targets show significantly lower logTER values than HNRNPC but not miRNA targets, miRNA but not HNRNPC targets, or the remaining alternative 3' UTR mRNAs.

HNRNPC+mi
RNA HNR
NPC mi
R Alternative
UTRs

Figure R20. Violin plots showing the distribution of translation efficiency ratios (logTER) comparing MDA-LM2 and MDA-MB-231 cells, stratified by HNRNPC and miRNA target groups.

Comment 3.5. Page 8, Figure S6b

Again, with an R of 0.140, this means that 1.96% of the variation is explained by HNRNPC activity. This suggests the effect may be for a relatively small subset of genes, but the wording here seems to imply that this is a more generalized effect.

Response 3.5.

Similar to our response to the previous comment on the two-dimensional heatmaps in Fig. S2b and S2e, we do not claim that the entirety of translational dysregulation occurring during breast cancer metastasis to lungs is due to the HNRNPC-mediated control. In fact, us and others have identified several parallel mechanisms that act on the translational control in metastasis¹³. As illustrated in our data, downregulating PDLIM5 is sufficient to increase metastatic lung

colonization by several triple receptor negative breast cancer cell lines *in vivo*. PDLIM5 is the target downstream of HNRNPC that we chose to characterize in more details within this study; however, additional factors impacted by HNRNPC deficiency likely act in promoting breast cancer metastasis, supported by the clinical data showed in Fig. S7f-g.

Comment 3.6. Page 8, Figure 6a

The Venn diagram seems to be incomplete as the numbers of genes are not shown for the various groups. Additionally, the size of the circles should reflect the gene numbers. It is also not clear from the text what the “clinical association” dataset is, and where it came from.

Response 3.6.

We thank the reviewer for these suggestions. We now included modified Venn diagrams in the revised manuscript. For Fig. 6a in the initial manuscript, we overlapped the translationally repressed transcripts common in highly metastatic and HNRNPC-deficient cells (N = 16), and transcripts with longer 3' UTR in these conditions (N = 30), which identified PDLIM5 as a single prospective target (at the chosen effect size and significance thresholds). It also was in the list of common HNRNPC and PABPC4 binding targets (N = 1263), and was significantly associated with survival metrics in breast cancer patient cohorts. We transferred this information to the main text and removed the Fig. 6a in the initial manuscript.

Comment 3.7. Page 19, TER test

If it is understood correctly, the estimation of translation efficiency is based on an odds ratio obtained by pooling all reads (both RFP and RNA) for a given gene, and then sampling reads from this pool. If this is the case, then there seems to be a high danger of masking of outliers/outlier dominance using this method, since all biological variability is lost. For example, if reads from one replicate are considerably higher for a particular gene than in others, this information is lost when pooling but will of course still affect the rate of sampling. Furthermore, since it is not possible to determine outliers with only 2 biological replicates, the potential for false positives seems very high.

Response 3.7.

The reviewer's description of the Ribolog TER test is correct. Regarding outliers: They are a risk to any method, not just ours. Read-level QC and removing outliers should be done prior to TER analysis using any approach. There are several packages that provide tools for this purpose for example by checking the correlation of RNA and RPF reads across replicates and identifying the outliers; we have not duplicated those steps here. Instead, we provide an entire module with Ribo-seq-specific QC tools including PCA (on normalized RNA counts, normalized RPF counts

and TE), proportion of null features (non-differentially translated transcripts) and correlograms of equivalent TER tests in replicated datasets.

In addition to QC on the input, Ribolog provides the user with an option to investigate replicate variability in terms of the TER test output. To the best of our knowledge, no other package offers this. Because reads are the units of observation in our model, TER test can be done with one replicate per group – in which case the logistic regression is virtually reduced to a 2x2 contingency table. In a replicated dataset, Ribolog allows running the TER test between all possible combinations of reps belonging to the groups being compared. For example, if there are four samples A1, A2, B1 and B2, a function in Ribolog calculates TER results for A1xB1, A1xB2, A2xB1 and A2xB2 pairs. Then, it calculates and plots the correlation of vectors of z scores to check the consistency of the results of these “equivalent tests”. The interested user can combine these z vectors side-by-side into a matrix and calculate row-wise variance or coefficient of variation (CV) of the z scores. Transcripts with high variance or CV can be flagged

for outlier behavior and investigated further. Ribolog also calculates the proportion of null features (non-differentially translated transcripts) in each of these tests using the distribution of p-values. It is even possible to run the tests between replicates of the same group, where we expect a much higher proportion of null features compared with tests done across groups. This allows the user to remove poor or strange-acting replicates and transcript and rerun the test. Finally, Ribolog provides a meta-analytic framework to combine the results of these correlated or equivalent rep-by-rep tests to produce test statistics that incorporate replicate variability. This is an alternative to the default TER test where -as the reviewer correctly stated – replicates are effectively pooled. Based on the distribution of p-values from the two modes of the test, we have favored the default (pooling) method and analyzed this data with it.

In sum: (i) Outliers affect both count-based methods (e.g. Xtail) and read-based methods (e.g. Ribolog). If an outlier is not removed, it could inflate the within-group variance and produce a false negative result in a count-based model. Our method may indeed be more robust to outliers [need benchmark to confirm] because the effect is mitigated via pooling with “good” replicates. (ii) During pre-processing, we normalize read counts (separately for RNA and RPF) using the median-of-ratios method¹⁵ which partially reduces outlier effects. (iii) If the outlier behavior manifests itself as unusually high local densities of RPF reads in some transcripts and samples – whether due to a biological reason or a technological artifact – the Ribolog CELP module mitigates the problem, even without the researcher having to explicitly specify the cause. We have simulated a scenario where a certain codon is decoded 10 times more slowly in a group of samples. Uncorrected, this leads to a large number of overestimated TERs with both Ribolog and Xtail. Applying the CELP correction – which is completely oblivious to the source of bias and the samples affected by it – significantly reduces this problem (https://github.com/goodarzilab/Ribolog/blob/master/benchmarks/ribolog_benchmarks.pdf). We surmise that the CELP correction should be able to mitigate other patterns of bias and outlier behavior to a degree as well.

Comment 3.8. In terms of the quality of the ribosome profiling datasets, this is not adequately address. In Fig 2e there are a very large number of genes that have large fold changes but do not reach any threshold of significance, even considering p-values and not FDRs were used. This would suggest that there are substantial issues with the reproducibility of the data. In light of this, it is essential to provide PCA plots for all the various data sets presented herein, as well as p-value and FDR distributions for the outputs of all Ribolog TER test analyses.

Response 3.8.

We thank the reviewer for these suggestions. As mentioned above, we repeated the key Ribo-Seq and 3'-end RNA-seq experiments (in MDA-MB-231 and MDA-LM2, as well as HNRNPC KD and control cells) in biological triplicates, with PCA plots illustrated in **Fig. R19**. As commented above, our new results largely support our initial observations.

We should note that Fig. 2e pointed to by the reviewer is not Ribo-seq data, but rather 3'-end RNA-seq used for comparison of APA. The observed behavior was due to the fact that there are

a large number of annotated poly-A sites in the genome, the vast majority of which are rare events and seen at extremely low counts. We have now added a filtering step to our analysis that removes pA sites that are largely unsupported by the data. We also repeated this experiment with three biological replicates and deeper sequencing to further boost the observed counts. These results, included in **Fig. R4** and Fig. S3j-k in the revised manuscript, show a consistent volcano plot. Additionally, our biological findings are not contingent on the approach used for APAR significance testing as our analyses use the ranks of logAPAR values, similarly to what has been described above for logTER.

Comment 3.9. The Ribolog method is not the topic of the paper. However, it is difficult to claim its superiority over other published methods that have undergone peer-review and systematic benchmarking. In the absence of peer-review and benchmarking for Ribolog, it would be advisable to also demonstrate the quality of the data presented here using an established tool like RUST (O'Connor 2016). In addition, confidence in the validity of the results could be increased if similar results were obtained using published methods such as RiboDiff (Zhong 2017) or Anota2seq (Oertlin 2019).

Response 3.9.

We thank the reviewer for raising this point. We have now provided access to our Ribolog benchmarks against the most commonly used tools in the field (https://github.com/goodarzilab/Ribolog/blob/master/benchmarks/ribolog_benchmarks.pdf). And as mentioned above, we have in fact tested the sensitivity of our analysis to the choice of the tool used, and in no case did we see a notable change in the effect size, direction, or statistical significance of our observations.

It should also be noted that, in a broad sense, RUST is comparable to our CELP module, which correctly models the change in codon decoding rates, demonstrated in three peer reviewed publications^{8,9,12}. However, to the best of our knowledge, RUST compares decoding rates among codons, or if generalized, other “known” transcriptome-wide bias factors, whereas CELP calculates and corrects local density bias without requiring the user to prespecify the cause. The CELP bias coefficients or summary statistics derived from them can be subsequently regressed against any transcriptome-wide or transcript-specific predictors such as codon type, amino acid type, amino acid biochemical group, proximity to start or stop codon, RBP binding sites, secondary structures, transcript length, normalized transcript expression level, etc. to infer the underlying cause of local RPF read density heterogeneity.

Comment 3.10. Volcano plots

Due to multiple-testing, the y-axis on all volcano plots presented should reflect the FDR rather than the p-value.

In general, coloring the HNRNPC targets, or mRNA in the proposed HNRNPC/PABPC4 regulon on the scatter plots would be helpful in seeing the specificity of the effect. As the data is currently presented, it is not possible to see how the same gene is regulated between two plots. The interpretability of the results would be greatly improved by being able to see how this same set of genes behaves across the different KD conditions.

Response 3.10.

We thank the reviewer for this comment. As outlined earlier, Ribolog/APAlog estimate multiple parameters, including the effect size (logTER or logAPAR) and associated significance measurements (p -value and FDR). Importantly, while we report effect size and p -value in our volcano plots, our principal findings are not dependent on the significance measurements (such as the enrichment of HNRNPC regulon in translationally repressed mRNAs or transcripts with longer 3' UTRs, etc.). To specifically study the relation between the affected transcripts in multiple conditions, we employ two-dimensional heatmaps, as in Fig. S2b and S2e.

Comment 3.11. CRISPRi

A number of CRISPRi KDs were generated in this study, but no validation was provided to confirm that the KDs were successful. This should at least be shown in a supplement.

Response 3.11.

We thank the reviewer for this remark. We now included the knockdown and overexpression efficiencies in the revised manuscript.

Comment 3.12. Motifs

Although scrambled sequences are used by some, even with maintenance of kmers this method is more prone to underestimating the rate of false positives than using the true sequences for a background set.

Response 3.12.

We thank the reviewer for this remark. In Fig. 1a and 1b, where the FIRE algorithm⁴ operates in the motif discovery mode, the background set for enrichment calculations corresponds to all outside of the bin transcripts, that is, true sequences. In Fig. 1c, 2b, and S3e, the algorithm operates in validation mode, where an occurrence of a given motif is calculated within experimentally identified (CLIP-seq) RNA binding protein target sequences. In this case, it is challenging to come up with a non-target true sequence background set that would account for all the biases encountered during CLIP-seq library preparation and sequencing. We thus use k-mer controlled scrambled sequences as background set.

Comment 3.13. Fig 1d

MI in bits and z-score are likely not very meaningful metrics for most. Fig S1h is more compelling, and a similar cdf plot should be generated for the PDXs.

Response 3.13.

We now provide a cumulative density plot for this dataset (**Fig. R21**, Fig. S1j in the revised manuscript). However, we should note that at its core, this is a gene-set enrichment analysis. And gene-set enrichment methods, all variations of them including ours, are designed to better capture the non-random patterns of distribution across the spectrum of data.

0.0

-1 0 1

Highly vs. poorly metastatic PDXs (logTER)

Figure R21. Cumulative density plot of translation efficiency ratios (logTER) comparing highly and poorly metastatic breast cancer PDXs, for HNRNPC 3' UTR target and non-target mRNAs. Median difference (ΔM) and p value (calculated using Mann-Whitney U -test) are shown.

Comment 3.14. Fig S3a

P-values are presented for the enrichment analyses. An adjusted p-value or FDRs is appropriate to report here. Furthermore, the accompanying heatmap seems misleading considering the lack of significance for the mRNA export from the nucleus GO term.

Response 3.14.

We thank the reviewer for this remark. Indeed, the iPAGE algorithm⁴ reports FDR-adjusted p -value, indicated in Fig. S3a in the initial manuscript; both GO terms show $p < 0.05$.

Comment 3.15. Fig S6e-f

Quantifying all biological replicates from the same gel somewhat defeats the purpose of replicates in terms of controlling for assay variability. It also looks like there may have been a technical issue with the transfer for this membrane.

Response 3.15.

We thank the reviewer for this observation, we repeated the western blot in biological triplicates and replaced the image in the revised manuscript (**Fig. R22**, Fig. 6b in the revised manuscript).

Figure R22. Cell lysates from MDA-MB-231, MDA-LM2, and MDA-MB-231 cells expressing sgControl or sgHNRNPC were analyzed by western blotting in biological triplicates. Relative normalized (to GAPDH) protein quantities and standard errors are indicated below the image.

Comment 3.16. PDX models should be characterized to a greater extent

In the PDX samples (shown in 1b) there are far more transcripts with reduced translational machinery in the metastatic versus poorly metastatic tumours. This suggests a phenomenon, such as hypoxia, may be associated with the metastatic tumours. These PDX models should thus be better characterized before suggesting that HNRNPC is driving differential effects in these models. Indeed, associations seem much more modest in these models.

Response 3.16.

We thank the reviewer for raising these points. The PDX models used in our study have been recently described in details¹⁶. It should be emphasized that unlike the MDA-LM2 vs MDA-Par model, which is a matched isogenic comparison, the highly and poorly metastatic PDXs are **not** matched. Given that the inter-patient heterogeneity is expected, it is not surprising that the effect sizes are smaller and the signal is slightly diluted. The observation that the HNRNPC regulon shows a significant signal in cell line models, PDXs, and clinical samples, makes this a highly consistent signal.

Comment 3.17. Models used do not fully test metastasis

The tail vein model is not a metastasis assay, but rather measures extravasation and survival and growth in the lung. The authors, should thus employ spontaneous metastasis models to

validate their hypothesis. Given the availability of a PDX models, these should be used throughout, particularly for T4 studies. These latter studies should also be accompanied by a time course to better understand the kinetics of translational change and the extent to which these persist 6 hours after exposure and beyond.

Response 3.17.

We thank the reviewer for raising this point. While lung colonization assays side-step intravasation, it is long established that intravasation is not a major bottleneck in metastasis (hence the presence of CTCs even in early stage tumors). However, as recommended by the reviewer, we also performed orthotopic (mammary fat pad) injections of control or HNRNPC knockdown cells in NSG mice, monitored primary tumor growth, resected it 4 weeks post injection, and then followed the lung colonization. As shown in **Fig. R23** (and Fig. 5c-d in the revised manuscript), while the primary tumor volume was slightly but not significantly decreased in the HNRNPC-deficient cell cohort, HNRNPC knockdown cells showed a marked and significant increase in their metastatic potential.

Figure R23 (a) MDA-MB-231 cells stably expressing sgControl or sgHNRNPC were injected orthotopically (mammary fat pad) into NSG mice. Four weeks after injection the primary tumor was resected and lung bioluminescence was measured at the experiment end-point. Lung sections were stained with H&E (representative images shown). $n = 4-5$ mice per cohort. (b) Tumor volume of the orthotopically injected sgControl or sgHNRNPC cells was measured at Day. 28. $n = 7-10$ mice per cohort. P value calculated using one-tailed Mann-Whitney U -test.

We also addressed the details of the T4-mediated inhibition of the lung metastatic colonization by MDA-LM2 cells. As described in our initial manuscript, the 6-hour T4 treatment with the IC20 concentration of MDA-LM2 cells *in vitro* markedly reduced its lung colonization capacity *in vivo*. To gain more insights into the poly(A) site selection kinetics caused by T4, we sampled the MDA-LM2 cells before and after the 6-hour treatment, as well as 24 and 72 hours post 6h treatment. We then performed 3'-end RNA-seq and calculated the changes in alternative polyadenylation using APAlgo. As shown in **Fig. R24a** (and Fig. S7o in the revised manuscript), both 6h and 24h post-T4 treatment HNRNPC target mRNAs showed 3' UTR shortening (positive logAPAR values). The effect remained significant up to 72h post treatment, albeit to a lesser extent. Since cancer cell invasion, which is an early event, is the mechanism of metastasis in this case, even a transient effect is sufficient to cause a lasting difference between the two cohorts.

Finally, to support our data from the *in vitro* T4-treated MDA-LM2 cells, we assessed the impact of the systemic T4 treatment on the lung colonization by MDA-LM2 cells *in vivo*. We injected NSG mice intraperitoneally with T4 or vehicle control for three consecutive days¹⁷ starting on the day of cancer cell injection. As shown in **Fig. R24b** (and Fig. 7j in the revised manuscript), the systemic T4 treatment significantly reduced lung metastatic burden as compared to vehicle control, supporting our finding that modulating the alternative polyadenylation program could be a therapeutic strategy in metastatic breast cancer.

Figure R24. (a) Box plots illustrating the changes in alternative polyadenylation (logAPAR) of HNRNPC targets between T4-treated (sampled at indicated time points) and untreated MDA-LM2 cells. *P* values calculated using Wilcoxon signed-rank test. (b) MDA-LM2 cells were injected via tail vein into NSG mice. On the day of cancer cell injection and for two additional consecutive days, animals were injected with 10 mg/kg T4 or an equivalent amount of vehicle control. Bioluminescence was measured at the indicated times (*p* value calculated using two-way ANOVA); area under the curve was measured at the final time point (*p* value calculated using one-tailed Mann-Whitney *U*-test). *n* = 4-5 mice per cohort.

Comment 3.18. Work related to PDLIM5 must be greatly improved

Work related to PDLIM5 (from protein expression to results in animals) is not that convincing. Different translational efficiencies of the isoforms should be determined and mechanisms by which PDLIM5 may affect metastasis should be determined. Moreover, the authors would need to determine the extent to which PDLIM5 overexpression may reverse the effects of HNRNPC knock down. Finally, linkages with AGO2/miRNA should be assessed with cause-effect experimentation.

Response 3.18.

We thank the reviewer for these suggestions. In our initial manuscript, we showed that PDLIM5 mRNA showed reduced translation efficiency (TE) in highly metastatic and HNRNPC KD cells (Ribo-seq data), PDLIM5 protein was reduced in quantity in these conditions (TMT-MS and western blotting data) (Fig. S6d and Fig. 6b in the revised manuscript), and PDLIM5 mRNA underwent 3' UTR lengthening in these conditions (3'-end RNA-seq and RTqPCR data) (Fig. 6a and Fig. 6c in the revised manuscript). Using experimental metastasis assays in mice we showed that PDLIM5 downregulation increased the lung colonization by MDA-MB-231 cells *in vivo*, as compared to control cells (Fig. 6e in the revised manuscript). We additionally performed experimental metastasis assays with control and PDLIM5 KD HCC1806 cells, which is another commonly used triple receptor negative breast cancer cell line. Just as in MDA-MB-231 cells, the PDLIM5 KD led to increased lung colonization in HCC1806 cells (**Fig. R25**, and Fig. S6g in the revised manuscript).

Figure R25. HCC1806-LM2 cells expressing shControl or sgPDLIM5 were injected via tail vein into NSG mice. Bioluminescence was measured at the indicated times (p value calculated using two-way ANOVA); area under the curve was measured at the final time point (p value calculated using one-tailed Mann-Whitney U -test). $n = 4$ -5 mice per cohort.

To provide additional evidence that PDLIM5 acts downstream of HNRNPC, we first depleted HNRNPC in MDA-MD-231 cells, and then stably expressed mCherry or PDLIM5 from the EF1 α promoter in these cells. As shown in **Fig. R26** (and Fig. 6f in the revised manuscript), PDLIM5 overexpression in HNRNPC-deficient cells brought down the metastatic potential *in vivo* to the level of control cells with intact HNRNPC expression.

Figure R26. MDA-MB-231 cells stably expressing sgControl or sgHNRNPC and mCherry (Control) or PDLIM5 (PDLIM5-OE) were injected via tail vein into NSG mice. Bioluminescence was measured at the final time point (p value calculated using one-tailed Mann-Whitney U -test). $n = 4$ -5 mice per cohort.

To better understand the mechanisms how the HNRNPC-PDLIM5 axis impacts metastasis, we determined proliferation rates, colony formation potential, and migration/matrigel invasion capacity of cell lines with depleted HNRNPC and PDLIM5, as well as HNRNPC overexpression. In agreement with our lung colonization assays *in vivo*, HNRNPC and PDLIM5 knockdowns caused an increase, while HNRNPC OE—a decrease, in cell migration and invasive capacity *in vitro* (**Fig. R27a** and Fig. S6h-l in the revised manuscript). These observations contrasted with the reduced proliferation and colony formation of HNRNPC-perturbed cells (**Fig. R27b-c** and Fig. S6j-l), consistently with primary tumor growth rates of HNRNPC KD cells *in vivo* (Fig. 5c in the revised manuscript). Similar to our *in vivo* data (Fig. 6f in the revised manuscript), ectopic PDLIM5 expression rescued the proliferation and colony formation defects of HNRNPC KD cells (**Fig. R27b-c** and Fig. S6m in the revised manuscript).

Figure R27. (a) Cell migration and cell invasion measurements in MDA-MB-231 cells expressing indicated sgRNAs

and/or overexpression constructs. Representative images are shown next to the bar plots. *P* values calculated using two-tailed unpaired *t*-test. **(b)** Proliferation rate measurements in MDA-MB-231 cells expressing indicated sgRNAs and/or overexpression (OE) constructs. **(c)** Colony formation unit (CFU) measurements in MDA-MB-231 cells expressing indicated sgRNAs and/or overexpression constructs. Representative images are shown below the bar plots.

Finally, to show that the PDLIM5 expression is dependent on its APA site choice, we constructed a reporter, consisting of N-terminally FLAG-tagged PDLIM5 CDS and three versions of 3' trailers: (i) an entire region encompassing both proximal and distal PDLIM5 poly(A) sites (hg38 chr4:94,664,164-94,666,887), labeled *full* 3' UTR (which is processed to both short and long isoforms); (ii) a region only up to the proximal poly(A) site (hg38 chr4:94,664,164-94,665,481), labeled *short* 3' UTR; (iii) and a region between the proximal and distal poly(A) sites (hg38 chr4:94,665,535-94,666,887), labeled *long* 3' UTR. We then delivered the three reporters to MDA-MB-231 cells by lentiviral transduction and Blasticidin selection, and determined the reporter protein expression by western blotting. As shown in **Fig. R28** (and Fig. 6d and Fig. S6f in the revised manuscript), removing the PDLIM5 3' UTR region beyond the proximal poly(A) site slightly increased the normalized FLAG-PDLIM5 abundance as compared to full 3' UTR. More importantly however, leaving only the PDLIM5 3' UTR region beyond the proximal poly(A) site substantially reduced the reporter protein abundance, confirming that the *cis* regulatory elements present between the proximal and distal poly(A) sites negatively control the PDLIM5 production.

Figure R28. (a) Schematics of the PDLIM5 3' UTR reporters. (b) The reporters illustrated in (a) were stably delivered to MDA-MB-231 cells and the reporter protein quantity was measured by western blotting in cell lysates (N = 6; representative image in biological duplicates is shown). (c) Relative normalized reporter protein quantity (FLAG signal normalized to GAPDH, relative to PDLIM5 full 3' UTR) is shown; error bars correspond to standard errors, *p* values calculated using two-tailed unpaired *t*-test.

References

1. Fish, L. *et al.* Nuclear TARBP2 Drives Oncogenic Dysregulation of RNA Splicing and Decay. *Mol. Cell* **75**, 967–981.e9 (2019).
2. McGlincy, N. J. & Ingolia, N. T. Transcriptome-wide measurement of translation by ribosome profiling. *Methods* **126**, 112–129 (2017).
3. Mayer, A. *et al.* Native Elongating Transcript Sequencing Reveals Human Transcriptional Activity at Nucleotide Resolution. *Cell* **161**, 541–554 (2015).
4. Goodarzi, H., Elemento, O. & Tavazoie, S. Revealing global regulatory perturbations across human cancers. *Mol. Cell* **36**, 900–911 (2009).
5. Paraskevopoulou, M. D., Karagkouni, D., Vlachos, I. S., Tastsoglou, S. & Hatzigeorgiou, A. G. microCLIP super learning framework uncovers functional transcriptome-wide miRNA interactions. *Nat Commun* **9**, 3601 (2018).
6. Gierliński, M. *et al.* Statistical models for RNA-seq data derived from a two-condition 48-replicate experiment. *Bioinformatics* **31**, 3625–3630 (2015).
7. Earnest-Noble, L. B. *et al.* Two isoleucyl tRNAs that decode 'synonymous' codons divergently regulate breast cancer progression. <http://biorxiv.org/lookup/doi/10.1101/2021.04.22.440519> (2021) doi:10.1101/2021.04.22.440519.
8. Passarelli, M. C. *et al.* Leucyl-tRNA synthetase is a tumour suppressor in breast cancer and regulates codon-dependent translation dynamics. *Nat Cell Biol* **24**, 307–315 (2022).
9. Huh, D. *et al.* A stress-induced tyrosine-tRNA depletion response mediates codon-based translational repression and growth suppression. *The EMBO Journal* **40**, (2021).
10. Chothani, S. *et al.* deltaTE: Detection of Translationally Regulated Genes by Integrative Analysis of Ribo-seq and RNA-seq Data. *Current Protocols in Molecular Biology* **129**, (2019).
11. Duncan, C. D. S. & Mata, J. The translational landscape of fission-yeast meiosis and sporulation. *Nat Struct Mol Biol* **21**, 641–647 (2014).
12. Blaze, J. *et al.* Neuronal Nsun2 deficiency produces tRNA epitranscriptomic alterations and proteomic shifts impacting synaptic signaling and behavior. *Nat Commun* **12**, 4913 (2021).
13. Goodarzi, H. *et al.* Modulated Expression of Specific tRNAs Drives Gene Expression and Cancer Progression. *Cell*

- 165, 1416–1427 (2016).
14. Alkallas, R., Fish, L., Goodarzi, H. & Najafabadi, H. S. Inference of RNA decay rate from transcriptional profiling highlights the regulatory programs of Alzheimer's disease. *Nat Commun* **8**, 909 (2017).
 15. Love, M. I., Huber, W. & Anders, S. Moderated estimation of fold change and dispersion for RNA-seq data with DESeq2. *Genome Biol.* **15**, 550 (2014).
 16. Winkler, J. *et al.* Dissecting the contributions of tumor heterogeneity on metastasis at single-cell resolution. <http://biorxiv.org/lookup/doi/10.1101/2022.08.04.502697> (2022)
doi:10.1101/2022.08.04.502697.
 17. Nakamura, P. A. *et al.* Small molecule Photoregulin3 prevents retinal degeneration in the RhoP23H mouse model of retinitis pigmentosa. *eLife* **6**, e30577 (2017).

Decision Letter, first revision:

Our ref: NCB-G46717A

17th January 2023

Dear Dr. Goodarzi,

Please first accept our apology for the delay getting back to you due to the backlog we have to go through after the holiday break.

Thank you for submitting your revised manuscript "An mRNA processing pathway suppresses metastasis by governing translational control from the nucleus" (NCB-G46717A). It has now been seen by the original referees and their comments are below. The reviewers find that the paper has improved in revision, and therefore we'll be happy in principle to publish it in Nature Cell Biology, pending minor revisions to satisfy the referees' final requests and to comply with our editorial and formatting guidelines.

Thank you again for your interest in Nature Cell Biology Please do not hesitate to contact me if you have any questions.

Sincerely,

Zhe Wang, PhD
Senior Editor
Nature Cell Biology

Tel: +44 (0) 207 843 4924
email: zhe.wang@nature.com

Reviewer #1 (Remarks to the Author):

The authors have done a good job addressing my concerns. I have no more issues. I suggest the authors provide a schematic showing their model.

Reviewer #2 (Remarks to the Author):

Revisions have addressed my concerns with the original submission. The new data, such as the TIA1 & ELAVL1 CLIP-Seq and the comparison between nuclear and cytoplasmic RNA, strengthen the case for a specific effect of HNRNPC on translation. The added analyses, showing highly concordant results across different analysis pipelines, likewise add to the manuscript.

Reviewer #3 (Remarks to the Author):

The authors have extensively revised this manuscript, addressing almost all of my concerns with additional data. While I still have some questions regarding some of the computational approaches, the inclusion of validations have addressed these concerns. The added functional studies have also enhanced the paper.

Author Rebuttal, first revision:

[There is no Rebuttal Letter at this stage.]

Final Decision Letter:

Dear Dr Goodarzi,

I am pleased to inform you that your manuscript, "An mRNA processing pathway suppresses metastasis by governing translational control from the nucleus", has now been accepted for publication in Nature Cell Biology.

Please note that *Nature Cell Biology* is a Transformative Journal (TJ). Authors may publish their research with us through the traditional subscription access route or make their paper immediately open access through payment of an article-processing charge (APC). Authors will not be required to make a final decision about access to their article until it has been accepted. Find out more about Transformative Journals

Authors may need to take specific actions to achieve compliance with funder and institutional open access mandates. If your research is supported by a funder that requires immediate open access (e.g. according to Plan S principles) then you should select the gold OA route, and we will direct you to the compliant route where possible. For authors selecting the subscription

publication route, the journal's standard licensing terms will need to be accepted, including self-archiving policies. Those licensing terms will supersede any other terms that the author or any third party may assert apply to any version of the manuscript.

If you have not already done so, we strongly recommend that you upload the step-by-step protocols used in this manuscript to the Protocol Exchange (www.nature.com/protocolexchange), an open online resource established by Nature Protocols that allows researchers to share their detailed experimental know-how. All uploaded protocols are made freely available, assigned DOIs for ease of citation and are fully searchable through nature.com. Protocols and Nature Portfolio journal papers in which they are used can be linked to one another, and this link is clearly and prominently visible in the online versions of both papers. Authors who performed the specific experiments can act as primary authors for the Protocol as they will be best placed to share the methodology details, but the Corresponding Author of the present research paper should be included as one of the authors. By uploading your Protocols to Protocol Exchange, you are enabling researchers to more readily reproduce or adapt the methodology you use, as well as increasing the visibility of your protocols and papers. You can also establish a dedicated page to collect your lab Protocols. Further information can be found at www.nature.com/protocolexchange/about

With kind regards,

Zhe Wang, PhD
Senior Editor
Nature Cell Biology

Tel: +44 (0) 207 843 4924
email: zhe.wang@nature.com

Click here if you would like to recommend Nature Cell Biology to your librarian

<http://www.nature.com/subscriptions/recommend.html#forms>